# *EZH2* mutations in follicular lymphoma distort H3K27me3 profiles and alter transcriptional responses to PRC2 inhibition

Pierre Romero [1,2,6], Laia Richart[1,6], Setareh Aflaki[1], Ambre Petitalot[1], Megan Burton[1], Audrey Michaud[1], Julien Masliah-Planchon[3], Frédérique Kuhnowski[4], Samuel Le Cam [1], Carlos Baliñas-Gavira [1], Céline Méaudre[2], Armelle Luscan[1], Abderaouf Hamza [3], Patricia Legoix[5], Anne Vincent-Salomon [2], Michel Wassef[1], Daniel Holoch [1] ✉ & Raphaël Margueron [1] ✉

Mutations in chromatin regulators are widespread in cancer. Among them, the histone H3 lysine 27 methyltransferase Polycomb Repressive Complex 2 (PRC2) shows distinct alterations according to tumor type. This specificity is poorly understood. Here, we model several PRC2 alterations in one isogenic system to reveal their comparative effects. Focusing then on lymphoma-associated *EZH2* mutations, we show that *Ezh2^{Y641F}* induces aberrant H3K27 methylation patterns even without wild-type *Ezh2*, which are alleviated by partial PRC2 inhibition. Remarkably, *Ezh2^{Y641F}* rewires the response to PRC2 inhibition, leading to induction of antigen presentation genes. Using a unique longitudinal follicular lymphoma cohort, we further link *EZH2* status to abnormal H3K27 methylation. We also uncover unexpected variability in the mutational landscape of successive biopsies, pointing to frequent co-existence of different clones and cautioning against stratifying patients based on single sampling. Our results clarify how oncogenic PRC2 mutations disrupt chromatin and transcription, and the therapeutic vulnerabilities this creates.

The regulation of chromatin structure is integral to the proper implementation of cell-type-specific transcriptional programs during differentiation and development. Consistent with this key role in supporting cell identity, around half of cancers display mutations in enzymes that control the spatial organization or covalent modification of chromatin[1–3].

The prevalence of oncogenic alterations affecting chromatin regulation is exemplified by Polycomb Repressive Complex 2 (PRC2), an ancient and developmentally critical methyltransferase which mono-, di- and tri-methylates histone H3 on lysine 27 (H3K27me1/2/3)[4,5]. PRC2

binding and H3K27me3 accumulation at promoter regions are strongly associated with lineage-specific gene silencing. The PRC2 core complex in mammals includes either of two paralogous catalytic subunits, Enhancer of Zeste Homolog 1 or 2 (EZH1 or EZH2), bound by two partners that are also essential for enzymatic activity, Embryonic Ectoderm Development (EED) and Suppressor of Zeste 12 (SUZ12), as well as the histone-binding protein RBBP4 or its paralog RBBP7[4,6]. Distinct types of alterations affecting PRC2 activity, including both loss- and gain-of-function mutations, have been reported in a wide range of tumor types and shown to promote tumorigenesis. Each of these genetic

[1]Institut Curie, INSERM U934/CNRS UMR 3215, Paris Sciences et Lettres Research University, Sorbonne University, Paris, France. [2]Institut Curie, Department of Pathology, Paris Sciences et Lettres Research University, Paris, France. [3]Institut Curie, Pharmacogenetics Unit, Department of Genetics, Paris Sciences et Lettres Research University, Paris, France. [4]Institut Curie, Department of Clinical Hematology, Paris Sciences et Lettres Research University, Paris, France. [5]Institut Curie, Genomics of Excellence (ICGex) Platform, Paris Sciences et Lettres Research University, Paris, France. [6]These authors contributed equally: Pierre Romero, Laia Richart. ✉e-mail: daniel.holoch@curie.fr; raphael.margueron@curie.fr

changes exhibits striking tumor-type specificity, suggesting that cell-type-specific complements of PRC2-repressed target genes constrain the nature of the alterations that are selected for in a given context[5,7].

Complete PRC2 loss of function through inactivation of either EED or SUZ12 is observed in a majority of malignant peripheral nerve sheath tumors (MPNST), as well as in a small subset of melanomas and glioblastomas[8]. Confirming the role of PRC2 as a tumor suppressor in this setting, deletion of *Suz12* in a mouse model of MPNST leads to accelerated tumor development and lethality[8]. In contrast, loss of function of PRC2 that is only partial−resulting from inactivating mutations in *EZH2* that leave the less active paralog *EZH1* intact, or alternatively from missense mutations in *EED* or *SUZ12*−is observed, albeit at lower frequencies, in several hematological malignancies of both myeloid and lymphoid lineages. Causal effects of these PRC2 alterations on tumorigenesis have been established in several mouse models[5,9]. Another type of tumor-specific partial disruption in PRC2 function arises from lysine-to-methionine or, more rarely, lysine-to-isoleucine substitutions at position 27 in one gene copy of histone H3 (*H3K27M/I*) and is highly prevalent in midline glioma, a deadly pediatric brain cancer[10–13]. While it is widely accepted that this H3K27M "oncohistone" acts dominantly to reduce overall PRC2 activity (reviewed by Mitchener and Muir[14]), some reports have linked it to a retention or even a gain in PRC2 binding and H3K27me3 at a subset of target loci[11,15–18]. Interestingly, the germline PRC2 inhibitory protein EZHIP, abnormally expressed in a majority of posterior fossa type A ependymomas, was recently shown to act through a critical peptide sequence that is mimicked by H3K27M[19].

At the other end of the spectrum, gain-of-function mutations that enhance the catalytic activity of EZH2, particularly substitutions at tyrosine 646 (Y646), are observed in a subset of B-cell lymphomas of germinal center (GC) origin, and specifically in around 25% of follicular lymphomas (FL)[20–24]. These mutations severely impair B cell differentiation following the GC reaction and alter the interactions of GC B cells with their microenvironment, events which together promote lymphomagenesis[25–27]. Intriguingly, while *EZH2*[Y646] mutations result in elevated cellular H3K27me3, the mark counterintuitively declines in abundance at existing sites of enrichment and instead becomes redistributed over much broader regions of the genome[28]. These changes in H3K27me3 have been reported to occur in a manner that is correlated within, much more than across, the boundaries of topologically-associated domains[29]. The implications of this altered distribution for FL treatment strategies are currently unclear. Indeed, while the PRC2 inhibitor tazemetostat has proved more effective in FL patients whose tumors bear *EZH2* gain-of-function mutations than in those with intact *EZH2*[30,31], the underlying molecular mechanisms have yet to be unraveled.

Distinct alterations in PRC2 function therefore appear to drive tumorigenesis in precise settings, and although they have each been investigated individually, no study has yet carefully examined their relative molecular and transcriptomic impacts using a single isogenic system. Here we report a collection of isogenic mouse cell lines bearing PRC2 alterations found in cancer and present their detailed comparative characterization. We find that *Eed*-KO, *Ezh2*-KO and *H3.3K27M*, despite disrupting PRC2 by distinct mechanisms, differ essentially in the degree of PRC2 inhibition that they cause. Meanwhile, *Ezh2*[Y641F] (corresponding to Y646 in humans) induces a redistribution of H3K27me3 which our experiments tie directly to the enzyme's elevated activity and which does not necessarily require cooperation with wild-type (WT) *Ezh2* or *Ezh1*. A pronounced rise in H3K27 acetylation (H3K27ac) is also visible in *Ezh2*[Y641F]-expressing cells, mainly at regions already enriched for this mark in WT cells. The mutant-specific H3K27me3 landscape is accompanied by complex secondary changes in the transcriptome and, importantly, in the transcriptional response to pharmacological PRC2 inhibition, notably at antigen presentation genes. In parallel, we analyzed a unique cohort of FL patient samples assembled over several decades consisting of both *EZH2*[WT] and *EZH2*[Y646]-mutant tumors and including, for some patients, biopsies at different successive relapses. Their comparison by ChIP-seq confirms the distorted H3K27me3 profile that we observed in mutant cell lines. It also reveals a strong heterogeneity in the profiles, including among longitudinal samples from the same patient, which suggests the co-existence of clonal populations that evolve differently over time. Altogether, our results reveal the essential mechanistic consequences of oncogenic PRC2 mutations, and shed light on the mechanisms that potentially underlie the enhanced vulnerability of *EZH2*-gain-of-function tumors to PRC2 inhibition as well as the appearance of resistance.

## Results

### The *H3.3K27M* oncohistone induces a straightforward PRC2 loss of function

Mutations affecting PRC2 activity are observed in a wide range of cancers and the nature of these mutations exhibits strong tumor-type specificity. The molecular consequences of these different alterations have not heretofore been investigated side by side in a single cellular model. In order to gain mechanistic insight into the intrinsic and comparative effects of PRC2 alterations in cancer, we constructed a unique set of isogenic cell lines recapitulating each relevant mutation. Taking advantage of a previously established line of *Ezh2*[flox/flox] immortalized mouse embryonic fibroblasts (iMEFs)[32], we used inducible Cre recombinase to delete *Ezh2* and CRISPR/Cas9-mediated gene editing to delete *Eed* or introduce the *H3.3K27M* substitution (Supplementary Fig. 1a, b), thereby modeling mutations found in hematological malignancies, MPNST and diffuse midline glioma, respectively.

Western blot analysis showed that deletion of *Eed* led to a dramatic loss of H3K27me1/2/3, as observed in MPNST[8], as well as a nearly complete destabilization of EZH2, as expected (Fig. 1a). In contrast, deletion of *Ezh2* led to a less severe though still pronounced reduction in H3K27me3 and to a milder drop in H3K27me2, indicating that Ezh1 partially compensates for loss of its more active paralog in these cells[33,34]. Meanwhile, *H3.3K27M* also produced a clear decrease in H3K27me2/3, consistent with previous reports of ectopic expression of this oncohistone[10,11,13], but to a lesser extent than observed in *Ezh2*-KO cells (Fig. 1a, Supplementary Fig. 1c). Examining other histone modifications, we found that monoubiquitylation of histone H2A on lysine 119 (H2AK119ub1) and trimethylation of histone H3 on lysines 4 and 36 (H3K4me3, H3K36me3) showed no obvious changes in mutant cells by Western blot (Fig. 1a). Importantly, however, levels of H3K27 acetylation (H3K27ac) rose in every mutant tested, consistent with previous studies and with the notion that H3K27 acetyltransferases are normally in competition with PRC2 over at least part of its target regions[13,35–38]. Altogether, these data suggest that each of these alterations results in a PRC2 loss of function whose magnitude increases in the following order: *H3.3K27M*, *Ezh2*-KO, *Eed*-KO. In agreement with this order of severity, the three alterations led to up-regulation of increasing numbers of H3K27me3-marked PRC2 target genes in sets that exhibited a high degree of overlap (Fig. 1b, c, Supplementary Fig. 1d–f).

The accumulation of H3K27me3 across the genome was also impaired according to the same trend, with *H3.3K27M*, *Ezh2*-KO and *Eed*-KO exhibiting progressively stronger depletion of the mark from peak regions identified in WT cells (Fig. 1d, Supplementary Fig. 1g). Several studies have reported that cells expressing the H3K27M oncohistone show a paradoxical pattern of global H3K27me3 losses accompanied by local H3K27me3 retention or even gains leading to stronger transcriptional repression[11,15–18]. In our isogenic model, however, H3K27me3 losses are widespread, and individual H3K27me3 peaks almost universally display a substantial reduction in enrichment (Fig. 1e, f, Supplementary Fig. 1h). This is remarkably similar to what is observed in WT cells treated with non-saturating-doses of the PRC2

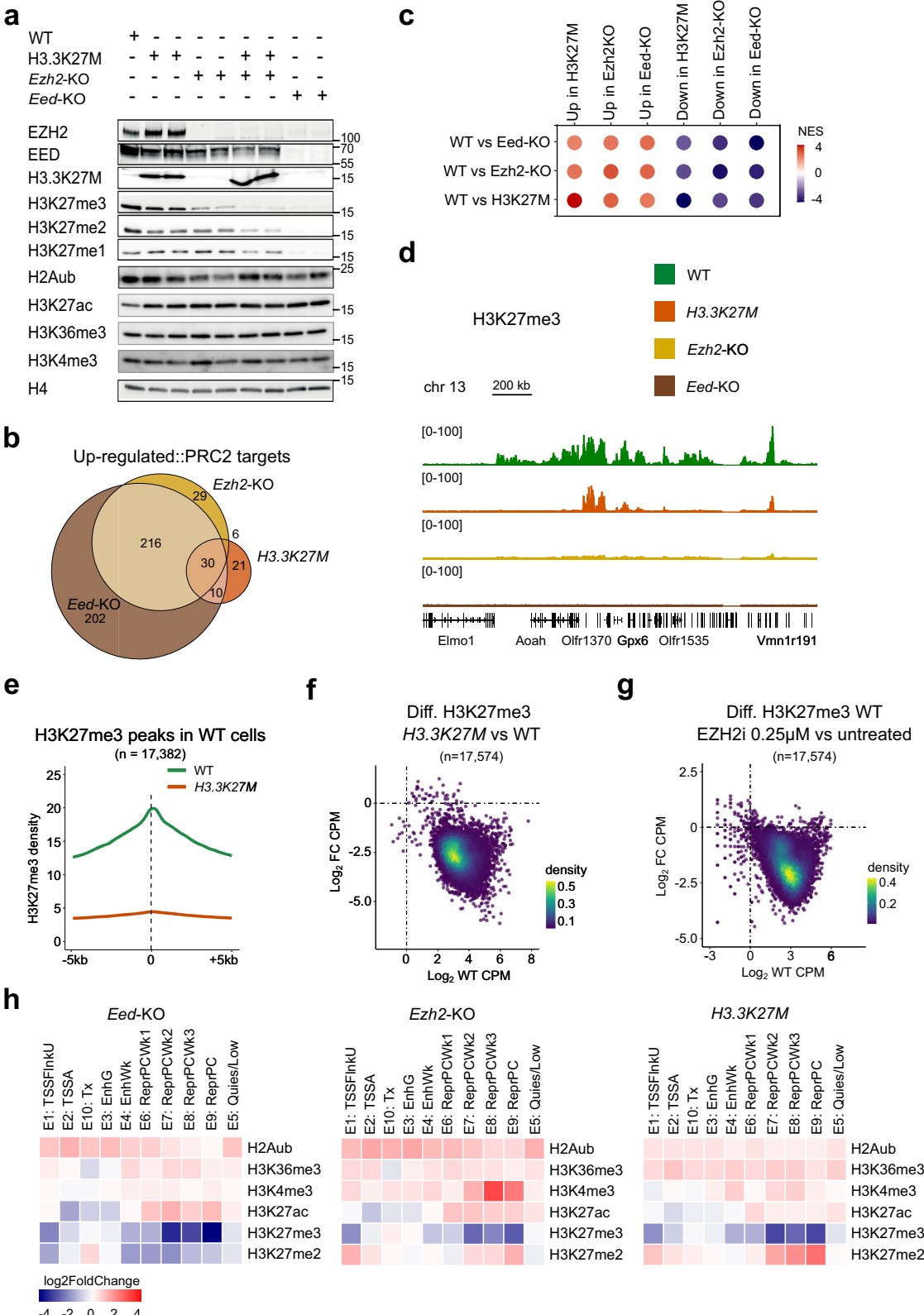

pharmacological inhibitor UNC1999 (Fig. 1g, Supplementary Fig. 1h, i). We conclude that the inherent consequence of the *H3.3K27M* mutation on H3K27me3 is a straightforward loss of function of PRC2, with paradoxical local effects representing a specificity of certain cellular contexts rather than a general property of H3K27M.

In order to gain insight into the broader impact of PRC2 alterations on the chromatin landscape, we performed comparative chromatin state discovery and annotation analysis using ChromHMM[39]. A panel of genome-wide profiling experiments for histone modifications associated with transcriptional activation (H3K27ac, H3K36me3, and H3K4me3) and repression (H2AK119ub1, H3K27me2, and H3K27me3) allowed us to distill nine ChromHMM states encompassing distinct regions of active and inactive chromatin. A tenth state (E5:Quies/Low), characterized by absent or low levels of the histone modifications we

**Fig. 1 | The *H3.3K27M* mutation induces a straightforward PRC2 loss of function. a** Western blot analysis of the indicated proteins and post-translational modifications in isogenic iMEFs of the indicated genotypes. Identical samples were run in parallel on different blots. Source data are provided as a Source Data file. **b** Venn diagrams showing the overlap among PRC2 target genes (see text) identified as up-regulated in iMEFs of each indicated genotype with respect to WT. **c** Dot plot summarizing the results of pre-ranked gene-set enrichment analyses assessing the ranked enrichment of all genes up-or down-regulated in each mutant condition (indicated in rows) among the sets of all genes up-or down-regulated in the other conditions (indicated in columns). NES, normalized enrichment score. **d** H3K27me3 CUT&RUN tracks for iMEFs of the indicated genotypes with corresponding genomic annotations shown below. **e** Mean density plot of H3K27me3 CUT&RUN reads detected in WT and *H3.3K27M* iMEFs across regions identified as H3K27me3 peaks in WT cells. **f** MA plot showing the ratio of H3K27me3 enrichment in *H3.3K27M* iMEFs with respect to WT within all individual H3K27me3 peak regions detected in either or both conditions. Reads are counted over intervals spanning 2 kb on either side of each peak summit. Density scale shows the relative number of peak regions per unit area of the plot. CPM counts per million, FC fold change. **g** MA plot showing the ratio of H3K27me3 enrichment in WT iMEFs treated with 0.25 μM of the EZH2 inhibitor UNC1999 with respect to untreated WT cells within the same peak regions as those defined in (**f**) and computed as in (**f**). **h** Heatmaps showing the relative changes in enrichment for the indicated histone modifications across 10 emission states defined by ChromHMM (see Supplementary Fig. 1j) for iMEFs of each indicated mutant genotype with respect to WT.

analyzed, covers the majority of the genome (Supplementary Fig. 1j). As expected, *Eed*-KO cells display loss of H3K27me3 across all ChromHMM states, with the fold changes of greatest magnitude occurring, consistently, within Polycomb-repressed states whose WT H3K27me3 enrichment is highest (Fig. 1h, Supplementary Fig. 1j). These repressed states are also characterized by a loss of H3K27me2 in *Eed*-KO cells. In *Ezh2*-KO and *H3.3K27M* cells, reductions in H3K27me3 over Polycomb-repressed states are less severe though still quite prominent; however, unlike in *Eed*-KO cells these states show an increase in H3K27me2 (Fig. 1h). This is consistent with an incomplete loss of function of PRC2 whose residual activity is still capable of catalyzing H3K27me2 at target regions, and in keeping with previous reports of H3K27me2 adopting a H3K27me3-like distribution in the presence of H3K27M or EZHIP[17,40,41].

Marks of active chromatin such as H3K27ac, H3K36me3 and H3K4me3 conversely showed a general tendency to rise, in particular over Polycomb-repressed states, although H3K27ac very strikingly declined within ChromHMM states corresponding to transcription start sites, transcribed regions and genic enhancers (Fig. 1h, Supplementary Fig. 1j). We further verified this pattern by observing that H3K27ac increased across the majority of genomic bins in mutant cells while concurrently decreasing in regions identified as peaks (Supplementary Fig. 1k, l). A global increase in H3K27ac in cells expressing H3K27M has been reported previously[13,36,42,43] and has also been found to occur pervasively at the expense of enrichment at genes and enhancers[15,44], suggesting that our findings reflect an intrinsic, cell-type-independent effect of the oncohistone. It is important to note, however, that because this shift in H3K27ac patterns is also observed in *Ezh2*-KO and *Eed*-KO cells, it is best understood as a general consequence of PRC2 loss of function rather than a unique property of *H3K27M*.

Altogether, our analysis of isogenic iMEFs with alterations affecting PRC2 activity reveals that *H3.3K27M* represents a straightforward loss-of-function mutation. Indeed, *H3.3K27M* results in changes in the H3K27me3 landscape and the transcriptome that are of a similar nature albeit less pronounced than those observed upon deletion of *Ezh2*, which in turn are similar to but less severe than those that occur upon deletion of *Eed*. Furthermore, the three mutations produce similar changes in the distribution of H3K27ac with dispersed gains overall coinciding with reductions at peaks.

### *Ezh2*[Y641F] alone induces an abnormal H3K27me3 landscape that is corrected by partial PRC2 inhibition

We next turned to *EZH2* gain-of-function mutations that enhance the catalytic activity of PRC2 and are present in a large subset of FL[20–24]. In order to model these alterations we used CRISPR/Cas9-mediated gene editing to introduce the *Ezh2-Y641F* substitution into iMEFs (Supplementary Fig. 2a, b). Previous in vitro studies have concluded that while such mutant forms of EZH2 are very adept at catalyzing the transition from H3K27me2 to H3K27me3, they are very inefficient at adding the first methyl group to unmethylated

H3K27 residues[23,45,46]. Consequently the excess cellular H3K27me3 found in *EZH2*-mutant tumors was proposed to require the complementary activities of the WT and mutant enzymes[23,45,46]. To test this hypothesis, we generated *Ezh2*[Y641F] iMEF subclones lacking the paralog *Ezh1* and used Cre-mediated recombination to selectively excise WT *Ezh2*[flox], thus allowing us to evaluate the properties of PRC2-EZH2[Y641F] in the presence and absence of WT forms of PRC2 (Supplementary Fig. 2b).

In agreement with previous studies[23,45,46], we found that *Ezh2*[Y641F] was sufficient to induce a sharp increase in total cellular H3K27me3, as assessed by Western blot, and a concomitant decrease in H3K27me2 (Fig. 2a). Remarkably, these effects can be mediated autonomously by PRC2-EZH2[Y641F], since they persist even after deletion of *Ezh2*[WT] and *Ezh1* (Fig. 2a). We conclude that, unexpectedly, *Ezh2*[Y641F] is an intrinsic gain-of-function mutation that does not require cooperation with a WT counterpart to trigger alterations in the cellular balance of H3K27me2/3.

Although this finding is in apparent contradiction with earlier reports that mutant EZH2 is minimally active on unmethylated H3K27 in vitro[23,45,46], it has also been noted that EZH2[Y641F] shows somewhat higher residual activity than other lymphoma-associated variants such as EZH2[Y641N] [23]. We therefore assessed whether expression of *Ezh2*[Y641N] was also sufficient for H3K27me3 accumulation in cells in the absence of *Ezh2*[WT] and *Ezh1*. In agreement with the earlier in vitro findings[23], however, cells overexpressing *Ezh2*[Y641N] could not restore H3K27me3 in an *Ezh1-Ezh2*-double-KO background, in contrast to their *Ezh2*[Y641F] counterparts (Fig. 2b). Thus, the intrinsic residual activity of PRC2-EZH2[Y641F] likely contributes to the robust H3K27 trimethylation observed in *Ezh1*-KO; *Ezh2*[Y641F/KO] cells. Yet, because this intrinsic activity has been reported to be very modest[23], we hypothesized that PRC2 accessory subunits present in the cellular context[47,48] might enable the mutant enzyme to attain higher levels of activity and thereby offset its difficulty engaging with unmethylated substrates. To test this idea, we conducted histone methyltransferase assays on recombinant nucleosomes using recombinant PRC2, AEBP2, and JARID2 purified from insect cells. Methylation of nucleosomal histone H3 by PRC2-EZH2[Y641F] could not readily be detected, consistent with previous results[23,45,46], but addition of AEBP2 and JARID2 stimulated both the WT and the mutant enzymes, allowing the latter to exceed the activity shown by the WT in the absence of these accessory factors (Fig. 2c). By contrast, PRC2-EZH2[Y641N] exhibited minimal activity under these conditions, even in the presence of stimulatory H3K27me3 peptides[49] (Fig. 2c). Thus, we conclude that EZH2[Y641F], but not EZH2[Y641N], is capable of carrying out initial methylation of unmethylated H3K27 independently of EZH2[WT] and propose that accessory subunits may contribute to this process.

Next, given the increased global abundance of H3K27me3 in cells expressing *Ezh2*[Y641F] (Fig. 2a), we examined the impact of the mutation on the mark's genome-wide landscape and found sharp declines at most WT peaks of enrichment accompanied by broadly distributed gains elsewhere (Fig. 2d, Supplementary Fig. 2c). These effects were much greater in magnitude than any differences due to the choice of

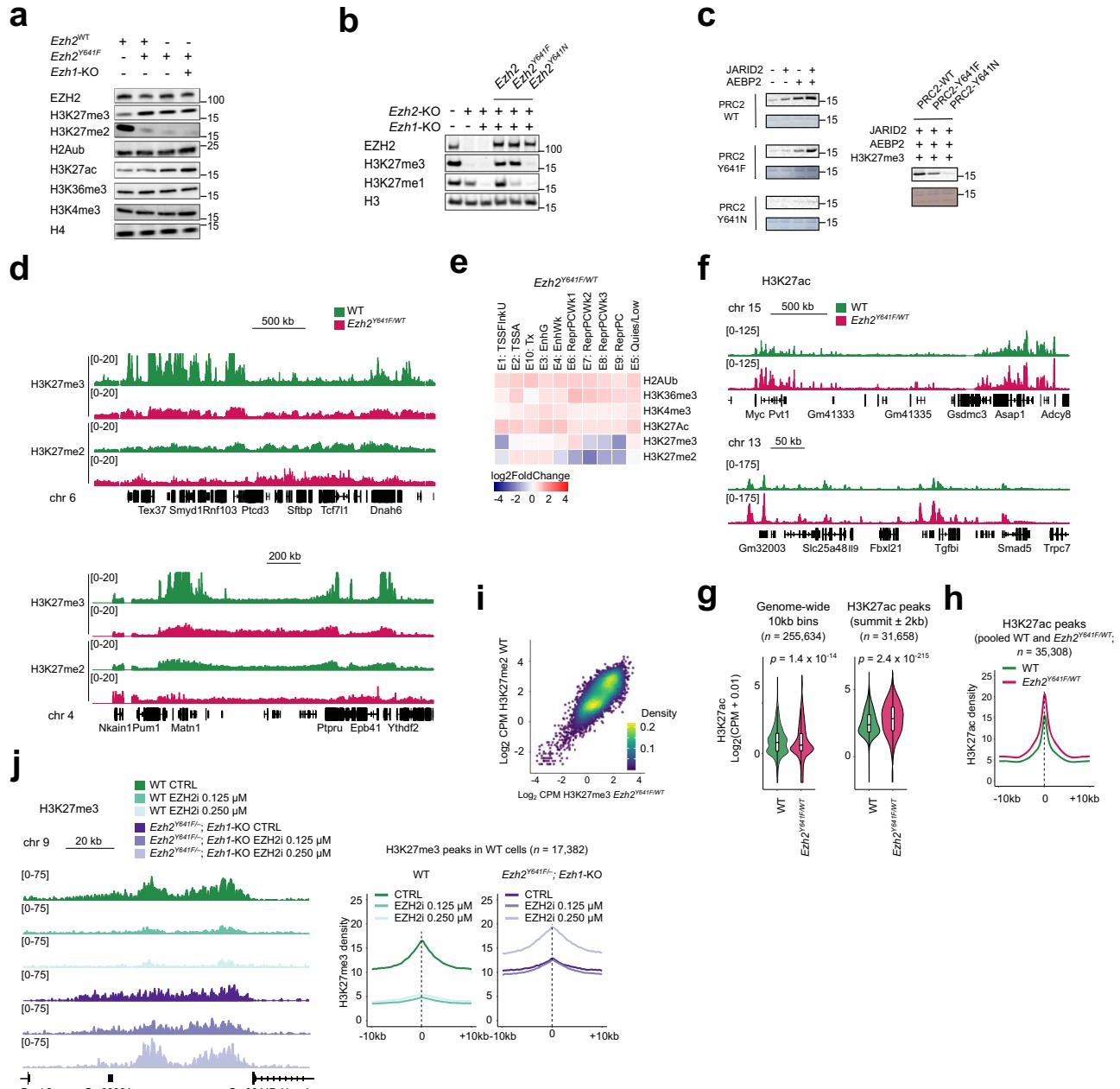

**Fig. 2 | *Ezh2^Y641F* is sufficient to drive aberrant H3K27me3 patterns and strengthened H3K27ac peaks. a**, **b** Western blot analysis in isogenic iMEFs. Identical samples were run in parallel on different blots. Source data are provided as a Source Data file. **c** In vitro histone methyltransferase assay using [3]H-labeled *S*-adenosyl-methionine comparing the activity of reconstituted PRC2 variants on recombinant nucleosomes in the presence of JARID2, AEBP2 or H3K27me3 peptide. Source data are provided as a Source Data file. **d** H3K27me3 and H3K27me2 CUT&RUN tracks for WT and *Ezh2^Y641F/WT* iMEFs with corresponding genomic annotations shown below (note that the H3K27me3 signal for WT cells exceeds the range displayed). **e** Heatmap of changes in enrichment for histone modifications across 10 emission states defined by ChromHMM (see Supplementary Fig. 1j) comparing *Ezh2^Y641F/WT* iMEFs to WT. **f** H3K27ac CUT&RUN tracks for WT and *Ezh2^Y641F/WT* iMEFs with corresponding genomic annotations shown below. **g** Violin

and box plots showing normalized H3K27ac CUT&RUN read counts for WT and *Ezh2^Y641F/WT* iMEFs within all 10-kb bins genome-wide (*n* = 255,634 bins) (left) or within H3K27ac peaks identified in WT cells (*n* = 31,658 peaks) (right). Boxes represent median and first and third quartiles, while whiskers show minimum and maximum. *p*-values represent the result of unpaired two-sided *t*-tests. **h** Mean density plot of H3K27ac CUT&RUN reads detected in WT and *Ezh2^Y641F/WT* iMEFs across H3K27ac peaks identified in either or both conditions. **i** Scatterplot showing H3K27me2 CUT&RUN read counts in WT iMEFs *versus* H3K27me3 CUT&RUN read counts in *Ezh2^Y641F/WT* iMEFs for all 10-kb bins genome-wide. **j** Left, H3K27me3 CUT&RUN tracks for WT and *Ezh2^Y641F/−*; *Ezh1*-KO iMEFs, either untreated (CTRL) or treated with the EZH2 inhibitor UNC1999, with corresponding genomic annotations shown below. Right, Mean density plot of H3K27me3 CUT&RUN reads detected in each condition within H3K27me3 peaks identified in WT cells.

normalization method (Supplementary Table 1), and coincided with a reduction in the enrichment of PRC2 at its target peaks as evaluated by SUZ12 profiling (Supplementary Fig. 2d). This flattened occupancy pattern has previously been observed in B cells and melanoma cells[28] and thus appears to be an intrinsic effect of the mutation. Meanwhile,

H3K27me2 is substantially reduced in *Ezh2^Y641F/WT* (Fig. 2a, d, Supplementary Fig. 2c).

A systematic analysis of the chromatin landscape in *Ezh2^Y641F/WT* cells according to the ChromHMM states defined above further underscores these changes in the distributions of H3K27me3 and

H3K27me2, with H3K27me3 falling within most Polycomb-repressed states while generally rising elsewhere, and H3K27me2 displaying more pronounced and widespread declines (Fig. 2e). Notably, the two states that cover the largest share of the genome (E5:Quies/Low and E6:Repressed-Polycomb-weak) both gain H3K27me3 and lose H3K27me2, in keeping with the global trend. The ChromHMM analysis also reveals a slight increase in H3K27ac genome-wide, which, in contrast to the PRC2 loss-of-function cases examined above, is most discernible within states corresponding to transcribed regions and enhancers. In agreement with this, we find that H3K27ac rises appreciably in $Ezh2^{Y641F/WT}$ cells at H3K27ac peaks present in WT cells (Fig. 2f–h), while displaying a more modest increase over the whole genome (Fig. 2g).

We next sought to understand the basis for the spatial spreading of H3K27me3 in $Ezh2^{Y641F/WT}$ cells. Given the observation that in $H3.3K27M$ cells the profile of H3K27me2 adopts that normally displayed by H3K27me3 in WT cells (this study and ref. 40), we reasoned that the opposite phenomenon might explain the pattern adopted by H3K27me3 in $Ezh2^{Y641F/WT}$ cells. Indeed, Fig. 2d and Supplementary Fig. 2c show that the distribution of H3K27me3 in $Ezh2^{Y641F/WT}$ cells closely resembles that of H3K27me2 in WT cells. Consistently, we found that H3K27me3 enrichment in $Ezh2^{Y641F/WT}$ correlates more strongly with WT H3K27me2 genome-wide than does WT H3K27me3 (Fig. 2i, Supplementary Fig. 2e). Altogether this suggests that $Ezh2^{Y641F}$ expression triggers invasion and replacement of H3K27me2 by H3K27me3.

While these observations account for the broad gains of H3K27me3 that characterize $Ezh2^{Y641F/WT}$ cells, they do not explain the marked local declines that occur at the regions where WT H3K27me3 enrichment is highest. We therefore asked next whether the totality of the aberrant H3K27me3 landscape (broad gains *and* local declines) can be directly attributed to the elevated catalytic activity of PRC2-EZH2$^{Y641F}$. To this end, we tested whether partially inhibiting the mutant enzyme restores a WT profile. We conducted this experiment using $Ezh2^{Y641F/-}$ cells lacking WT $Ezh2$ and $Ezh1$ in order to isolate the specific effects of catalytic inhibition of mutant PRC2. Similarly to the global increase in H3K27me3, EZH2$^{Y641F}$ did not require cooperation from WT EZH2 or EZH1 to generate a flattened H3K27me3 pattern (Fig. 2j). We inhibited PRC2 by culturing $Ezh2^{Y641F/-}$ cells in the presence of a dose of UNC1999 sufficient to lower total H3K27me3 to the range observed in WT cells by Western blot (Supplementary Fig. 2f). Remarkably, this treatment caused H3K27me3 peaks to regain their WT shape and height; in contrast, a similar dose applied to WT cells led to sharply reduced enrichment that became confined to H3K27me3 nucleation sites (Fig. 2j, Supplementary Fig. 2g)[50]. These data provide strong evidence that the elevated catalytic activity of EZH2$^{Y641F}$ is necessary and sufficient to produce the aberrant H3K27me3 landscape observed in cells containing this gain-of-function mutant.

## $Ezh2^{Y641F}$ alters the transcriptome and its response to PRC2 inhibition

We next sought to determine how the extensive rearrangement of H3K27me3 in cells expressing $Ezh2^{Y641F}$ affects transcription. To deepen this analysis we compared transcriptional differences to those induced by PRC2 inhibition, taking advantage of the fact that UNC1999 treatment had no effect on the growth of our cell line regardless of $Ezh2$ status (Supplementary Fig. 3a), thus allowing us to separate its effects on transcriptional regulation from any potential impact on cell viability. We reasoned that the widespread losses and gains of the repressive H3K27me3 mark described above would result in numerous genes experiencing either increased or decreased transcriptional activity. Consistently, we found hundreds of genes either up-regulated or down-regulated in $Ezh2^{Y641F/WT}$ cells compared to WT controls (Fig. 3a). However, in the case of genes whose transcript levels increased, this did not appear to be primarily due to loss of PRC2-dependent

repression, as they showed little overlap with genes up-regulated in WT cells treated with a high dose of UNC1999 (Fig. 3b, top). Conversely, for genes whose transcript levels decreased in $Ezh2^{Y641F/WT}$ cells, this was generally not due to de novo PRC2-dependent repression, as they largely failed to rise again in $Ezh2^{Y641F/WT}$ cells treated with the high dose of UNC1999 (Fig. 3b, bottom). We conclude that most transcriptional changes do not result directly from local shifts in PRC2 activity, but instead represent complex secondary consequences of the aberrant H3K27 methylation landscape. Thus, the impact of the EZH2$^{Y641F}$ mutant enzyme on global gene expression is profound but mainly indirect.

FL tumors bearing *EZH2* gain-of-function mutations display a stronger response to pharmacological PRC2 inhibition than do those lacking *EZH2* mutations, a finding whose mechanistic basis is not well understood[30,31]. We took advantage of our isogenic model to gain insight into the impact of the $Ezh2^{Y641F}$ mutation on the response to PRC2 inhibition at the transcriptomic level. We found that both moderate and high doses of UNC1999 led mainly to transcriptional up-regulation events (Supplementary Fig. 3b), as expected, but that these involved remarkably disparate sets of genes in WT and $Ezh2^{Y641F/WT}$ cells (Fig. 3c). This analysis demonstrates that the $Ezh2^{Y641F}$ mutation not only substantially alters the transcriptome but also reprograms its response to PRC2 inhibition. We considered the possibility that PRC2 inhibition in mutant cells might act to correct the transcriptional changes caused by $Ezh2^{Y641F}$ expression. However, the failure of the inhibitor treatment to restore expression of genes down-regulated in $Ezh2^{Y641F/WT}$ cells, as noted above (Fig. 3b, bottom), argues against this scenario. Accordingly, a principal component analysis (PCA) reveals that the transcriptome of $Ezh2^{Y641F/WT}$ cells remains distant from that of WT cells regardless of the dose of inhibitor (Fig. 3d). We conclude that the inhibitor treatment does not restore WT gene expression patterns in $Ezh2^{Y641F/WT}$ cells but instead leads to the up-regulation of target genes distinct from those up-regulated in WT cells.

We further examined these differential transcriptomic responses to PRC2 inhibition by performing unsupervised hierarchical clustering of genes up-regulated only in WT cells, those up-regulated only in $Ezh2^{Y641F/WT}$ cells and the small subset up-regulated in both (Fig. 3e). This revealed four major clusters among the genes whose behavior diverges between the two conditions, which can be classified according to their PRC2 inhibitor response and expression status in each genotype. As remarked above (Fig. 3b), genes whose baseline expression levels are altered by the mutation (clusters 1 and 4) are generally not those whose response to the inhibitor treatment most strongly suggests that they lose (cluster 2) or gain (cluster 3) PRC2-dependent regulation (Fig. 3e). This further underscores the predominantly indirect effects of the mutant enzyme on transcription. In agreement with this notion, the genes with the strongest differential responses to PRC2 inhibition (clusters 2 and 3) did not show major differences in H3K27me3 enrichment in WT and $Ezh2^{Y641F/WT}$ cells (Fig. 3f), pointing instead to complex changes in the transcriptional circuitry of mutant cells.

To better understand the nature of these changes, we analyzed the enrichment of transcription factor (TF) binding motifs on the promoters of genes found to be differentially expressed in $Ezh2^{Y641F/WT}$ compared to WT cells. Overrepresented motifs included those recognized by many TFs which are themselves up-regulated in mutant cells and some of which are predicted to form a protein-protein interaction network (Fig. 3g, h). Intriguingly, among these TFs are several apparent direct targets of PRC2, two of which lie at the core of the interaction network and retain an active, H3K27me3-free state in $Ezh2^{Y641F/WT}$ cells even upon EZH2 inhibition (Fig. 3g, h, Supplementary Fig. 3c; compare to Fig. 2j). These findings illustrate how transcriptional networks can become persistently altered upon mutation of $Ezh2$ and help explain why gene expression changes may extend well beyond H3K27me3-marked loci to the targets of dysregulated TFs.

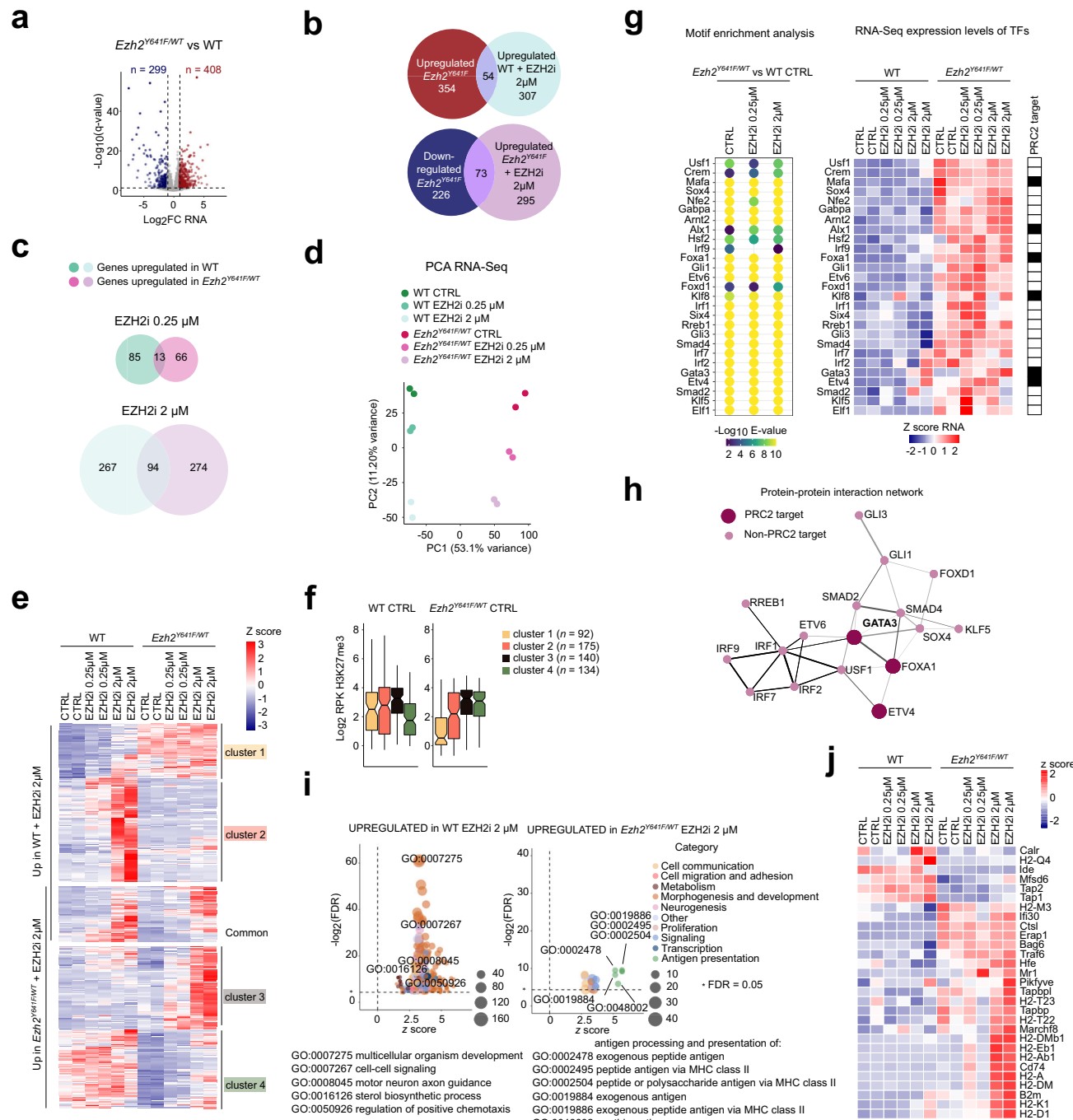

**Fig. 3 | *Ezh2^Y641F* induces complex changes in the transcriptome and its response to PRC2 inhibition. a** Comparative gene expression between *Ezh2^Y641F/WT* and WT iMEFs highlighting significantly up-regulated (red) and down-regulated (blue) genes. FC, fold change. **b** Top, Overlap between genes up-regulated in *Ezh2^Y641F/WT* *versus* WT iMEFs and those up-regulated in 2-μM-UNC1999-treated WT iMEFs. Bottom, Overlap between genes down-regulated in *Ezh2^Y641F/WT* *versus* WT iMEFs and those up-regulated in 2-μM-UNC1999-treated *Ezh2^Y641F/WT* iMEFs. **c** Overlap between genes up-regulated in WT and *Ezh2^Y641F/WT* iMEFs treated with 0.25 μM (top) or 2 μM (bottom) UNC1999. **d** PCA of variance among RNA-seq of WT and *Ezh2^Y641F/WT* iMEFs in the indicated treatment conditions. **e** Heatmap representing RNA-seq expression *z*-scores for all genes up-regulated in either WT or *Ezh2^Y641F/WT* iMEFs upon 2-μM UNC1999 treatment. Genes are ordered in clusters according to genotype-specific expression level and up-regulation. **f** Box plots showing gene-length-normalized average H3K27me3 enrichment in untreated WT and *Ezh2^Y641F/WT* iMEFs for each

cluster shown in (**e**) (*n* = 92, 175, 140, and 134 genes, respectively). Boxes represent median and first and third quartiles, while whiskers show minimum and maximum. RPK, read per kilobase. **g** Left, results of MEME motif enrichment analysis over promoters of genes differentially expressed in *Ezh2^Y641F/WT* *versus* WT iMEFs focusing on TFs whose expression is up-regulated. Right, heatmap representing RNA-seq expression *z*-scores for the genes encoding these TFs. TF, transcription factor. TSS, transcription start site. **h** Protein-protein interaction network built from the TFs identified in (**g**) with STRING and Cytoscape, with the thickness of each edge reflecting the degree of evidence for interaction. **i** Bubble plots showing GO terms significantly enriched among genes up-regulated in 2-μM-UNC1999-treated WT and *Ezh2^Y641F* iMEFs. GO, gene ontology. **j** Heatmap representing RNA-seq expression *z*-scores for genes corresponding to GO:0019886, the most highly enriched GO term among genes up-regulated in 2-μM-UNC1999-treated *Ezh2^Y641F* iMEFs.

We verified that our results in iMEFs were also representative of lymphoma by assessing the transcriptomic effects of PRC2 inhibition in *EZH2*-WT and *EZH2*$^{Y646F}$-expressing OCI-Ly19 cells[29]. OCI-Ly19 was derived from a germinal-center-B-cell-like diffuse large B-cell lymphoma (DLBCL), the DLBCL subtype in which, as in FL, *EZH2* gain-of-function mutations are specifically found[22], but since OCI-Ly19 itself is *EZH2*-WT it provides an opportunity to model the effect of acquiring an *EZH2* mutation in the disease context. Ectopically expressed *EZH2*$^{Y646F}$ leads to increased global H3K27me3[29], but both *EZH2*-WT and *EZH2*$^{Y646F}$-expressing cells exhibited severe loss of the mark when treated with high doses of either UNC1999 or the clinically approved PRC2 inhibitor tazemetostat (also called EPZ6438) (Supplementary Fig. 3d). Within each of the two *EZH2* backgrounds, the two inhibitors produced very similar alterations to the global transcriptome (Supplementary Fig. 3e), indicating that UNC1999 and tazemetostat affect transcriptional regulation comparably in cell culture. As we observed in iMEFs, however, PRC2 inhibition in *EZH2*-mutant cells did not seem to restore an *EZH2*-WT-like transcriptomic profile. Instead, we once again found large classes of genes whose responses to PRC2 inhibition sharply differed depending on *EZH2* status, with some genes becoming up-regulated only in a WT context and others only in a mutant context (Supplementary Fig. 3f). We conclude that major alterations in the transcriptional response to PRC2 inhibition represent an intrinsic property of *EZH2*$^{Y646F}$/*Ezh2*$^{Y641F}$ mutants, whose manifestation includes but is not limited to the lymphoma setting.

Gene ontology (GO) analysis showed that categories relating to the regulation of development, transcription and proliferation are over-represented among the genes up-regulated in WT iMEFs upon PRC2 inhibition (Fig. 3i, left, Supplementary Fig. 3g, clusters 1 and 2, Supplementary Data 1), in line with the canonical repressive function of the complex. Remarkably, the categories most enriched among genes that respond to PRC2 inhibition in *Ezh2*$^{Y641F/WT}$ iMEFs are those related to antigen presentation (Fig. 3i, right, Supplementary Fig. 3g, cluster 3, Supplementary Data 1). A detailed review of the genes belonging to these annotations reveals that a majority of MHC genes (known as H2 complex genes in mouse) are the most differentially activated by PRC2 inhibition in *Ezh2*$^{Y641F/WT}$ compared to WT cells (Fig. 3j). PRC2-dependent repression of antigen processing pathways has previously been identified as a conserved mechanism for immune evasion by tumor cells that can potentially be counteracted by PRC2 inhibition, including in lymphoma cells carrying *EZH2* gain-of-function mutations[51–53]. It is therefore intriguing to observe that *Ezh2*$^{Y641F}$ expression is sufficient in our iMEF model to render these pathways amenable to this mode of activation. Although genes up-regulated in *EZH2*-mutant OCI-Ly19 upon PRC2 inhibition were not statistically enriched for the MHC category per se, they did include a more broadly defined category of immunity-related genes (Supplementary Fig. 3h). This lends support to the possibility that sensitization of immune pathways to activation through PRC2 inhibition could be a fundamental property of *EZH2* gain-of-function mutations.

Overall, our isogenic system reveals that the gain-of-function mutant *Ezh2* frequently found in FL causes H3K27me3 to become broadly redistributed independently of WT *Ezh2* and that a WT-like average profile is restored in mutant cells subjected to partial PRC2 inhibition. This result establishes a direct link between elevated catalytic activity and altered H3K27me3 landscape. We also find major transcriptomic changes that primarily reflect the indirect impact of a shifting chromatin landscape. Finally, our data shed light on the potential basis for the differential efficacy of pharmacological PRC2 inhibition in WT and *EZH2*-mutant FL, as mutant iMEF and OCI-Ly19 cells show a substantially altered transcriptomic response that includes activation of antigen presentation genes.

### *EZH2*-mutant follicular lymphomas systematically display altered H3K27me3 profiles

In light of the dramatic changes in histone modification patterns we observed in iMEFs upon *Ezh2*$^{Y641F}$ expression, we were eager to assess whether these findings extend to FL tumors that carry equivalent *EZH2* mutations, along with other mutations affecting chromatin-modifying genes. Our current understanding of how mutant *EZH2* influences chromatin states in FL relies essentially on inferences from chromatin immunoprecipitation (ChIP) experiments carried out using mouse models[25,26,28]. These experiments, together with extensive transcriptomic analyses of human tumors, have shed light on the ability of mutant *EZH2* to modulate the composition of the FL immune niche. To our knowledge, however, no study has directly investigated the histone modification landscape in human FL as a function of *EZH2* mutation status.

In order to fill this critical gap in the field we assembled a retrospective cohort of 160 FL patients who were followed at our institution and whose biopsies were collected in many cases at multiple time points spanning several years of disease progression (Fig. 4a, b). We determined using Sanger sequencing that 19.4% of cases (31/160) carried gain-of-function mutations in *EZH2*, consistent with previously reported rates (Supplementary Table 2)[21,22,24,54]. Y646N was the most frequent variant, representing nearly half of *EZH2*-mutant cases (Supplementary Table 3). We then narrowed our cohort to 29 patients (10 *EZH2*-WT and 19 *EZH2*-mutant) on the basis of available material. A subset of new FL diagnoses at our institution involve patients initially treated for breast cancer, which accounts for the skewed sex ratio in our cohort (~2.5:1 female:male) (Supplementary Tables 4 and 5).

We used Determination of 571 Relevant Altered Genes in Oncology by NGS (DRAGON), an in-house tool[55], to comprehensively characterize genetic anomalies within our restricted cohort of 29 patients (Supplementary Data 2). This analysis confirmed the *EZH2* mutations uncovered by Sanger sequencing. Four cases contained multiple mutations in *EZH2*: in addition to mutations at Y646, we detected a splice-site mutation inducing no protein changes (1 case), E641D (1 case), and V637A (1 case). Finally, one case simultaneously carried 10% Y646F and 22.8% Y646N mutant alleles. All *EZH2* mutations were heterozygous, with a median Variant Allelic Frequency (VAF) of 21.54% [7.4–51.8]. In addition and as expected, the majority of both *EZH2*-WT and *EZH2*-mutant patients displayed mutations in either *CREBBP* or *KMT2D*, or both (Supplementary Fig. 4a), in agreement with the well-established prevalence of lesions affecting all three of these chromatin regulators in FL, often in combination[54,56,57]. Copy number variation (CNV) analysis performed in 27 cases (2 were non-assessable) revealed the presence in at least 3 cases each of 1p cnLOH (copy-number-neutral loss of heterozygosity), 2p+, 6p cnLOH, 7p+, 8q+, 10q−, 12+, 12q+, 16+, 16p cnLOH, and 17q+. We did not, however, find evidence of *EZH2* amplification or statistically significant co-occurrence of specific CNV abnormalities with EZH2 mutations (Supplementary Fig. 4b).

We performed RNA-seq in a total of 28 cases in order to collectively compare our *EZH2*-mutant tumor samples to their *EZH2*-WT counterparts and detected 211 differentially expressed genes (DEGs) (Fig. 4c). Importantly, a PCA using this set of DEGs did reasonably well at segregating *EZH2*-WT and *EZH2*-mutant tumors from an independent, previously reported cohort, the Primary RItuximab and MAintenance (PRIMA) study (Supplementary Fig. 4c)[24,58]. Furthermore, gene set enrichment analysis revealed a striking correlation between our cohort and the PRIMA cohort for GO terms depleted in *EZH2*-mutant *versus* *EZH2*-WT transcriptomes, which relate to mature B-cell functions (Fig. 4d). Intriguingly, antigen-presentation-related pathways were not significantly repressed in *EZH2*-mutant compared to *EZH2*-WT tumors in our cohort, in contrast to what has been highlighted elsewhere in cell line models[52], but this was also true of the PRIMA cohort. Overall, these analyses demonstrate that our tumor samples exhibit transcriptomic disparities as a function of *EZH2* mutation status that are consistent with established observations.

We conducted H3K27me3 ChIP-seq in duplicate on a more limited subset consisting of 4 *EZH2*-WT and 8 *EZH2*-mutant samples (Fig. 4a, b), choosing the latter on the basis of high VAF (Supplementary Table 6) in

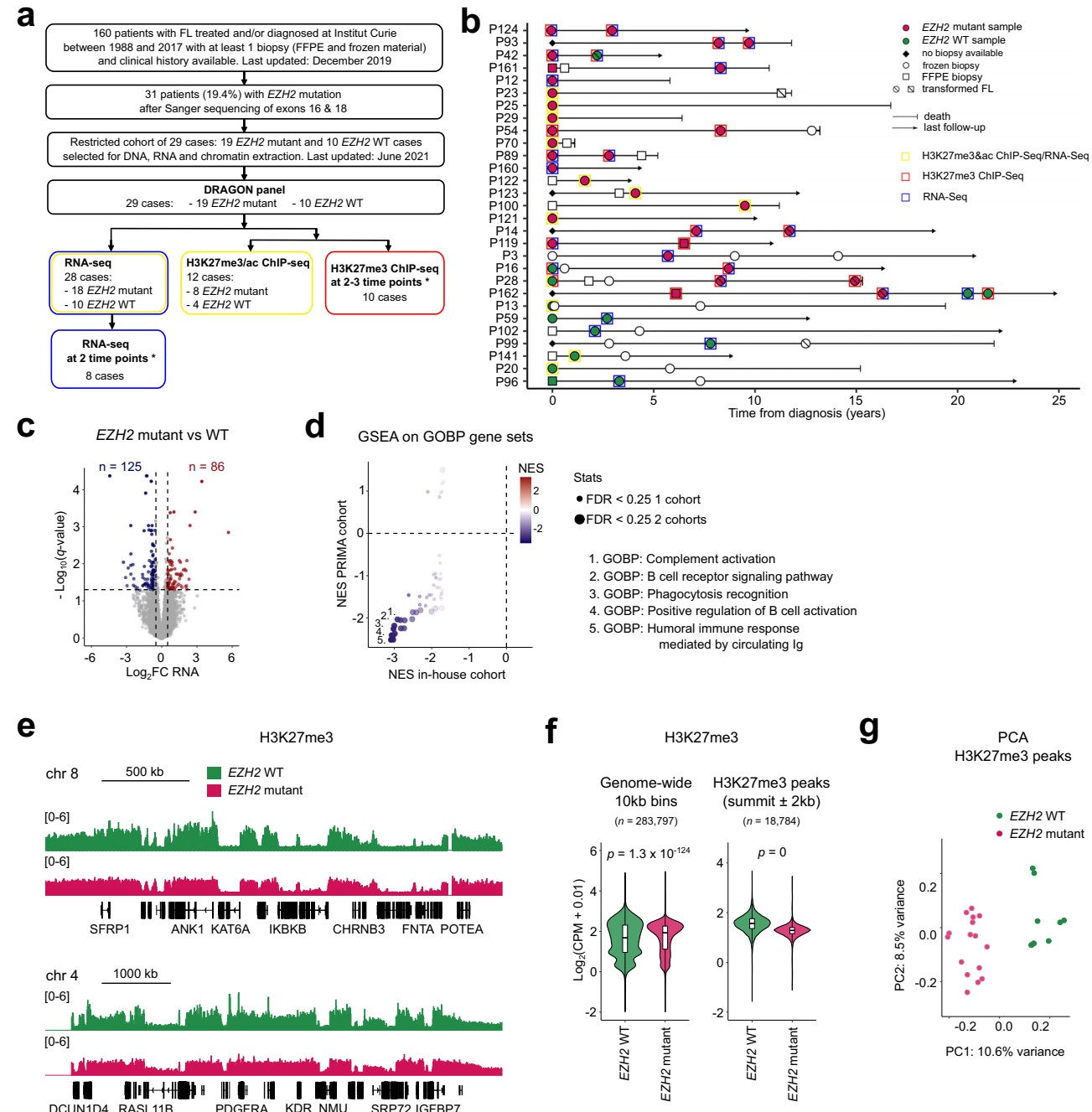

**Fig. 4 | *EZH2*-mutant follicular lymphomas systematically display altered H3K27me3 profiles. a** Workflow of cohort sample selection and molecular analysis. (*For tumors analyzed at multiple time points, DRAGON analysis was performed at all time points.) **b** Disease event timeline indicating timing of biopsy collection and types of analysis conducted. **c** Volcano plot comparing gene expression between *EZH2*-WT and *EZH2*-mutant lymphoma samples and highlighting differentially expressed genes in red (up-regulated) and blue (down-regulated). FC, fold change. **d** Gene ontology biological process (GOBP) gene sets enriched (FDR < 0.25) among genes identified as differentially expressed between *EZH2*-WT and -mutant lymphomas were further analyzed by gene set enrichment analysis (GSEA) to compare their normalized enrichment scores (NES) among genes differentially expressed in both our cohort and the Primary RItuximab and MAintenance (PRIMA) cohort[58].

**e** H3K27me3 ChIP-seq tracks for merged *EZH2*-WT (*n* = 4, each in duplicate) and merged *EZH2*-mutant lymphoma samples (*n* = 8, each in duplicate) with corresponding genomic annotations shown below. **f** Violin and box plots showing mean normalized H3K27me3 ChIP-seq read counts for *EZH2*-WT and -mutant lymphoma samples within all bins of 10 kb across the genome (*n* = 283,797) (left) or within regions identified as enriched in H3K27me3 (*n* = 18,784) (right). Boxes represent median and first and third quartiles, while whiskers show minimum and maximum. *p*-values represent the result of unpaired two-sided *t*-tests. **g** Principal component analysis (PCA) of variance among H3K27me3 ChIP-seq experiments performed in duplicate on *EZH2*-WT and *EZH2*-mutant lymphoma samples, considering only the 5000 most variable H3K27me3-enriched regions.

order to maximize the chances of uncovering clear trends. Upon inspection of merged genomic tracks it is apparent that the H3K27me3 mark adopts a flatter profile in *EZH2*-mutant tumors than in *EZH2*-WT FL (Fig. 4e), reminiscent of the pattern observed in iMEFs expressing

*Ezh2*^Y641F (Fig. 2d). Indeed, systematic counting at the genome-wide level shows that H3K27me3 rises in *EZH2*-mutant cases compared to *EZH2*-WT cases in the majority of bins, with the important exception of H3K27me3-enriched regions, where the mark is clearly reduced (Fig. 4f,

Supplementary Fig. 4d). Remarkably, these global trends are visible in all but one of 24 individual ChIP-seq replicates (Supplementary Fig. 4e). Moreover, a PCA on H3K27me3-enriched regions perfectly segregates all samples by *EZH2* mutation status (Fig. 4g). We conclude that *EZH2* gain-of-function mutations are consistently associated with a distorted H3K27me3 landscape in FL, independently of the other genomic anomalies, including mutations in chromatin-modifying genes, that are frequently found in this highly heterogeneous disease.

Next, we asked whether the differences in gene expression between *EZH2*-WT and *EZH2*-mutant tumors that we had observed in our transcriptomic analysis might be linked to alterations in the distribution of H3K27me3. To address this question we plotted changes in RNA-seq counts against changes in H3K27me3 enrichment for individual genes, but this did not uncover a compelling relationship between the two (Supplementary Fig. 4f). It is worth bearing in mind that increased H3K27me3 at down-regulated genes and reduced H3K27me3 at up-regulated genes also often represent downstream consequences of the respective changes in expression. We therefore propose that, as in our isogenic cultured iMEF system (Fig. 3b), gain-of-function *EZH2* variants in FL and the aberrant H3K27me3 patterns they create likely exert much of their impact on global gene expression through complex secondary effects on the gene-regulatory network.

Finally, we sought to determine whether the genome-wide increase in H3K27ac we had observed in iMEFs expressing *Ezh2*[Y641F] (Fig. 2f–h) was recapitulated in *EZH2*-mutant FL. To this end we performed H3K27ac ChIP-seq on the same set of samples as that used for H3K27me3 ChIP-seq, but found no appreciable difference in the average distribution of the mark between *EZH2*-WT and *EZH2*-mutant lymphomas, whether assessed over the whole genome or specifically at H3K27ac-enriched regions (Supplementary Fig. 4g). We conclude that in contrast to the flattened H3K27me3 pattern, which appears to be a singular feature of *EZH2* gain-of-function variants, effects on H3K27ac resulting from mutant EZH2 activity depend on the context. We speculate that both *EZH2*-WT and *EZH2*-mutant FL undergo substantial changes in their histone modification balance in the course of pathogenesis, notably as a consequence of *CREBBP* and *KMT2D* mutations, that obscure the specific influence of mutant EZH2 on H3K27ac that we were able to isolate in iMEFs.

### H3K27me3 profile tracks with *EZH2* mutation status in patients with heterogeneous FL

Given the long-term nature of most cases of FL, we next wondered whether H3K27me3 profiles undergo recurrent evolution over time in either *EZH2*-WT or *EZH2*-mutant forms of the disease. To explore this possibility we performed H3K27me3 ChIP-seq on multiple FL biopsies collected from each of ten patients at distinct time points separated by several years (Fig. 4b). From a cursory look at the data it was apparent that the H3K27me3 pattern is very heterogeneous even when considering different time points from the same patient (Fig. 5a, b). This prompted us to characterize the mutational status of these tumors at each time point using the DRAGON targeted sequencing approach (Fig. 4b, Supplementary Data 2). We found that in several cases the *EZH2* mutation status varied over time, in either direction, within individual patients (Figs. 4 and 5a, b). When we examined H3K27me3 ChIP-seq tracks corresponding to pairs of samples with divergent *EZH2* status collected from the same patient, we tended to observe a correlation between mutation of *EZH2* and a flattened distribution of the histone mark (Fig. 5a). However, the average H3K27me3 density plots over enriched regions at each time point revealed a more tenuous correspondence between *EZH2* status and H3K27me3 distribution pattern (Fig. 5b). For instance, the density plot for patient 16 remains unchanged between time points 1 and 2 even though its *EZH2* mutational status evolves.

To evaluate this question more systematically we mapped the H3K27me3 ChIP-seq replicates for each patient's different time points onto the PCA shown in Fig. 4g to assess their position with respect to known *EZH2*-WT and *EZH2*-mutant samples (Fig. 5c, Supplementary Fig. 5a). While the longitudinally collected samples tend to map within or near the regions corresponding to their *EZH2* genotypes (e.g., patient 42), this is not a strict rule as exemplified again by patient 16. In contrast to the analysis reported in Fig. 4, here we did not filter out the *EZH2*-mutant patients with a low VAF; we therefore wondered whether this could explain the apparent discrepancies in the PCA. Indeed, the more prominent exceptions in which *EZH2*-mutant samples clustered in the *EZH2*-WT zone were characterized by a low VAF (e.g., patients 16 and 54; Fig. 5c, Supplementary Fig. 5a, Supplementary Table 6). We conducted a similar comparison of RNA-seq profiles for each patient's time points to the known *EZH2*-WT and *EZH2*-mutant samples shown in the PCA in Supplementary Fig. 4c (left), but the link we observed in this case was more modest (Fig. 5d, Supplementary Fig. 5b), pointing toward the potential involvement of other genetic events.

Notably, the DRAGON analysis of longitudinal biopsies also revealed frequent evolution in the mutational status of other genes, including *CREBBP* and *KMT2D*, which altogether suggests that the apparent heterogeneity in the chromatin landscape at different time points actually reflects the coexistence of distinct clones within the same patient (Fig. 5e, Supplementary Data 2). This possibility is supported by earlier reports of subclonal *EZH2* mutations in FL[54,56,59,60] and by a recent study using single-cell RNA-seq coupled with B-cell receptor sequencing that revealed site-to-site heterogeneity with independent evolution in certain FL patients[61]. Incidentally, it is worth mentioning that the FL of 3 out of the 4 patients whose *EZH2* mutational status changed over time underwent transformation (Fig. 4b), an observation that warrants further investigation.

We conclude from this analysis that the chromatin landscape of FL is very heterogenous, including within the same patient, but that rather than reflecting a progressive evolution of the transformed cells this is instead likely due to the coexistence of distinct clones with different genetic backgrounds.

## Discussion

A variety of different mutations affecting PRC2 function are found in cancer, and each of these alterations exhibits distinct tumor-type specificity[5,7]. Our understanding of their molecular and transcriptomic consequences is complicated by the influence of the cell-type-specific contexts in which they arise. The approach presented here, using isogenic cell lines to model and compare mutations with known impacts on PRC2 function, has allowed us to pinpoint the properties that are intrinsic to each. We find in particular that *H3.3K27M*, an alteration found in the vast majority of diffuse midline gliomas, acts to impair PRC2 activity throughout the genome, causing losses of H3K27me3 at essentially every target region and transcriptomic changes resembling milder forms of those observed in isogenic cells lacking the core PRC2 subunits EZH2 and EED. This is in contrast to previous reports that substantial sets of genes retain or even gain H3K27me3 in diffuse midline glioma and other *H3.3K27M*-expressing neural cells, which has been proposed to consolidate their repression[11,15–18]. An alternative interpretation is that in these cell types, the mutant histone indirectly leads a number of genes to become down-regulated, which in turn results in the appearance or maintenance of H3K27me3. Whatever their basis, our approach identifies these H3K27me3 retention and increase events as a specificity of the neural lineage and separates this context-specific impact of *H3.3K27M* from its intrinsic PRC2-inhibitory effect. Understanding how a mutation that fundamentally impairs PRC2 leads to a more nuanced outcome in a cell-type-dependent manner represents an important ongoing challenge in the study of this malignancy[62,63].

Our isogenic system also proved a powerful model in which to unravel the consequences of *EZH2* gain-of-function substitutions at Y646 (Y641 in mouse) frequently found in FL. We made the surprising

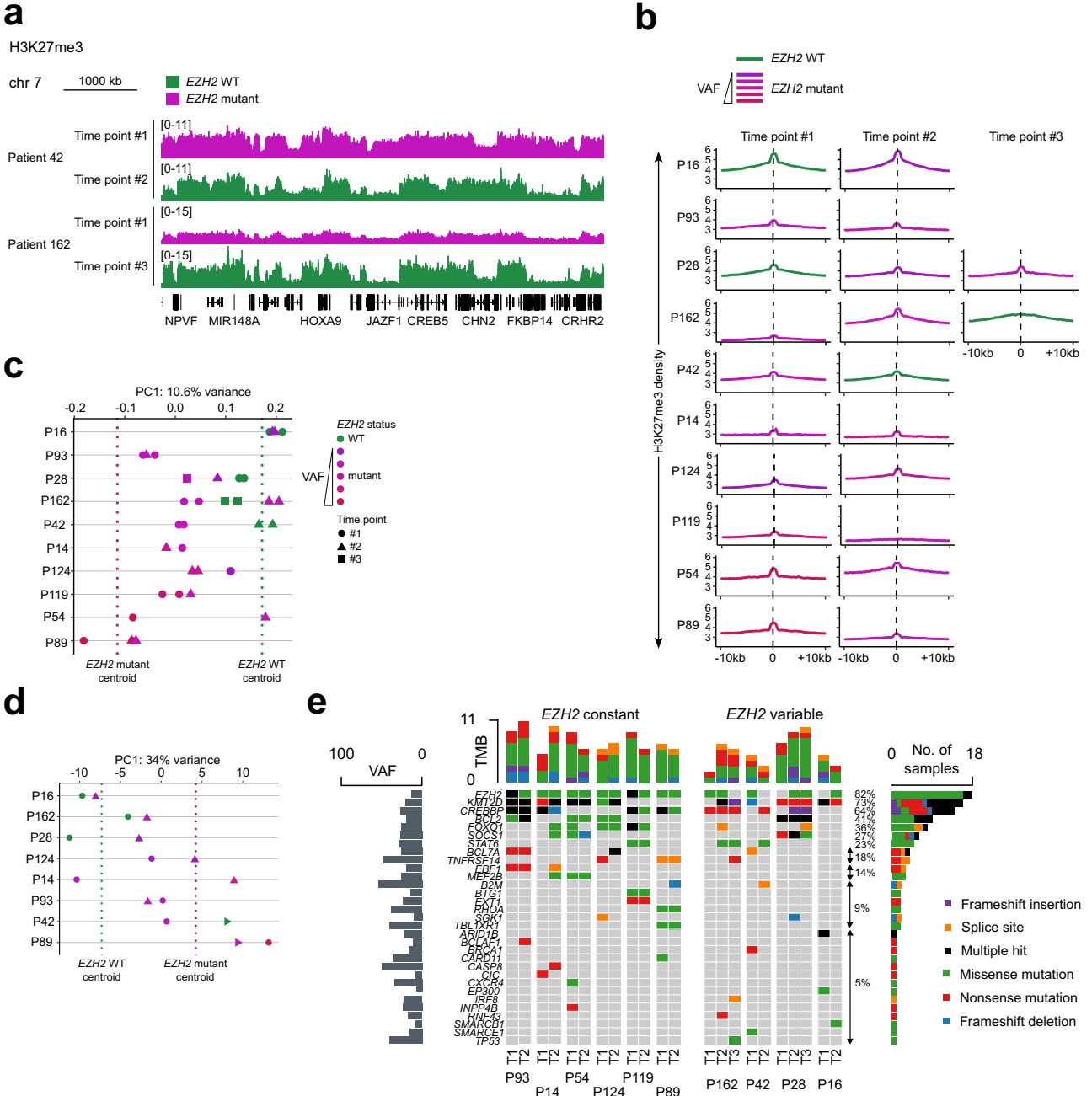

**Fig. 5 | *EZH2* mutation status can change over time in follicular lymphoma with concomitant changes in H3K27me3 distribution. a** H3K27me3 ChIP-seq tracks for follicular lymphoma (FL) samples from individual patients at distinct time points (see Fig. 4b). **b** Mean density plots of H3K27me3 ChIP-seq reads for FL samples from individual patients at distinct time points, across H3K27me3 peaks identified in the analysis presented in Fig. 4. *EZH2*-mutant data are color-coded according to VAF as indicated. Note that patient 28, time point #1 is also part of the analysis presented in Fig. 4. VAF, variant allele frequency. **c** Position of H3K27me3 ChIP-seq samples from 10 individual patients at distinct time points along principal component 1 (PC1) when mapped onto a principal component analysis (PCA) of variance among the full set of H3K27me3 ChIP-seq performed on FL samples, considering only the 5000 most variable H3K27me3-enriched regions, as shown in Fig. 4g. Positions of centroids of

*EZH2*-WT and -mutant subsets along PC1 are indicated. (Full PCA plot shown in Supplementary Fig. 5a.) **d** Position of RNA-seq samples from 8 individual patients at distinct time points (see Fig. 4b) along principal component 1 (PC1) when mapped onto a PCA of variance among the full set of RNA-seq experiments performed on FL samples, considering only the 211 differentially expressed genes displayed in Fig. 4c. Positions of centroids of *EZH2*-WT and -mutant subsets along PC1 are indicated. (Full PCA plot shown in Supplementary Fig. 5b.) **e** Frequency and type of mutations affecting 31 genes among 10 patients from whom multiple samples were analyzed by ChIP-seq. Rows represent genes and columns represent patient samples. For each sample, bars above show the number of genes containing a mutation. For each gene, bars to the right show the number of samples carrying a mutation, and bars to the left show the mean VAF among samples with mutations.

discovery that mutant *Ezh2*, long thought to require cooperation with a WT counterpart to induce changes in H3K27me3 patterns, acts autonomously to drive higher overall levels and a highly aberrant distribution of the histone mark. While we confirmed previous reports

that the in vitro activity of PRC2-EZH2$^{Y641F}$ on unmethylated H3 is minimal[22,23,46], our collection of isogenic cell lines allowed us to demonstrate that the mutant enzyme is capable, alone, of causing excessive and altered accumulation of H3K27me3 in cells. Our in vitro

results suggest that this ability may result from the action of PRC2 accessory subunits. It bears noting that conditional mouse models expressing *Ezh2^{Y641F}* in B lymphocytes have produced divergent conclusions regarding the necessity of a cooperating *Ezh2^{WT}* allele for tumorigenesis or even GC formation[28,64]. While our data indicate no inherent requirement for *Ezh2^{WT}* in enabling *Ezh2^{Y641F}* hyperactivity, they do not rule out a critical contribution of *Ezh2^{WT}* in certain specific situations and strongly support such a role in the case of the Y641N substitution.

The isogenic system also allowed us to draw a precise connection between the increased activity of the EZH2^{Y641F} enzyme and the abnormal H3K27me3 landscape in *Ezh2^{Y641F}*-expressing cells. Indeed, cells expressing this variant alone recovered a normal average H3K27me3 peak profile when treated with moderate doses of a PRC2 inhibitor, suggesting a clear relationship between total PRC2 catalytic activity and genomic distribution of the modified histone. The prominent correlation we observed between H3K27me3 occupancy in mutant cells and H3K27me2 in WT cells is fully consistent with the erosion of H3K27me3 peaks and indicates that the elevated PRC2 activity is to a large degree diverted toward regions usually marked with H3K27me2. Our SUZ12 profiling analysis reveals that PRC2 occupancy is reduced at its normal target sites when *Ezh2^{Y641F}* is present, which further helps explain the paradox of lower H3K27me3 peaks despite increased PRC2 activity. Intriguingly, increased total H3K27me3 accompanied by increased spreading of the mark at the expense of peak enrichment, as occurs upon *Ezh2^{Y641F}* expression, has also been observed in mouse embryonic stem cells (mESCs) maintained in the ground state of pluripotency with MEK1 and GSK3 inhibitors as well as in mESCs or MEFs with impaired DNA methylation[65-68]. This apparent link between H3K27me3 abundance and H3K27me3 distribution, further supported by our PRC2 inhibition experiment, therefore seems to apply to a range of situations. Understanding the mechanistic underpinnings of this phenomenon will require further study.

While PRC2 inhibition corrected the average H3K27me3 profile in iMEFs expressing *Ezh2^{Y641F}*, it unequivocally failed to restore a WT transcriptome. Thus, the action of the mutant enzyme leads to alterations in transcription states that are irreversible regardless of whether overall PRC2 activity is rescued. We recently showed that transient disruption of PRC2 in differentiated somatic cells causes many PRC2 target genes to switch persistently to an active state[32]. Such *cis*-acting transcriptional memory may also be at work here, preventing the transcriptional changes that occur at PRC2 targets in response to EZH2^{Y641F} activity from being reversed when that elevated activity is offset by PRC2 inhibition. We also find that PRC2 inhibition leads to radically different transcriptomic changes in WT and *Ezh2^{Y641F/WT}* cells, suggesting that the environment of *trans*-acting regulators of transcription is also substantially altered. This highly divergent response to PRC2 inhibition, to the degree that it reflects an intrinsic effect of *Ezh2^{Y641F}*, could partly underpin the increased efficacy of clinically approved EZH2 inhibitors on FL patients bearing *EZH2* mutations[30]. Most remarkably, the presence of *Ezh2^{Y641F}* made our iMEF model specifically poised to up-regulate antigen presentation genes in response to PRC2 inhibition. While this conforms with similar observations made in DLBCL of GC origin[52], our results in a non-immune cell type speak to the strength of the link between mutant *EZH2* and the ability of MHC genes to be induced via PRC2 inhibition. This is an especially important consideration given the critical role that the loss of B-cell interactions with T follicular helper cells plays in FL[26].

To understand the extent to which the insights we derived from isogenic cultured cells apply to the FL disease setting, we undertook a first-of-its-kind ChIP-seq analysis of patient biopsies that we assembled to form a longitudinal cohort. Strikingly, our H3K27me3 ChIP-seq results show an altered landscape for this histone modification in FL that tracks precisely with the presence of *EZH2^{Y646}* mutations (when the

VAF is substantial) and that is highly reminiscent of the profile observed in *Ezh2^{Y641F/WT}* iMEFs. We therefore conclude that this aberrant H3K27me3 landscape, which our cell culture experiments indicate the *EZH2* mutation is sufficient to produce, is robust to the various additional mutations that recurrently arise in FL, notably those affecting other chromatin regulators. The intersection of H3K27me3 changes with transcriptomic changes as a function of *EZH2* status in our FL samples suggests that the impact of this altered landscape on gene expression is complex and involves many indirect effects. This hypothesis is also strongly supported by our mechanistic experiments in the isogenic iMEF model. Further work is needed to unravel more precisely the sequence of events by which mutant *EZH2* leads to transcriptional reprogramming.

In contrast to the closely matching H3K27me3 patterns, our H3K27ac ChIP-seq on *EZH2*-WT and *EZH2*-mutant FL samples did not recapitulate the H3K27ac changes we observed in *Ezh2^{Y641F/WT}* iMEFs. In the former case, *Ezh2^{Y641F}* led to a widespread increase in the histone mark that was especially pronounced at H3K27ac peaks, whereas the FL samples did not exhibit a discernible difference according to *EZH2* mutation status. We speculate that mutant *EZH2* does indeed independently promote elevated H3K27ac as suggested by our isogenic system, but that additional factors in both *EZH2*-WT and *EZH2*-mutant FL, including (but not limited to) mutations in other chromatin regulators, may mask this effect when comparing H3K27ac patterns in tumors.

Another insight that emerged from our longitudinal collection was the shifting *EZH2* mutation status, in both directions, of individual patients sampled at different times in the course of their illness, and the consistent corresponding change in H3K27me3 distribution that further confirmed the singular relationship between *EZH2* mutations and aberrant H3K27me3 landscape. This link is further supported by the correlation between the VAF for mutant *EZH2* and the degree of H3K27me3 distortion. Several other studies have also reported transcriptomic and genetic heterogeneity of FL tumors within the same patient, including with respect to *EZH2* status[54,59,61,69], while some studies have specifically examined paired FL and progressive or transformed FL[70,71]. The latter reports agree on a divergent evolutionary model in which transformation results from the expansion of a clone very rarely detected at earlier phases of the disease, with *EZH2* mutation occurring either as an ancestral or a clonal event. In keeping with this hypothesis, we find, albeit based on a limited number of cases, that changes in *EZH2* mutation status frequently coincide with FL transformation.

Given the promise that PRC2 inhibitors have shown against *EZH2*-mutant FL in the clinic[30], the inter-tumoral heterogeneity that we corroborate and that we link, for the first time, to a specific H3K27me3 signature suggests that a better assessment of *EZH2* status might help better predict the clinical benefit of this therapy throughout disease evolution. Indeed, we speculate that the abundance of *EZH2*-WT FL subclones may be predictive of a shorter time to relapse after PRC2 inhibitor treatment. It will therefore be very informative to evaluate not only the presence of *EZH2* mutations but also their VAF, to search for *EZH2* mutations in liquid biopsies (circulating DNA)[72] or to analyze *EZH2* mutations from more than one biopsy. Furthermore, *EZH2* mutation status has recently been suggested to potentially predict response to treatments other than PRC2 inhibitors[73]. Future work will be necessary to investigate these avenues and delineate strategies to improve disease outcomes.

## Methods

### Regulatory approval

For portions of the study involving clinical data, approval was first provided by the institutional review board and local ethical committee (Groupe Thématique de Travail−Hematology section, Institut Curie), under project ID BS#2014-311, when written informed patient consent

was available, or else by the Comité de Protection des Personnes Sud Méditerrannée I, under project ID#RCB2019-A000620-57, when written consent was not available due to death or lost to follow-up, as also described under heading "Patients," subheading "Clinical data," below.

## Cell culture assays

**Cell lines.** Mouse embryonic fibroblasts (MEFs) were isolated from a 13.5-day-old *Ezh2^flox/flox^*; 4-hydroxytamoxifen-inducible *Rosa26::CreERT2* embryo and immortalized by transduction of pMXs-hc-MYC (Addgene 17220) followed by limiting dilution and clone derivation (see "iMEF B" in ref. [32]). Cells were grown in Dulbecco's Modified Eagle Medium (DMEM) high-glucose (4.5 g/L) supplemented with 10% fetal calf serum (FCS), nonessential amino acids and 2 mM L-glutamine at 37 °C with 5% $CO_2$ and 98% humidity. OCI-Ly19 cells transduced with an empty vector (control) or with a vector expressing *EZH2^Y646F^* were a kind gift from N. Katanayeva and E. Oricchio. They were grown in RPMI medium supplemented with 10 % FCS at 37 °C with 5% $CO_2$ and 98% humidity.

**CRISPR/Cas9-mediated gene editing.** The sequences of the oligonucleotides used to clone the single-guide-RNA expression constructs for gene editing are listed in Supplementary Data 3.

**Introduction of heterozygous *Ezh2^Y641F^* and *H3.3K27M* mutations.** CRISPR/Cas9-mediated cut of target DNA was used along with a repair template containing the *Ezh2^Y641F^* mutation (Supplementary Fig. 2a) as previously described[74]. In addition to introducing the point mutation, this procedure eliminates the 5′ *LoxP* site, rendering the *Ezh2^Y641F^* allele insensitive to Cre-mediated excision. In brief, the right arm of the repair template partly overlaps with *Ezh2* exon 16, and an adenine-to-thymine substitution it contains results, upon incorporation, in the mutation of tyrosine Y641 of *Ezh2* to phenylalanine (Y641F). Between the two homology arms, a selection cassette flanked by FRT sequences is introduced, consisting of a splice acceptor, T2A, hygromycin resistance gene, and polyA signal, temporarily interrupting the endogenous transcription unit. After selection of resistant cells, excision of the resistance cassette is achieved by expression of Flp recombinase through transient transfection. After excision, the transcription unit is restored with a single FRT sequence remaining in the intron. Sanger sequencing of genomic DNA validated the heterozygous substitution of adenine to thymine in codon 641 resulting in the Y641F mutation, and equal expression of mutant and WT alleles was verified by Sanger sequencing analysis of cDNA. Tetraploidization was observed after Flp-FRT recombination in both independent *Ezh2^Y641F/WT^* clones analyzed for the study; although a diploid clone proved difficult to obtain, transcriptomic analysis confirmed that diploid and tetraploid *Ezh2^Y641F/WT^* cells exhibited similar changes.

The *H3.3K27M* mutation was introduced using the same approach, by targeting the *H3F3A* locus with the K27M substitution encoded in the repair template, verifying its presence by Sanger sequencing of genomic DNA and subsequently excising the resistance cassette through transient expression of Flp recombinase.

**Conditional deletion of *Ezh2* wild-type allele.** Deletion of the SET catalytic domain of *Ezh2* on the WT allele was induced by supplementing the culture medium with 1 µM 4-hydroxytamoxifen (4-OHT) for 7 days. Addition of an equivalent volume of ethanol (EtOH) to the medium was used as a control. Cells were collected at the end of the 7 days and the deletion was validated by PCR.

**Generation of KO cell lines.** Generation of *Eed-* and *Ezh1*-KO MEFs was performed using CRISPR/Cas9. In brief, a stop cassette containing an antibiotic resistance gene followed by a polyadenylation sequence was inserted into early exons of target genes by homologous recombination. After antibiotic selection, clones were genotyped and complete KO was validated by Western blot.

**Chemical inhibition of PRC2 activity.** iMEF cells were treated with 0.125 µM, 0.25 µM, or 2 µM of UNC1999 or mock control (UNC2400) for 10 days. OCI-Ly19 cells were treated with mock (DMSO), 2 µM of UNC1999 or 3 µM of EPZ6438 for 10 days. Culture medium was renewed every 2 or 3 days. Cells were passaged so as to remain constantly below 80% confluence. Cells were harvested for RNA and protein extraction or CUT&RUN immediately at the conclusion of the treatment.

**RNA extraction, RNA-seq analysis.** All cells were grown simultaneously and RNA was extracted 3 days after plating 200,000 cells in 25cm² flasks per condition per clone. Total RNA was isolated using Trizol-Chloroform extraction and isopropanol precipitation. After RNA was extracted, a gel electrophoresis analysis was performed in order to verify the quality of the RNA. cDNA was further obtained using the High Capacity cDNA RT kit (Applied Biosystems). RNA-seq libraries were prepared using the Swift Biosciences RNA library kit according to manufacturer's instructions and sequenced on an Illumina HiSeq2500 (SE50) or an Illumina NovaSeq 6000 (PE100). Libraries were sequenced at Institut Curie Next Generation Sequencing Core Facility.

Adapters and low-quality bases (<Q20) were removed from single- and paired-end reads with TrimGalore (v0.4.0) and Cutadapt (v1.8.2)[75]. Trimmed reads were then mapped to the mouse reference genome mm10 using the STAR (v2.7.0a) aligner[76] and the corresponding gene annotation from GENCODE: release 23/GRCm38.p6. The following parameters were used for mapping of RNA-seq reads against the reference genome: −seedSearchStartLmax 20 −alignEndsType End-ToEnd. Gene counts were generated using FeatureCounts (Subread v1.5.1)[77]. Since iMEF RNA-seq data include SE50 and PE100 datasets, we removed any potential batch effects with the removeBatchEffect function of the BioconductoR limma package (v3.58.1)[78] prior to conducting the differential expression analysis with DESeq2 (v1.42.0)[79].

**Gene Ontology (GO) analysis.** We first identified genes differentially expressed between WT and *Ezh2^Y641F/WT^* iMEFs ($log_2FC \geq 1$ | $log_2FC \leq -1$; adjusted $P < 0.05$). We also defined genes significantly up-regulated in either WT or *Ezh2^Y641F/WT^* iMEFs upon treatment with 2 µM UNC1999. The four lists of deregulated genes were then submitted to http://geneontology.org/ for identification of overrepresented GO terms[80–82]. To build the bubble plots in Fig. 3i and Supplementary Fig. 3g, h, we retrieved the enrichment and significance values for all GO terms and discarded those with a fold enrichment >100 or <0.01, corresponding to GO terms encompassing a very small number of genes. We then manually annotated the significantly overrepresented GO terms (fold enrichment ≥ 2, FDR < 0.05) into broad categories (e.g., cell communication, cell migration, and adhesion, or metabolism). Finally, we computed a median $z$-score for the significantly deregulated genes in each significant GO term. Bubble plots were generated with the ggplot2 R library (v3.5.0)[83].

**Preparation of nuclear extracts and immunoblotting.** From 25cm² flasks used to grow cells for RNA-seq (see above), once 70–80% confluency was obtained, $10^6$ cells were plated in 75 cm² flasks. Nuclei were extracted 5 days later at 70–80 % confluency. Cells were incubated in buffer A (10 mM HEPES-NaOH pH 7.9, 2.5 mM $MgCl_2$, 0.25 M sucrose, 0.1% NP40, 0.5 mM DTT, 10 µg/mL aprotinin, 1 µg/mL leupeptin, 1 µM pepstatin A, 1 mM PMSF) for 10 min on ice. After centrifugation at 8000 g, the nuclear pellet was then resuspended in buffer B (25 mM HEPES-NaOH pH 7.9, 1.5 mM $MgCl_2$, 700 mM NaCl, 0.5 mM DTT, 10 µg/mL aprotinin, 1 µg/mL leupeptin, 1 µM pepstatin A, 1 mM PMSF, 0.1 mM EDTA, and 20% glycerol) and incubated 10 min on ice. Nuclei were then sonicated using a Branson Sonifier for 45 s at 10% amplitude, then centrifuged at 20,000 g for 15 min at 4 °C. After Bradford quantification, nuclear extracts were acetone-precipitated. All samples were

**Table 1 | List of antibodies used in the study**

| Antibody | Host | Application | Source | Clone/identifier |
|---|---|---|---|---|
| EZH2 | rabbit polyclonal | WB | homemade[33] | N/A |
| H3K27me3 | rabbit monoclonal | WB, C&R,ChIP | CST | C36B11 |
| H3K27me2 | mouse monoclonal | WB, C&R | Active Motif | 324 |
| H3K27me1 | mouse monoclonal | WB | Active Motif | 321 |
| EED | rabbit polyclonal | WB | homemade[33] | N/A |
| H3.3K27M | rabbit monoclonal | WB | Millipore | RM192 |
| H2Aub | rabbit monoclonal | C&R | CST | 8240S |
| H3K27ac | rabbit polyclonal | C&R | Abcam | Ab4729 |
| H3K4me3 | rabbit monoclonal | C&R | CST | C42D8 |
| H3K36me3 | rabbit polyclonal | C&R | Abcam | Ab9050 |
| H4 | rabbit polyclonal | WB | Active Motif | AB_2636967 |
| SUZ12 | rabbit monoclonal | C&R | CST | D39F6 |
| Rabbit IgG | goat polyclonal | WB secondary | BioRad | 12004161 |
| Mouse IgG | goat polyclonal | WB secondary | BioRad | STAR117D800GA |

*WB* Western blot, *C&R* CUT&RUN, *ChIP* chromatin immunoprecipitation, *CST* Cell Signaling Technology.

mixed with loading buffer containing SDS and β-mercaptoethanol and run on homemade 15% or commercial 4-15% gradient acrylamide gels (Bio-Rad). Semi-dry transfer was performed on a Trans-Blot Turbo transfer system (Bio-Rad). Correct transfer was verified by Ponceau staining. The following primary antibodies (Table 1) were used: anti-EZH2 (homemade; 1/3000), anti-H3K27me3 (Cell Signaling Technology; 1/3000), anti-H3K27me2 (Active Motif; 1/5000), anti-H3K27me1 (Active Motif; 1/3000), anti-EED (homemade; 1/2000), anti-HK27M (Millipore; 1/3000), Anti-H4 (Active Motif; 1/3000). Starbright Blue 700 and Dylight 800 fluorescent secondary antibodies (BioRad; 1/5000) were used subsequently. Anti-H4 served to verify that equal protein levels were used for each sample. Homemade antibodies are described elsewhere[33]. Imaging of Western blots was carried out using the ChemiDoc System (Biorad).

**CUT&RUN.** CUT&RUN was performed as previously described[84] with minor modifications. In brief, 1 million cells were pelleted at 600 *g* for 3 min at RT. After washing twice with 1 mL of wash buffer (20 mM HEPES pH 7.5, 150 mM NaCl, 0.5 mM spermidine (Sigma) and protease inhibitors), cells were resuspended in wash buffer and ready for binding with beads. 10 μL of Concanavalin A beads (Bangs Laboratories) was washed once with 1 mL binding buffer (20 mM HEPES pH 7.9, 10 mM KCl, 1 mM CaCl$_2$ and 1 mM MnCl$_2$) and placed on magnet stand to remove the liquid. 10 μL of binding buffer was used to resuspend the beads; then the slurry was transferred to cells and incubated for 10 min at RT with rotation. After brief spin-down, tubes were placed on magnet to quickly withdraw the liquid. 50 μl of anti-body buffer (wash buffer supplemented with 0.1 % digitonin (Millipore), 2 mM EDTA, and 1:100 dilution of antibody of interest) was pipetted and cells were incubated for 10 min at RT with mild agitation. Antibody solution was then carefully removed and permeabilized cells were washed once with 1 mL dig-wash buffer (0.1 % digitonin in wash buffer). A secondary rabbit anti-mouse antibody (ab6709, Abcam) binding step was carried out according to the same procedure if the host species of the primary antibody was mouse. 50 μL of pA-MNase diluted in dig-wash buffer (final pA-MNase concentration of 700 ng/mL) was incubated with cells for 10 min at RT with agitation. After 2 washes with 1 mL dig-wash buffer, beads were resuspended with 100 μL dig-wash buffer and placed on heat block immersed in wet ice to chill down to 0 °C. 2 μL of 100 mM CaCl$_2$ was added to activate pA-MNase and incubation at 0 °C was carried out for 30 min. 100 μL of 2× stop buffer (340 mM NaCl, 20 mM EDTA, 4 mM EGTA, 0.02 % digitonin, 50 μg/mL RNase A and 200 μg/mL glycogen) was added to quench pA-MNAse, and fragments were released by 10 min incubation at 37 °C

with rotation. After centrifugation at 20,000 *g* for 5 min at 4 °C, DNA fragments were recovered by NucleoSpin (Macherey-Nagel) or phenol-chloroform purification. Libraries were prepared using the Accel-NGS 2S plus DNA library kit (Swift Biosciences) for Illumina barcoded system with 16 cycles for amplification. After post-library size selection, library size distribution and concentration were validated by TapeStation 4200 (Agilent). Libraries were sequenced on an Illumina Novaseq 6000 (PE100) at the Institut Curie Next Generation Sequencing Core Facility.

**CUT&RUN-seq data analysis.** Adapters and low-quality bases (<Q20) were removed from reads with TrimGalore (v0.4.0) and Cutadapt (v1.8.2)[75]. Trimmed reads were mapped to the mouse reference genome mm10 with Bowtie2 (v2.2.5)[85] using default parameters. PCR duplicates were removed with Picard Tools MarkDuplicates (v1.97).

BigWig files were created using DeepTools bamCoverage (v3.0.2) (ref. 86) with the following parameters: −normalizeUsingRPKM −ignoreForNormalization chrX chrY −samFlagInclude 64 −extendReads. Reads overlapping problematic, blacklisted regions (mm10-black-list.v2.bed) (ref. 87) were excluded from the computation of coverage with the option −blackListFileName. In addition to scaling by library size, in order to remove any composition biases, we calculated a normalizing factor using the BioconductoR csaw::normFactors function (v1.36.1)[88,89]. This function counts reads in 10-kb genome-wide non-overlapping bins and uses the trimmed mean of *M*-values method to correct for any systematic fold change in the coverage of bins. The normalization factor was passed on to DeepTools bamCoverage with the parameter −scaleFactor. BigWig files of biological replicates were merged with UCSC tools (v2017.05.03) bigWigMerge and bedGraphToBigWig.

For purposes of comparison, we also computed scaling factors using *Drosophila melanogaster* sequences recovered during H3K27me3 CUT&RUN from S2 cells added in equal numbers to each sample at the outset of the procedure. These factors appear in Supplementary Table 1 and were calculated as previously described[32].

H3K27me3 and H3K27ac peaks were called with MACS2 (v2.0.10) callpeak[90] on deduplicated BAM files with the following parameters: -q 0.05 −broad −broad-cutoff 0.05. The CUT&RUN-seq experiments for H3K27me3 on *Eed*-KO cells were used as control for H3K27me3 peak calling. CUT&RUN-Seq experiments performed with IgG were used as control for H3K27ac peak calling. After discarding peaks overlapping blacklisted regions, peaks lying less than 5 kb apart were merged using BEDtools merge (v2.2.2)[91]. To locate H3K27me2-rich regions, we used csaw[88,89] to identify 10-kb, 5-kb, and 1-kb windows with an enrichment at least 2-fold higher than that of the background. The resulting

collection of enriched bins was concatenated into one consensus set of regions, where bins within 5 kb were merged together. PRC2 target genes were defined as those with a H3K27me3 peak in their promoters (TSS ± 3 kb). Average plots of signal over peaks were generated with Deeptools computeMatrix (v3.0.2)[86] and in-house scripts.

ChromHMM (v1.25)[39] was used to identify chromatin states in WT iMEFs. The genome was analyzed at 200-bp intervals and the tool was used to learn models from the six histone marks profiled by CUT&RUN-seq: H3K27me3, H2Aub, H3K27me2, H3K27ac, H3K4me3, and H3K36me3. A model of 10 states was selected and given functional annotation based on the combination of histone marks and their fold enrichment over genomic features.

**Motif enrichment and protein-protein interaction network analysis.** Motif enrichment analysis was conducted using MEME AME (v5.5.5)[92] on the promoters (TSS ± 3 kb) of genes identified as differentially expressed between WT and *Ezh2*$^{Y641F/WT}$ iMEFs (log$_2$FC ≥ 1 | log$_2$FC ≤ −1; adjusted $P < 0.05$). Results are presented in Fig. 3g for all TFs that were also identified as up-regulated in *Ezh2*$^{Y641F/WT}$ *versus* WT iMEFs (log$_2$FC > 0; adjusted $P < 0.05$). Genes encoding these TFs were classified as PRC2 targets if their promoter (TSS ± 3 kb) overlapped a H3K27me3 peak in WT cells. A protein-protein interaction network was built using STRING (v11.5)[93] and Cytoscape (v3.10.1)[94].

**HMT assays**
HMT assays were performed as described previously[95]. Briefly, the reaction containing 500 ng of PRC2-EZH2-WT or PRC2-EZH2-Y641F, 1 µg of substrates (recombinant nucleosomes), and 0.2 M DTT was incubated in methylation reaction buffer (50 mM Tris-HCl pH 8.5, 2.5 mM MgCl$_2$) in presence of $^3$H-*S*-adenosyl-methionine at 30 °C for 15 min or 30 min. Reactions were stopped by boiling 5 min in SDS sample buffer, run on acrylamide gels, and transferred onto PVDF membranes. When added to the reaction, JARID2 and AEBP2 polypeptides are at 100−500-nM concentrations. Nucleosomes were generated by salt dialysis. H3K27me3 nucleosomes were generated as described by ref. 96.

**Patients**
**Clinical data.** All tissue samples were fully anonymized before processing and sequencing. Study approval was first provided by the institutional review board and local ethical committee (Groupe Thématique de Travail−Hematology section, Institut Curie), under project ID BS#2014-311, when written informed patient consent was available, or else by the Comité de Protection des Personnes Sud Méditerrannée I, under project ID#RCB2019-A000620-57, when written consent was not available due to death or lost to follow-up. RNA-seq raw data from patients included in the PRIMA trial (NCT00140582) were courtesy of Drs. Huet, Tesson and Salles (Hospices Civils de Lyon, Pierre-Bénite; INSERM U1052, Université de Lyon; Carnot Calym, Pierre-Bénite; France).

**Biological samples.** All patients with FL were recruited through Institut Curie (Paris, France) based on frozen sample availability and clinical history at the Biological Resource Center. Patient sex was determined from medical records and due to limited numbers was not considered in study design. In total, 160 patients were diagnosed and/or treated at Institut Curie for grade 1/2/3a FL between 1988 and 2017. Every patient had at least 1 Formalin-Fixed Paraffin-Embedded (FFPE) and fresh frozen sample stored, while 106 had multiple time points at which either FFPE only or both FFPE and fresh frozen samples were available. For 160 patients, DNA was extracted from frozen samples after diagnostic confirmation by tumor cellularity during data collection phase. RNA was extracted from 126 frozen samples. The numbers of samples used to generate each type of data presented in the study are summarized in Fig. 4a.

**DNA extraction & *EZH2* exon 16 and 18 sequencing.** 20 mg of fresh frozen tissue from 160 cases were digested with 600 µL of lysis buffer (50 mM Tris-HCl pH 8, 50 mM EDTA, 10 mM NaCl, 1% SDS), then with 0.25 mg of proteinase K (PK) before incubation at 55 °C overnight. 60 µg of RNase A were added before incubation at 37 °C for 1 h. DNA was isolated by phenol-chloroform extraction using standard procedures. The concentration of DNA was assessed via optic density (Nanodrop spectrophotometer, Thermofisher) and 260/280 and 260/230 ratios were calculated. The quality of the DNA was assessed by 0.8% agarose gel electrophoresis.

PCR primers were designed to amplify exons 16 and 18 and flanking intronic sequences of *EZH2*. Primers were designed using the Primer3 software and NCBI tool Primer-BLAST (https://www.ncbi.nlm.nih.gov/tools/primer-blast/). The program was configured to design primers for PCR products ~500 bp in length (for exon 16: sequence of the forward primer is 5′- TAATGTTCATAGCCATTCTCAGCAG-3′; sequence of the reverse primer is 5′- CACAATCCAGTTACTAAGCATG CAA-3′; for exon 18: sequence of the forward primer is 5′-GCT CTCTTGGCAAAAATACCTATCC-3′; sequence of the reverse primer is 5′-GCTTTTGAGTCAGATAACCATCTTG-3′). PCR of genomic DNA was carried out using 100 ng of genomic DNA, 1 µM each primer, 0.2 mM each dNTP, 1X GoTaq buffer (Promega), 1.25 U GoTaq Enzyme (Promega) and 2 mM MgCl$_2$. Cycling was performed on a Bio-Rad T100 thermal cycler. Following an initial denaturation step of heating to 95 °C for 2 min, were 35 cycles of denaturation at 95 °C for 60 s, annealing at 53 °C for 30 s for exon 16 or at 51 °C for 30 s for exon 18 and extension at 72 °C for 60 s and a final extension step at 72 °C for 5 min. PCR products were evaluated by electrophoresis and then further processed with the PCR clean-up Kit (Macherey-Nagel) according to the manufacturer's instructions. Sanger sequencing of PCR products was carried out by Eurofins Genomics, Germany.

**RNA extraction & RNA-seq analysis.** 126 samples were extracted using the RNeasy mini kit (Qiagen, CA, USA) according to the manufacturer's instructions: In brief, 20 mg of fresh frozen thawed tissue samples were transferred to 500 µL QIAzol-filled stainless steel bead tubes. Samples were homogenized in mixer mill Retsch MM400 for 2 min at 30 Hz and then transferred and briefly centrifuged in a phase-lock gel heavy (PLGH) tube before adding another 200 µL of QIAzol. After 5 min incubation at RT, 140 µl of chloroform were added and the samples were centrifuged for 10 min at 14,000 g, at 4 °C. The RNA was precipitated with 525 µL 100% ethanol at RT and then briefly centrifuged. 500 µL of the samples were transferred to a Rneasy Mini Spin Column, centrifuged, and washed once with adequate buffers. After elution, RNA concentration was assessed via Nanodrop, and 260/280, 260/230 and 28S/18S ratios were calculated. The quality of the RNA was assessed on automated Tapestation 4200 platform (Agilent) according to the manufacturer's instructions. RNA Integrity Numbers (RINs) were used to evaluate the integrity of the RNA samples with >7.0 considered intact and <7.0 considered degraded.

RNA-seq libraries were prepared using the Illumina TruSeq Stranded mRNA kit and sequenced on an Illumina NovaSeq 6000 (PE100) at Institut Curie Next Generation Sequencing Core Facility. Adapters and low-quality bases (<Q20) were removed from paired-end reads with TrimGalore (v0.4.0) and Cutadapt (v1.8.2)[75]. Trimmed reads were then mapped to the human hg19 assembly using the STAR (v2.7.0a) aligner[76] and the corresponding gene annotation from GEN-CODE (release 19/GRCh37.p13). Gene counts were generated using FeatureCounts (Subread v1.5.1)[77] and differential expression analysis was performed with DESeq2[79]. Differential enrichment of GO Biological Process gene sets in *EZH2*-mutant *vs.* -WT FL cases was tested using Gene Set Enrichment Analysis (GSEA, GSEA_MacApp_4.3.3)[97,98].

**ChIP & ChIP-seq analysis.** For fresh frozen FL samples, ~30 mg of tissue were pulverized in liquid nitrogen, cross-linked with 1% formaldehyde

for 8 min, and treated with 250 mM glycine in order to stop the cross-linking reaction for 5 min. After 2 washes in PBS, the pellet was incubated successively in two lysis buffers (LB1: 50 mM HEPES-KOH pH 7.5, 140 mM NaCl, 1 mM EDTA pH 8, 10% glycerol, 0.5% NP-40, 0.25% Triton X-100; LB2: 200 mM NaCl, 1 mM EDTA pH 8, 0.5 mM EGTA pH 8, 10 mM Tris-HCl pH 8) and then suspended in a sonication buffer (1 mM EDTA pH 8, 0.5 mM EGTA pH8, 10 mM Tris-HCl pH8, 100 mM NaCl, 0.1% Na-Deoxycholate, 0.5% N-lauroyl Sarcosine). Each sample was processed using a Covaris S220 sonicator at peak power 175, duty factor 10 cycle/burst 200 for a duration of 4 min.

For FFPE FL samples, chromatin extraction was performed as previously described by ref. 99. In brief, 8 sections of 10-μm thickness per sample were deparaffined in xylene 3 times, then progressively rehydrated in ethanol solutions of decreasing concentration. Samples were then incubated at 40 °C for 1 h before treatment for 5 min with PK. Reaction was stopped with Protease inhibitor cocktail containing 1 mM PMSF, 10 μg/mL aprotinin, 1 μg/mL leupeptin, and 1 μM pepstatin A. Each sample was then sonicated using a Covaris S200 sonicator at peak power 240, duty factor 20, cycle/burst 200 for a duration of 40 min.

The sonicated samples (~5 μg chromatin/sample) were cleared for 2 h using non-coupled magnetic beads (Protein G Dynabeads, Invitrogen), then further mixed with 5 μL antibody (anti-H3K27me3, CST) or 2 μL antibody (anti-H3K27ac, Abcam) and incubated overnight at 4 °C on rotator. Complexes were pulled down using Protein A Dynabeads (ThermoFisher Scientific), washed with RIPA buffer, and de-cross-linked, and genomic DNA was isolated by standard phenol-chloroform extraction.

Library preparation was conducted using the Illumina TruSeq ChIP kit. Sequencing and sequence analysis were performed using methods similar to those applied to the CUT&RUN-seq experiments on iMEFs described above. Reads, however, were mapped to the human reference genome assembly hg19. H3K27ac peaks were identified using the same MACS2 parameters as in iMEFs. Csaw[88,89] was used to map H3K27me3-rich regions by first identifying 10-kb, 5-kb, and 1-kb windows with an enrichment at least 2-fold above background levels and then concatenating the resulting collection of bins. As with iMEFs, H3K27ac and H3K27me3 peaks less than 5 kb apart were merged together.

**DRAGON panel.** The design of the NGS panel called DRAGON (Determination of 571 Relevant Altered Genes in Oncology by NGS) was developed specifically for the molecular analysis of tumors at Institut Curie, Unit of Genetics (Paris, France)[55]. It is composed of 571 genes that are of interest in oncology from the points of view of diagnosis, prognosis and molecular therapy. The nucleotide sequence (variants) as well as the number of copies (deletion and focal amplification) are explored for all of these 571 genes. The panel also includes 86 microsatellite sequences to assess the MSI (microsatellite instability) status. Finally, the coding sequence size of the panel exceeds 1.5 Mb and thus makes it possible to assess the Tumor Mutational Burden (TMB) as reliably as with whole exome sequencing[100].

10 ng of input DNA are processed with the SureSelect XT-HS library preparation kit, which incorporates molecular barcodes (UMIs) to detect variations with very low allelic ratios and effectively eliminate background noise. Total panel size is 2.7 Mb. Sequencing is carried out on an Illumina Novaseq sequencer with an average depth of 2000X and a minimum depth of 300X.

Bioinformatics pipeline for DRAGON panel is as follows: quality controls, identity controls (based on the polymorphism clustering present in the design as well as on 37 tri-allelic nucleotide polymorphisms to detect inter-sample contaminations, variant calling using Varscan2 (v2.4.3), calculation of TMB based on the number of non-synonymous coding variants per Mb. Using the backbone included in the design, the copy number profile for each tumor is estimated using Facets (v0.5.1) with the median coverage as control. The ploidy, estimated cellularity, and LOH can also be evaluated by this method.

**Somatic mutation calling and annotation.** All of the variants were filtered through the COSMIC Cancer Gene Census (CGC) v86 (cancer.sanger.uk). Mutations affecting these putative driver genes were annotated as driver mutations if they passed the following filters:

- The mutation must be detected with an allelic frequency >5% in the sample, with a coverage >100x, GnomeAD all must be <0.5%

- We selected frameshift indels, as well as stop-gain mutation types. Intronic mutations were removed.

Genes identified as recurrently mutated in two large studies of FL were systematically screened[54,101].

### Statistics and reproducibility
For all Western blot analyses (Figs. 1a, 2a, b, Supplementary Fig. 1c, 2f, 3d), at least three independent biological replicates were assessed. The in vitro HMT assays were performed at least twice using independent reconstitutions.

### Reporting summary
Further information on research design is available in the Nature Portfolio Reporting Summary linked to this article.

## Data availability
The RNA-seq, CUT&RUN-seq, and ChIP-seq data generated for this study have been deposited in the Gene Expression Omnibus under accession code GSE218717. The following databases and datasets were used for data analysis in this study: Mouse reference genome mm10, Mouse genome annotation GENCODE: release 23/GRCm38.p6, GSM2475229, http://geneontology.org/, *Drosophila melanogaster* reference genome dm6, PRIMA trial (NCT00140582) RNA-seq raw data (courtesy of Drs. Huet, Tesson, and Salles (Hospices Civils de Lyon, Pierre-Bénite; INSERM U1052, Université de Lyon; Carnot Calym, Pierre-Bénite; France), human reference genome hg19, COSMIC Cancer Gene Census (CGC) v86. Source data are provided with this paper.

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

## Acknowledgements

The authors thank Drs. Huet, Tesson and Salles (Hospices Civils de Lyon, Pierre-Bénite; INSERM U1052, Université de Lyon; Carnot Calym, Pierre-Bénite; France) for kindly providing RNA-seq raw data from patients included in the PRIMA trial (NCT00140582), and N. Katanayeva and E. Oricchio (Ecole Polytechnique Fédérale de Lausanne, Switzerland) for kindly providing OCI-Ly19 cells transduced with an empty vector (control) or with a vector expressing $EZH2^{Y646F}$. Work in the lab of R.M. was supported by funding from the ARC Foundation, the FRM (no. EQU202003010312) and by an award from the Fondation de France. P.R. was supported by the MD-PhD program at the Institut Curie. L.R. was a recipient of a postdoc fellowship from the ARC. D.H. was supported by a postdoctoral fellowship from the FRM (no. SPF20150934266). High-throughput sequencing was performed by the NGS platform of the Institut Curie, supported by grants ANR-10-EQPX-03 and ANR-10-INBS-09-08 from the Agence Nationale de le Recherche (investissements d'avenir) and by the Canceropôle Ile-de-France. We thank Corinne Ajri-zov for retrieving the informed consent of the patients.

## Author contributions

P.R., S.A., A.P., M.B., A.M., S.L.C., C. B.-G., C.M., A.L., and D.H. performed the experiments. P.R., M.W., and R.M. conceived the study. P.R. and L.R. did the bioinformatics analysis. J.M.-P. performed DRAGON analysis. A.H. performed the CNV analysis. P.L. oversaw all other next-generation sequencing. F.K. contributed to retrieving clinical data. A.V.-S. contributed to the assembly and access to the patient cohort. P.R., L.R., D.H., and R.M. processed the data and prepared the manuscript with input from all the authors.

## Competing interests

The authors declare no competing interests.
