## [Peer Review File · Nature Communications]

EZH2 mutations in follicular lymphoma distort H3K27me3 profiles and alter transcriptional responses to PRC2 inhibitionREVIEWER COMMENTS

Reviewer #1 (Remarks to the Author):

Romero and colleagues investigated in this study how the gain-of-function mutations/aberrations in EZH2, the catalytic subunit of PRC2, induce aberrant H3K27 methylation patterns, triggering gains in existing H3K27 acetylation peaks and extensive gene expression changes. For this purpose, they used a set of isogenic cell lines (iMEFs) recapitulating relevant mutations in EZH2 found in hematological malignancies, such as follicular lymphoma where EZH2 mutations are frequent (around 25%). Importantly, on a clinical level, using a limited cohort of individual patients' samples at different times in the course of the disease, they hypothesize that changes in EZH2 mutation status frequently coincide with FL transformation.

As the transcriptional expression altered by EZH2 mutations in follicular lymphoma is not completely understood in this incurable disease, the topic is of strong biological and clinical interest. While the manuscript is clearly written, the methods of this work are in line to what are widely used in Omics analysis, although deeper experimental approaches (e.g. single-cell) are necessary to achieve relevant conclusions. Indeed, some concerns are needed to be clarified/addressed:

Main comments:

1. The proposed experimental model using the iMEFs mutants raises some doubts in me. Although iMEF model is an interesting system to test the effect of EZH2 mutations in H3K27 trimethylation and consequently the alteration in gene expression patterns, iMEF system is a totally different cell model from follicular lymphoma, even from a different species. It is widely described that each cell type has its particularities in the packaging and distribution of genes and chromosomal regions. In addition, it is known that during B-cell development, the Germinal Center reaction is a complex multistep process in which epigenetic regulators dynamically suppress or induce transcriptional programs. In particular, in Follicular Lymphoma, epigenetic gene mutations alter the regulation of transcriptional programs, changing Germinal Center B cell function and distorting differentiation towards tumor cells. All these processes are not taken into account in the iMEF model. What are the arguments to defend that your results in EZH2 mutants in iMEF system can be extrapolated to the human Follicular Lymphoma model? In addition, other cofounding alterations, as BCL2 overexpression are not introduced in this system, as done previously using transgenic mice (Béguelin W et al, Cancer Cell 2020).

2. Despite the fact that the models of Follicular Lymphoma cell lines do not fully recapitulate the disease model, it would have been more convenient to use a Follicular Lymphoma cell line to introduce the mutations and test the mutations effect on the transcription.

3. In the figures 1B and Supplemental Figure 1C corresponding to Venn diagrams showing the up-regulated genes induced by the mutations in PRC2 complex using iMEF system, the authors described a "high degree of overlap". However, in the case of genes related to the PRC2 targets, the overlap between the different mutants is remarkable as expected (258 upregulated genes and 47 downregulated genes not included in the overlap). But at a global level (All genes) the overlap decreases and it exist a higher number of genes not included in the overlap (663 upregulated and 430 downregulated) that must to be considered a significant number. It could be interesting to perform an in-depth GEP analysis (GSEA, for example) to analyze exactly which are the differential alterations caused by each mutation in order to find an explanation for the decrease in methylation between EED-KO > EZH2 > H3.3K27M mutants.

4. It has been described that the EZH2 inhibitor used (UNC1999), even at lower doses than those used in this work, can induce in some cancer models cell cycle arrest or cell death through different mechanisms (Apoptosis, Autophagy, Necroptosis...). Could this affect to the results obtained? Have been performed any viability analysis on iMEF cells treated with UNC1999?

5. On the other hand, have you tested additional EZH2 inhibitors in this study in order to verify if the effect is similar to that of UNC1999? In this study perhaps it would have been more pertinent the use of Tazemetostat, that has been approved for patients with relapsed or refractory follicular lymphoma. Also, in recent years, high selective inhibitors for EZH2 (Instead EZH1) or mutated EZH2 have appeared (CPI-169, CPI-1205 or GSK126 among others).

6. Considering EZH2 inhibition with UNC1999 does not restore WT expression patterns in EZH2 mutated cells (Figures 3B-C and Supplemental Figures 3B-C), I suggest to perform an in-depth analysis extending it to downregulated genes. Do the clusters shown in Figure 3E correspond to some known Gene Sets?

7. Data shown of enriched gene sets from Bubble plots (Figure 3G and Supplemental Figure 3B-C) is very scarce. It would be convenient to show a table with all the results to visualize them in detail, instead of duplicate this bubble plot in supplemental figure 3.

8. In the cohort of 29 selected patients, it is specified that some of these patients were treated for breast cancer. Which particular treatment received these patients? Treatment or treatment regimens should be detailed. Furthermore, these results were validated in patients from the PRIMA trial, in which all patients are treated with RCHOP w/wo RCHOP+R maintenance. From the sentence "...in both our cohort and the PRIMA cohort", is inferred that the validation analysis was performed mixing treated and not treated patients. Could gene expression be altered due to the different treatments that these patients have followed?

9. Figure 4C shows a Volcano Plot of the altered genes in the comparison between EZH2mut and WT in Follicular Lymphoma. Are the same genes that the authors observed in the previous comparison between iMEFs with EZH2mut and EZH2 WT (Figure 3A)?

10. In Figure 4B there are some cases in which the EZH2 mutation disappears (P42 and P162). How can you explain this phenomenon? Is it due to tumor heterogeneity? it would be convenient to carry out a single cell expression analysis to better define the alterations caused by the EZH2 mutation and circumvent this problem.

11. Supplemental Table 2 shows the different type of EZH2 mutations included in this study. These mutations are presented in different proportions. Among them, are differences in terms of cause-effect on enzymatic activity or in global molecular alterations? All the mutants have the same behavior as the one with the highest incidence (Y646N)? Have these mutations similar RNAseq, ChIP-seq, etc... profiles?

Minor comments:

1. What have been the criteria to select the chromatin regions of CUT&RUN tracks for the comparison between the mutants throughout the article?

2. It seems that graphs titles of Figure 2D are incorrect according to the associated text in results section and y-axes of the graphs.

3. Figure 3G and Sup 3B are duplicated.

Reviewer #2 (Remarks to the Author):

This manuscript aims to investigate the role of aberrant PRC2 activity in tumorigenesis through the use of genome-wide chromatin immunoprecipitation (ChIP) and RNA sequencing. Specifically, the study intends to elucidate the molecular mechanisms underlying how misregulation of H3K27 methylation promotes cancer development. To achieve this, the study constructs isogenic cell lines that recapitulate each relevant mutation and explores how these mutations contribute to tumor development and progression.

The study found that different mutations lead to a loss of function of the PRC2 complex, with the degree of loss increasing in the order of H3.3K27M, Ezh2-KO, and Eed-KO. These findings are consistent with previous studies. In addition, the study investigated EZH2 gain-of-function mutations in follicular lymphoma and the impact of the Ezh2-Y641F substitution on the cellular balance of H3K27me_{2/3}. The results of this investigation are similar to those observed in previous studies carried out in DLBCL cells. As noted in several previous studies, most recently by Beguelin et al. 2020 and Zimmerman et al. 2022, expression of Y641F leads to increased H3K27me₃ and

the up- or down-regulation of hundreds of genes, including genes involved in antigen presentation. The mutation also alters the transcriptome's response to PRC2 inhibition, resulting in the up-regulation of different target genes than in wild-type cells.

The authors also examined a cohort of follicular lymphoma patients and found that 19.4% had gain-of-function EZH2 mutations. They narrowed their cohort to 29 patients, 19 of whom were EZH2-mutant. The study found that EZH2-mutant FL tumors displayed altered H3K27me3 profiles, with H3K27me3 marks adopting a flatter profile than in EZH2-WT FL. The analysis also revealed heterogeneity in the chromatin landscape at different time points, likely due to the coexistence of distinct clones with different genetic backgrounds within the same patient.

While the study's strengths lie in the collection of follicular lymphoma patient samples, there are several weaknesses in the study design. These include the lack of new mechanistic insight into the histone or EZH2 mutations and the limited understanding of the disease mechanism of EZH2 mutations in follicular lymphoma. Additionally, the experiments depicted in Figure 1 (K27M) seem out of place in the study, which otherwise focuses on the Y641F mutation. Furthermore, these experiments do not effectively recapitulate previous studies and may have been conducted inadequately.

Therefore, while the study provides some insights into the role of aberrant PRC2 activity in tumorigenesis, it falls well short of providing sufficient novel information to justify publication in Nature Communications.

1) The initial title of the results section, "The H3.3K27M oncohistone induces a straightforward PRC2 loss of function," is not an accurate reflection of the complex effects of the K27M histone mutation on gene expression. While some studies have reported global reductions in H3K27me3 associated with K27M, others have shown that this mutation can lead to residual peaks of H3K27me3 at specific genes that remain repressed.

In contrast to the complicated effects of the H3.3K27M mutation on PRC2 function, mutations in EZH2 associated with Weaver Syndrome appear to result in a more straightforward loss of function of PRC2. In a study by Lee et al. published in Molecular Cell in 2018, the authors investigated the effects of mutations in EZH2 on PRC2 function. They found that mutations that disrupted the SET domain of EZH2, which is responsible for its catalytic activity, resulted in a significant reduction in both H3K27me2 and H3K27me3 levels.

The effects of the H3.3K27M mutation on gene expression have been a subject of much research and have yielded some intriguing and paradoxical results. While it is often associated with global reductions in the H3K27me3 histone modification, which is typically associated with gene repression, it is also associated with remaining peaks of H3K27me3 at specific genes that remain repressed. This apparent contradiction has been resolved by recent studies. Specifically, that K27M inhibits the allosterically activated form of PRC2, which may lead to global reductions in H3K27me3, as well as the upregulation of many genes that are normally repressed by PRC2 (Stafford et al 2018, Jain et al. 2020). However, the targeting of PRC2 to CpG islands and H3K27me3 near target sites is unaffected. This proposed mechanism helps to reconcile the seemingly paradoxical results of ChIP-seq experiments on K27M histones and provides a more comprehensive understanding of the effects of this mutation on gene expression.

2) The authors of the study appear to have used a Western blot and ChIP-seq for H3K27me3 in K27M cells to support their conclusion that the mutation induces a straightforward loss of function in PRC2. However, these experiments actually show anything but a straightforward loss of function. For example, Figure 1C of the study demonstrates that residual peaks of H3K27me3 remain in K27M cells, suggesting that the mutation is not universally disrupting PRC2 function. Additionally, Figure 1B shows that only a subset of genes become derepressed or upregulated upon H3.3K27M, further indicating that the effects of the mutation are complex and not necessarily straightforward.

3) Moreover, the Western blot in Figure 1A contradicts published reports of a "clear decrease" in H3K27me2/3 in cells expressing H3.3K27M. This discrepancy goes beyond the nuance of K27M

effects on K27me3 - it suggests a technical/experimental flaw. Where is the K27me3 that apparently exists in K27M cells? It almost certainly isn't outside of K27me3 (WT) peaks but if it is then it needs to be identified/quantified. Something is awry - either the western or the CUT/RUN data.

4) The data presented in Figure 1 of CUT&RUN experiments have been read-normalized. However, it is critical to note that quantitative comparisons made without spike-in normalization are highly questionable. Spike-in normalization is essential to compare the CUT&RUN data with different samples and experiments accurately. Without spike-in normalization, it is difficult to ascertain the actual differences between samples, and any comparisons made may be misleading.

5) The authors of this study appear to suggest that their isogenic system is unique, but in reality, it is not a novel approach. Many other studies have previously manipulated K27M mutations in different isogenic contexts, and this is a well-established method in the field.

6) It is important to note that, despite some incongruencies and suboptimal spike-in calibration in Figures 1A and 1D, the decreased K27me3 signal in 1D is not necessarily inconsistent with other reports. The studies referenced by the authors and others have reported a paradoxical retention of K27me3 to occur at a subset of PRC2 target sites, while other sites show decreased K27me3 signal in K27M cells. Furthermore, some studies have reported similar K27me3 enrichment only at retained sites, while others that display K27me3 enrichment (by heatmap or metaplot) at all target sites in WT cells show the same decreased K27me3 signal as reported in Figure 1D.

7) The authors' conclusion that the H3.3K27M mutation leads to a straightforward loss of function of PRC2, with paradoxical local effects being specific to certain cellular contexts rather than a general property of H3K27M, is not accurate. Their conclusion does not align with the literature on this topic. Furthermore, the authors' own data does not support their conclusion that the H3.3K27M mutation causes a global loss of K27me3. Their K27M+ MEFs display near-equal levels of K27me3 globally compared to WT, as shown in Figure 1A by western blot analysis. This finding suggests that the global loss of K27me3 seen in other systems is context-dependent. As Stafford et al. 2018 demonstrated, the molar ratio of PRC2:K27M in cells determines the degree of PRC2 inhibition and, therefore, the global K27me3 levels. The PRC2:K27M molar ratio in these MEFs may be too high to cause dramatic decreases in K27me3, whereas in human DMGs, the opposite is true.

8) The results presented in Figure 2 are not surprising since H3K27-methylated nucleosomes persisted in Y641F cells even after depletion of EZH1 and wildtype EZH2. This suggests that Y641F might be responsible for maintaining the methylation of these nucleosomes in the absence of the wildtype enzyme. However, to fully understand Y641F's ability to methylate unmodified nucleosomes in a cellular setting, it is crucial to delete both wildtype EZH2 and EZH1, which would effectively start with no H3K27 methylation, and then express Y641F. Only then can we accurately assess the role of Y641F in maintaining H3K27 methylation. The authors cannot make conclusions about Y641F in the context of pre-existing K27me1.

Furthermore, Sneeringer et al. 2010 demonstrated in their Figure 2 and Table 1 that Y641F has some ability to methylate unmodified H3K27, unlike Y641N/H/S mutations. Therefore, it is possible that if the authors had used a Y641N mutation in the current study, they might not have observed the persistence of H3K27me3 in the same way as Y641F. This highlights the importance of studying different Y641 mutations to gain a better understanding of their effects on H3K27 methylation and their role in disease development. However, it is worth noting that Y641 mutations always occur as heterozygotes in the disease, and it is unclear how the present study expands our understanding of Y641F or its role in the development of DLBCL or FL.

9) The authors conducted a comprehensive survey of follicular lymphoma samples containing EZH2 mutations and wildtype EZH2, using ChIP-seq to analyze the H3K27me3 and H3K27ac profiles. The results revealed that EZH2 gain-of-function mutations create a distorted H3K27me3 profile independent of other mutations, which was not unexpected given the known role of EZH2 in H3K27 methylation. The small differences observed in H3K27ac profiles between wildtype and mutant EZH2 ChIP-seq patterns were mostly expected, as the major H3K27ac peaks are found at

gene enhancers related to cell type identity ('super enhancers') and would persist despite gain of EZH2 activity elsewhere in the genome. The small number of peaks that do change in Figure 4I are likely related to the differences in gene expression resulting from the gain of H3K27me3 in EZH2 mutants.

While the longitudinal studies on follicular lymphomas containing the EZH2 mutations showed differences in EZH2 status over time, the authors did not provide clear conclusions regarding these experiments, except to say that there is heterogeneity among different patients. It is possible that there are many patient-specific variables that are unknown, which could affect the outcome of the study. Moreover, it is unclear how this result informs our understanding of EZH2 mutations or follicular lymphoma.

Overall, while the survey conducted by the authors provides important insights into the distorted H3K27me3 profile caused by EZH2 gain-of-function mutations, the results are not entirely surprising. However, the small differences observed in H3K27ac profiles and the longitudinal studies on follicular lymphomas containing EZH2 mutations provide additional information that could be further explored in future studies.

Reviewer #3 (Remarks to the Author):

Romero et al. report on the effects of various EZH2 mutations on H3K27 methylation profiles and the transcriptome, both in embryonic fibroblasts (iMEF cells) and follicular lymphoma (FL) samples. Using isogenic lines of iMEFs, they carefully document the profiles observed as a result of different EZH2 levels of activity, with a particular focus on a mutation that causes EZH2 to be hyperactive (Y641F). Whereas H3K27 methylation profiles can be rescued by inhibiting the hyperactive EZH2 enzyme, the transcriptome effects cannot, a finding that is currently not explained. In the second part of the manuscript, they analyze the effects of hyperactive EZH2 in human patient FL samples with or without the mutation. The global profiles of H3K27 methylation tend to correlate well with EZH2 mutation status.

This is a manuscript with many interesting observations and a careful experimental approach. The comparison between 'clean' genetic changes (in the iMEF cells) and human patient FL samples adds interest, depth and complexity to the study. The overall redistribution of H3K27me3 with hyperactive EZH2 is not a new finding (for example Souroullas et al. Nat Med. 2016, cited by the authors). On a technical level, the authors need to address how quantitative the CUT&RUN analyses are and how this impacts the analysis. Also, the relationships between H3K27 methylation profiles and transcriptomic changes remain unclear.

Specific comments:

1. Regarding the H3K27M oncohistone, the authors report, in contrast to other studies, "In our isogenic model, however, H3K27me3 losses are widespread and individual H3K27me3 peaks almost universally display a substantial reduction in enrichment (Figure 1D-E)." However, Fig. 1D appears to show for one particular interval on chr13, relatively strong retention of H3K27me3 at some peaks and a much stronger reduction elsewhere. In a global quantitative analysis (MA density plot over a set of 17,000 peaks; Fig. 1E), many peaks are reduced by about 5-fold ($2\text{LogFC} \sim -2$). However, the figure shows a cloud with a very broad distribution of fold changes in oncohistone versus WT cells, ranging from unchanged ($\text{Log}_2\text{FC} \sim 0$) to strongly down ($\text{Log}_2\text{FC} \sim -5$). To me this all seems in relatively good agreement with previous conclusions contradicted by the authors. This also relates to the conclusion at the end of the first paragraph of the results "that H3.3K27M represents a straightforward loss-of-function mutation". This appears to be an oversimplification that is not fully supported by the data shown.
2. How quantitative is the CUT&RUN-seq approach, as implemented? There is no spike-in and the data would need to be quantitative in order to compare conditions. The need for quantitation is more or less a general point for all ChIP/CUT&RUN-type analyses, but it is even more important in the analyses of EZH2[Y641F], in which case outside-of-peaks reads may represent real signal rather than background noise. RPKM normalization typically corrects for sequencing depth. This

works generally well in cases where the majority of sequenced reads corresponds to non-specific genomic background (and therefore is independent of different mutations). With hyperactive EZH2 this requirement appears to be violated, because EZH2 deposits methyl groups genome-wide. This concern applies to both the iMEF experiments as well as the FL analyses (Figs. 2D and 4F).

Whereas the genomic redistribution of H3K27me3 in EZH2[Y641F] cells beyond doubt is real, could the observed reduction of signal in WT-peaks be an artefact of normalization?

3. "addition of AEBP2 and JARID2 stimulated both the WT and the mutant enzymes, allowing the latter to exceed the basal activity of the WT (Figure 2B)." The figure seems to suggest the WT enzyme (left side of Fig. 2B) to be more active than the mutant one (right side).

4. In Supplemental Figure 2D, there does not seem to be a global increase in H3K27me3 in WT compared to EZH2[Y641F] cells, in contrast to main Figure 2A. Please explain. What is the fold difference? Is the blot overexposed? Can you quantify?

5. The authors show that pharmacological inhibition of EZH2 in EZH2[Y641F] cells restores a WT-like pattern of H3K27me3. This is an important and interesting experiment that suggests that it is the increased catalytic activity of EZH2[Y641F] that is driving the changes in EZH2[Y641F] cells, rather than some other property of the enzyme that affects its targeting to genomic regions. However, these observations do not explain why canonical H3K27me3 peaks are found to be reduced in EZH2[Y641F] cells (unless they are not really reduced; cf. comment on normalization).

6. Using RNA-sequencing, the authors show there is very little rescue of the EZH2[Y641F]-mediated changes in the transcriptome by EZH2 inhibition (Fig. 3E). "Indirect" or "secondary" as the transcriptional consequences may be, how can this be reconciled with the patterns of H3K27me3 (Fig. 2J) which are interpreted as a rescue? One would expect indirect gene expression effects to be rescued as well. The authors acknowledge this issue, but simply state that the hyperactive EZH2 mutant "reprograms its response to PRC2 inhibition". Clearly, the enhanced catalytic activity of EZH2[Y641F] is not "necessary and sufficient" to cause the transcriptomic effects. What does that mean, and how would this work if not through genomic histone modification patterns?

7. Also, the summarizing conclusion on p. 10 ("We also find major transcriptomic changes that primarily reflect the indirect impact of a shifting chromatin landscape.") is not adequate, as there must be something in addition to indirect effects of altered H3K27me3 patterns to explain the lack of transcriptomic rescue of when inhibiting EZH2 activity. This inadequate reasoning is also found on p. 12 ("... through complex secondary effects on the gene-regulatory network") and in the abstract ("our results provide a mechanistic foundation for understanding how oncogenic PRC2 mutations disrupt chromatin and transcription").

8. The authors conclude on p. 12 that "EZH2 gain-of-function mutations are necessary and sufficient to create a distorted H3K27me3 landscape in FL, independently of the other genomic anomalies", but in fact only show that the gain-of-function mutation can explain a distorted H3K27me3 landscape in FL. The iMEF experiments go further ("necessary and sufficient"), but it would take a rescue experiment to establish the same for FL. I would suggest to simply rephrase this particular conclusion.

We wish to thank the reviewers for their time and valuable comments. We have striven to address most of them by performing additional experiments. We believe that the new data further strengthen our initial conclusions and hope that the reviewers will share this opinion.

Below is our point-by-point response to reviewers' remarks.

Reviewer #1 (Remarks to the Author):

Romero and colleagues investigated in this study how the gain-of-function mutations/aberrations in EZH2, the catalytic subunit of PRC2, induce aberrant H3K27 methylation patterns, triggering gains in existing H3K27 acetylation peaks and extensive gene expression changes. For this purpose, they used a set of isogenic cell lines (iMEFs) recapitulating relevant mutations in EZH2 found in hematological malignancies, such as follicular lymphoma where EZH2 mutations are frequent (around 25%). Importantly, on a clinical level, using a limited cohort of individual patients' samples at different times in the course of the disease, they hypothesize that changes in EZH2 mutation status frequently coincide with FL transformation.

As the transcriptional expression altered by EZH2 mutations in follicular lymphoma is not completely understood in this incurable disease, the topic is of strong biological and clinical interest. While the manuscript is clearly written, the methods of this work are in line to what are widely used in Omics analysis, although deeper experimental approaches (e.g. single-cell) are necessary to achieve relevant conclusions. Indeed, some concerns are needed to be clarified/addressed:

Main comments:

1. The proposed experimental model using the iMEFs mutants raises some doubts in me. Although iMEF model is an interesting system to test the effect of EZH2 mutations in H3K27 trimethylation and consequently the alteration in gene expression patterns, iMEF system is a totally different cell model from follicular lymphoma, even from a different species. It is widely described that each cell type has its particularities in the packaging and distribution of genes and chromosomal regions. In addition, it is known that during B-cell development, the Germinal Center reaction is a complex multistep process in which epigenetic regulators dynamically suppress or induce transcriptional programs. In particular, in Follicular Lymphoma, epigenetic gene mutations alter the regulation of transcriptional programs, changing Germinal Center B cell function and distorting differentiation towards tumor cells. All these processes are not taken into account in the iMEF model. What are the arguments to defend that your results in EZH2 mutants in iMEF system can be extrapolated to the human Follicular Lymphoma model? In addition, other confounding alterations, as BCL2 overexpression are not introduced in this system, as done previously using transgenic mice (Béguelin W et al, Cancer Cell 2020).

We understand the reviewer's concern that the iMEF model is distant from the follicular lymphoma (FL) disease context. Nevertheless, we offer two major arguments to support the use of this system to illuminate the effects of *EZH2* mutations in FL.

First, as the reviewer rightfully highlights, *EZH2* mutations in FL occur in the context of other genetic alterations (including most prominently *BCL2* rearrangements), which complicates the investigation of their precise molecular consequences. Our iMEF model is free of these confounding effects and critically enables us to isolate the intrinsic impact of FL-associated *EZH2* mutations on the chromatin landscape and gene expression.

Second, unlike lymphoma cells, our iMEFs show no difference in proliferation in the presence of *EZH2* inhibitors as a function of *EZH2* mutation status (see response to point 4 below and Supplementary Fig. 3a). Therefore, the iMEF system allows us to cleanly assess the differential impact of *EZH2* inhibitor treatment on transcriptional regulation in wild-type and *EZH2*-mutant cells without secondary changes resulting from loss of viability.

For these reasons, the isogenic iMEF model is in fact uniquely suited to addressing the mechanistic questions that we chose to pursue in this study, with advantages that a more FL-like system alone could not have provided.

All the same, we do agree with the reviewer that a complete understanding of how *EZH2* mutations contribute to lymphomagenesis requires examining their synergy with other FL-associated alterations such as *BCL2*. We therefore also approach the role of *EZH2* mutations from more disease-proximal angles. In addition to the first-of-its-kind chromatin profiling of patient samples that we presented in the original version of our manuscript, we now also analyze the impact of *EZH2* gain-of-function mutations when introduced into the DLBCL cell line OCI-Ly19, which harbors the t(14; 18) *BCL2* rearrangement, on transcription and the response to *EZH2* inhibition (see response to point 2 below and Supplementary Fig. 3 c-e, g).

2. Despite the fact that the models of Follicular Lymphoma cell lines do not fully recapitulate the disease model, it would have been more convenient to use a Follicular Lymphoma cell line to introduce the mutations and test the mutations effect on the transcription.

To complement our isogenic iMEF system (see response to point 1 above) we followed the reviewer's recommendation and performed similar transcriptomic analyses in a lymphoma cell line. Rather than FL, which is difficult to culture, we chose the highly tractable DLBCL line OCI-Ly19. This cell line is derived from an *EZH2*-wild-type tumor, but represents the GCB subtype of DLBCL in which *EZH2* gain-of-function mutations are specifically and frequently observed, as in FL. It is therefore a well-suited model for testing the effects of these *EZH2* mutations. Using this cell line and its isogenic counterpart in which *EZH2*^{Y646F} was introduced by Donaldson-Collier et al., 2019, we examined the transcriptome and its response to the *EZH2* inhibitors UNC1999 and tazemetostat. We found the following:

- PCA of these different conditions (wild-type and mutant, treated and untreated), computed using the 1000 most variable genes, revealed that UNC1999 and tazemetostat treatments produce highly similar changes in gene expression

(Supplementary Fig. 3d in our revised manuscript). This strong similarity suggests that the changes are a direct result of EZH2 inhibition rather than a non-specific effect of the individual compounds.

- The PCA also recapitulated a major conclusion from our analysis of iMEFs (Fig. 3d in our manuscript), which is that inhibition of EZH2 in cells carrying the *EZH2* gain-of-function mutant does not restore a gene expression profile resembling that of *EZH2*-wild-type cells.
- Finally, our transcriptomic analysis of OCI-Ly19 cells reproduced another major finding from our analysis of iMEFs (Fig. 3e), which is that EZH2 inhibition leads to the up-regulation of highly distinct sets of genes in *EZH2*-wild-type versus *EZH2*-mutant cells (Supplementary Fig. 3e in our revised manuscript).

Therefore, we conclude that the main mechanistic insights from our iMEF model regarding *EZH2* gain-of-function mutations and the transcriptomic response to EZH2 inhibitors also hold true in lymphoma cells.

3. In the figures 1B and Supplemental Figure 1C corresponding to Venn diagrams showing the up-regulated genes induced by the mutations in PRC2 complex using iMEF system, the authors described a “high degree of overlap”. However, in the case of genes related to the PRC2 targets, the overlap between the different mutants is remarkable as expected (258 upregulated genes and 47 downregulated genes not included in the overlap). But at a global level (All genes) the overlap decreases and it exist a higher number of genes not included in the overlap (663 upregulated and 430 downregulated) that must to be considered a significant number. It could be interesting to perform an in-depth GEP analysis (GSEA, for example) to analyze exactly which are the differential alterations caused by each mutation in order to find an explanation for the decrease in methylation between EED-KO > EZH2 > H3.3K27M mutants.

We fully agree with the reviewer that the overlap in differentially expressed genes between the *H3.3K27M*, *Ezh2*-KO and *Eed*-KO conditions is high for PRC2 target genes, as we highlight in the manuscript, and more modest when considering all genes. We believe this is consistent with our proposed model that the three mutants represent PRC2 loss-of-function alterations of differing severity. Indeed, the model predicts that genes directly regulated by PRC2, in other words PRC2 targets up-regulated in mutant cells, should exhibit a strong overlap among the three conditions (Fig. 1b). Other transcriptomic changes, such as down-regulation of PRC2 targets and up- and down-regulation of other genes, are expected to reflect more indirect consequences of the alterations in PRC2 activity and thus not to overlap as neatly, as we observe (Supplementary Fig. 1d). Since the mutations’ putative *direct* effects on gene expression differ in magnitude, it is not surprising that their putative *indirect* effects should exhibit a fair degree of divergence, particularly when measured using a binary classification of genes into either differentially expressed or not.

Along these lines, we also agree with the reviewer that it would be valuable to further test our model by conducting a finer comparison of the transcriptomic alterations induced by the

three mutations. To that end we followed his/her suggestion and performed a pre-ranked gene-set enrichment analysis (GSEA) to assess the ranked enrichment of all genes up- or down-regulated in each mutant condition among the sets of all genes up- or down-regulated in the other conditions. As displayed in Fig. 1c and Supplementary Fig. 1f in our revised manuscript, this analysis showed a highly significant enrichment for each of these pairwise comparisons. These data show that the transcriptomic changes caused by each of the three mutations exhibit a very high degree of similarity, even beyond what was evident in the analysis we presented previously. These results strengthen our conclusion that *H3.3K27M*, *Ezh2*-KO and *Eed*-KO trigger comparable alterations in transcriptional regulation that differ primarily in their intensity.

4. It has been described that the EZH2 inhibitor used (UNC1999), even at lower doses than those used in this work, can induce in some cancer models cell cycle arrest or cell death through different mechanisms (Apoptosis, Autophagy, Necroptosis...). Could this affect to the results obtained? Have been performed any viability analysis on iMEF cells treated with UNC1999?

We agree with the reviewer that it is important to consider the possibility that the transcriptomic responses to EZH2 inhibitor treatment might reflect overall loss of cell fitness rather than impairment of PRC2 function per se. To check, we measured cell population doublings of both wild-type and *Ezh2*^{Y641F}-expressing iMEFs in the presence and absence of the highest dose of UNC1999 inhibitor used in the study. Cell proliferation rates for the four resulting conditions were indistinguishable for at least 17 days (Supplementary Fig. 3a in our revised manuscript). We conclude that, in our model, the response to EZH2 inhibitor treatment provides insight into PRC2-mediated regulation without major confounding effects on cell fitness.

5. On the other hand, have you tested additional EZH2 inhibitors in this study in order to verify if the effect is similar to that of UNC1999? In this study perhaps it would have been more pertinent the use of Tazemetostat, that has been approved for patients with relapsed or refractory follicular lymphoma. Also, in recent years, high selective inhibitors for EZH2 (Instead EZH1) or mutated EZH2 have appeared (CPI-169, CPI-1205 or GSK126 among others).

We agree with the reviewer that the conclusions of our experiments would be strengthened if reproduced using an independent EZH2 inhibitor, in particular tazemetostat since it has been approved in the clinic and since one of our objectives is to better understand its differential efficacy against *EZH2*-wild-type and *EZH2*-mutant FL.

To this end, we included in our efforts to address point 2 above a treatment of OCI-Ly19 *EZH2*-wild-type and *EZH2*-mutant cells with tazemetostat as well as UNC1999. As described in our response to point 2 above, transcriptomic analysis revealed that the two compounds

produce highly similar changes in gene expression (Supplementary Fig. 3d in our revised manuscript), suggesting that these effects are a specific result of EZH2 inhibition.

6. Considering EZH2 inhibition with UNC1999 does not restore WT expression patterns in EZH2 mutated cells (Figures 3B-C and Supplemental Figures 3B-C), I suggest to perform an in-depth analysis extending it to downregulated genes. Do the clusters shown in Figure 3E correspond to some known Gene Sets?

To determine whether the individual clusters in Fig. 3e correspond to specific gene sets we conducted GO analysis on each of them, as displayed in Supplementary Fig. 3f of our revised manuscript. This revealed that clusters 1 and 2 (genes upregulated upon EZH2 inhibitor treatment in wild-type cells only) contained mainly developmental GO terms, as expected for classical Polycomb target genes. In contrast, clusters 3 and 4 (genes upregulated upon EZH2 inhibitor treatment in *Ezh2*^{Y641F/WT} cells only) contained fewer, largely unrelated terms, with antigen processing and presentation annotations prominent among them, as already noted in the original version of the manuscript.

Importantly, when we conducted the same experimental treatment and analysis using OCI-Ly19 cells, we again observed largely non-overlapping sets of up-regulated genes in *EZH2*-wild-type and *EZH2*-mutant cells upon EZH2 inhibition (Supplementary Fig. 3e in the revised version of our manuscript). This finding confirms that the transcriptomic reprogramming we uncovered in our iMEF model is a general property of the *EZH2* gain-of-function mutation. We also used our OCI-Ly19 data to follow the reviewer's recommendation and compare the sets of genes that became down-regulated in EZH2-inhibitor-treated cells. Similarly to the up-regulated genes, we found a very substantial difference in which genes were down-regulated upon inhibitor treatment between *EZH2*-wild-type and *EZH2*-mutant cells, as shown in the heatmap just below.

7. Data shown of enriched gene sets from Bubble plots (Figure 3G and Supplemental Figure 3B-C) is very scarce. It would be convenient to show a table with all the results to visualize them in detail, instead of duplicate this bubble plot in supplemental figure 3.

We agree with the reviewer that the presentation of the bubble plot data in the original version of the manuscript was somewhat redundant. To rectify this, we now include a more detailed analysis of the individual clusters from Fig. 3e, displayed in Supplementary Fig. 3f of our revised manuscript, as described in our response to point 6 above. Furthermore, as requested by the reviewer, we now provide a full table of all significantly enriched GO terms among genes that become upregulated when either wild-type or $Ezh2^{Y641F/WT}$ cells are treated with 2 μ M of the UNC1999 EZH2 inhibitor (Supplementary Tables 1 and 2, respectively, in our revised manuscript).

8. In the cohort of 29 selected patients, it is specified that some of these patients were treated for breast cancer. Which particular treatment received these patients? Treatment or treatment regimens should be detailed. Furthermore, these results were validated in patients from the PRIMA trial, in which all patients are treated with RCHOP w/wo RCHOP+R maintenance. From the sentence "...in both our cohort and the PRIMA cohort", is inferred that the validation analysis was performed mixing treated and not treated patients. Could

gene expression be altered due to the different treatments that these patients have followed?

Indeed, out of 19 EZH2 mutant FL cases included in our cohort, three were diagnosed in patients with prior history of breast cancer (BC). Two out of 10 EZH2 WT FL patients were also treated at our institution for BC before the diagnosis of FL. This information as well as the specific type of BC treatment received by these five patients is now included in Supplementary Table 6. Of note, one patient in the EZH2 mutant subgroup and one patient in the EZH2 WT subgroup received hormone therapy, which has been demonstrated to increase the risk of developing Non-Hodgkin Lymphoma (NHL) in the early follow-up of BC; in contrast, there doesn't seem to be association between radio/chemotherapy and NHL occurrence following BC^{1,2}. We believe that our cohort size is too small to make any inference about the risk of developing FL after BC or about the molecular alterations found in FL in such cases. We amended the text and added the following sentence to our manuscript: "A subset of new FL diagnoses at our institution involve patients initially treated for breast cancer."

1. Cuzick, J. *et al.* Tamoxifen for prevention of breast cancer: extended long-term follow-up of the IBIS-I breast cancer prevention trial. *Lancet Oncol* **16**, 67–75 (2015).
2. Kang, D. *et al.* Risk of non-Hodgkin lymphoma in breast cancer survivors: a nationwide cohort study. *Blood Cancer J* **11**, 200 (2021).

As detailed in Huet *et al.*³, patients and samples from the PRIMA cohort were selected as follows: "Fresh-frozen tumor-biopsy specimens were obtained from 159 FL patients included in the open-label, international, multicenter randomized PRIMA study, which enrolled 1135 patients with untreated high-tumor burden FL (NCT00140582). Tumor biopsies were obtained at FL diagnosis and stored at the bio-bank Plateforme de Ressources Biologiques, Henri Mondor Hospital, Créteil (bio-bank ID number: BB-0033-00021)." As a consequence, gene expression signatures observed in both our cohort and the PRIMA cohort remain naive from potential influence of therapy.

3. Huet, S. *et al.* EZH2 alterations in follicular lymphoma: biological and clinical correlations. *Blood Cancer J* **7**, e555–e555 (2017).

9. Figure 4C shows a Volcano Plot of the altered genes in the comparison between EZH2mut and WT in Follicular Lymphoma. Are the same genes that the authors observed in the previous comparison between iMEFs with EZH2mut and EZH2 WT (Figure 3A)?

The question raised here by the reviewer is important but difficult to address considering how different the two models are. Indeed, on one side, we have an isogenic cell model where *EZH2* status is the only genetic parameter that varies. On the other side, we are comparing genetically heterogeneous tumors that have evolved with a mutation of *EZH2* versus another set of heterogeneous tumors that have evolved in an *EZH2*-wild-type context. Consequently, while we have a clear and robust signature in the first case, the picture is noisier when comparing tumors in the second case. This is perhaps best illustrated by the dramatically higher number of both up- and down-regulated genes in *EZH2*-mutant versus

EZH2-wild-type OCI-Ly19 lymphoma cells as compared to the *EZH2*-mutant versus *EZH2*-wild-type tumor samples.

The Venn diagrams reveal a partial overlap between the two sets of genes but we do not feel comfortable making strong conclusions considering the distinct nature of the samples being compared.

10. In Figure 4B there are some cases in which the *EZH2* mutation disappears (P42 and P162). How can you explain this phenomenon? Is it due to tumor heterogeneity? it would be convenient to carry out a single cell expression analysis to better define the alterations caused by the *EZH2* mutation and circumvent this problem.

Indeed, we observe a change in *EZH2* mutational status over time in 4 out of 10 patients included in our longitudinal study. As the reviewer suggests, this phenomenon most likely reflects intratumoral heterogeneity. We are confident that the *EZH2* mutation is indeed absent in the relevant samples: while the variant allele frequency (VAF) threshold we applied for detecting mutations using the DRAGON panel overall was 5%, we further verified for these four patients that the samples classified as wild-type for *EZH2* lacked any detectable mutant reads, irrespective of the threshold. To strengthen this observation, we performed RNA-seq on the samples undergoing a change in *EZH2* mutational status (as part of the analysis now presented in Fig. 5d and Supplementary Fig. 5b) and could not detect any reads in the samples classified as *EZH2*-wild-type that would suggest the existence of subclonal *EZH2*-mutant populations. These observations rule out, in our view, the possibility that the fragment of tumor included in the biopsy contains minor *EZH2*-mutant clones.

Importantly, in a recent study published by Schrorers-Martin et al.⁴, the authors also demonstrate tumor heterogeneity within patients in a longitudinal FL cohort using ultra-deep sequencing technology (CAPP-Seq) that can detect mutations with VAF < 0.05%. It is thus very likely that sub-clones emerge/disappear through time and space either as a reflection of the natural history of the disease or through treatment selection. Single-cell expression would indeed represent an interesting approach to unveil the extent and effect of such heterogeneity in general; however, beyond the fact that our retrospective tumor bulk-based study design limits us in the utilization of such technology, it is unclear that it would improve our sensitivity to detect rare subclonal populations. Besides, to assess the

potential space heterogeneity, we believe that such a study would require evaluating more than one biopsy/patient /time point and such material is not available to us.

4. Schroers-Martin, J. G. *et al.* Tracing Founder Mutations in Circulating and Tissue-Resident Follicular Lymphoma Precursors. *Cancer Discov* **13**, 1310–1323 (2023).

11. Supplemental Table 2 shows the different type of EZH2 mutations included in this study. These mutations are presented in different proportions. Among them, are differences in terms of cause-effect on enzymatic activity or in global molecular alterations? All the mutants have the same behavior as the one with the highest incidence (Y646N)? Have these mutations similar RNAseq, CHIP-seq, etc... profiles?

We thank the reviewer for urging us to investigate this very interesting point. We address it in several ways in our revised manuscript.

First, we generated iMEF cells lacking both *Ezh1* and *Ezh2* and then re-introduced *Ezh2*^{WT}, *Ezh2*^{Y641F} or *Ezh2*^{Y641N} (Fig. 2b in the revised version of our manuscript). This experiment reinforced our previous conclusion that the *Ezh2*^{Y641F} allele is capable, alone and in the absence of a wild-type counterpart, of generating high bulk cellular levels of H3K27me3, this time even showing that it can restore such levels from zero. By comparison, we found that the *Ezh2*^{Y641N} variant is very strongly impaired. The current consensus in the field is that mutant and wild-type alleles must cooperate to produce excess H3K27me3 levels; while our *Ezh2*^{Y641F} results challenge the universality of this view, our *Ezh2*^{Y641N} indicate that it is indeed accurate for some variants.

Second, we conducted *in vitro* histone methyltransferase assays on recombinant oligonucleosome substrates to compare the activity of PRC2-Ezh2^{Y641N} to that of the wild-type and Ezh2^{Y641F}-containing complexes already analyzed in the original version of the manuscript. While the activity of PRC2-Ezh2^{Y641F} is reduced compared to the wild-type, we found that PRC2-Ezh2^{Y641N} displayed the weakest activity (Fig. 2c in our revised manuscript). These results are in agreement with the findings of Sneeringer *et al.*, 2010⁵. The addition of PRC2 accessory subunits AEBP2 and JARID2 in our experiment enabled PRC2-Ezh2^{Y641F} to exceed the basal activity of the wild-type complex, which we previously speculated may explain the ability of this variant to produce high levels of H3K27me3 even in cells lacking wild-type Ezh2. While AEBP2 and JARID2 also enhance the activity of PRC2-Ezh2^{Y641N}, even this enhanced level remains very low, also consistent with the inability of *Ezh2*^{Y641N} to generate robust bulk H3K27me3 levels in iMEF cells when present as the only *Ezh2* allele (Fig. 2b).

5. Sneeringer, C. J. *et al.* Coordinated activities of wild-type plus mutant EZH2 drive tumor-associated hypertrimethylation of lysine 27 on histone H3 (H3K27) in human B-cell lymphomas. *Proceedings of the National Academy of Sciences* **107**, 20980–20985 (2010).

Third, we compared FL samples according to the type of *EZH2* mutation they harbored. Given the restricted number of patients included in our series assessed by CHIP-seq/RNA-seq, we could not draw any solid conclusion. We therefore took advantage of the larger

sample size of the PRIMA cohort (n = 146 patients, including 41 mutant cases, see PCA below, left panel, from Supplementary Fig. 4c) to determine whether mutant samples clustered according to their mutation type at the transcriptomic level. As illustrated by the color-coded PCA below (right panel), *EZH2* mutation type does not allow us to infer a putative stronger impact of one mutation type compared to another on the *EZH2*-mutation-related transcriptomic signature that we observe in FL.

Minor comments:

1. What have been the criteria to select the chromatin regions of CUT&RUN tracks for the comparison between the mutants throughout the article?

We selected regions that we considered representative of the overall trends we sought to illustrate in each track, with no other specific criteria. Importantly, our conclusions are also supported in each case by systematic analyses that do not depend on the selection of particular regions.

2. It seems that graphs titles of Figure 2D are incorrect according to the associated text in results section and y-axes of the graphs.

Upon careful examination, we have determined that these graphs, which now appear in Supplementary Fig. 2c of our revised manuscript, are in fact correctly labeled. We state in the text that H3K27me3 levels show “sharp declines at most WT peaks of enrichment accompanied by broadly distributed gains elsewhere.” This is indeed what is shown in the two graphs labeled “H3K27me3” on the vertical axis: strong loss of H3K27me3 at H3K27me3 peaks (third graph from left) and gains elsewhere, here at H3K27me2 peaks (leftmost graph). We also conclude that “H3K27me2 is substantially reduced,” which is also what is shown in the two graphs labeled “H3K27me2” on the vertical axis, with falling H3K27me2 values at both H3K27me2 and H3K27me3 peaks.

3. Figure 3G and Sup 3B are duplicated.

We agree with the reviewer that this information was redundant. As described in our response to major points 6 and 7 above, we now instead complement the GO analysis in Fig.

3g with a more granular analysis of the individual clusters defined in Fig. 3e. This is displayed in Supplementary Fig. 3f in our revised manuscript.

Reviewer #2 (Remarks to the Author):

This manuscript aims to investigate the role of aberrant PRC2 activity in tumorigenesis through the use of genome-wide chromatin immunoprecipitation (ChIP) and RNA sequencing. Specifically, the study intends to elucidate the molecular mechanisms underlying how misregulation of H3K27 methylation promotes cancer development. To achieve this, the study constructs isogenic cell lines that recapitulate each relevant mutation and explores how these mutations contribute to tumor development and progression.

The study found that different mutations lead to a loss of function of the PRC2 complex, with the degree of loss increasing in the order of H3.3K27M, Ezh2-KO, and Eed-KO. **These findings are consistent with previous studies.** In addition, the study investigated EZH2 gain-of-function mutations in follicular lymphoma and the impact of the Ezh2-Y641F substitution on the cellular balance of H3K27me_{2/3}. The results of this investigation are similar to those observed in previous studies carried out in DLBCL cells. As noted in several previous studies, most recently by Beguelin et al. 2020 and Zimmerman et al. 2022, expression of Y641F leads to increased H3K27me₃ and the up- or down-regulation of hundreds of genes, including genes involved in antigen presentation. The mutation also alters the transcriptome's response to PRC2 inhibition, resulting in the up-regulation of different target genes than in wild-type cells.

The authors also examined a cohort of follicular lymphoma patients and found that 19.4% had gain-of-function EZH2 mutations. They narrowed their cohort to 29 patients, 19 of whom were EZH2-mutant. The study found that EZH2-mutant FL tumors displayed altered H3K27me₃ profiles, with H3K27me₃ marks adopting a flatter profile than in EZH2-WT FL. The analysis also revealed heterogeneity in the chromatin landscape at different time points, likely due to the coexistence of distinct clones with different genetic backgrounds within the same patient.

While the study's strengths lie in the collection of follicular lymphoma patient samples, there are several weaknesses in the study design. These include the lack of new mechanistic insight into the histone or EZH2 mutations and the limited understanding of the disease mechanism of EZH2 mutations in follicular lymphoma. Additionally, the experiments depicted in Figure 1 (K27M) seem out of place in the study, which otherwise focuses on the Y641F mutation. Furthermore, **these experiments do not effectively recapitulate previous studies** and may have been conducted inadequately.

We thank the reviewer for his/her attentive critique; however, we respectfully disagree with his/her assessment on three points:

- The reviewer considers our findings on the cellular impact of *EZH2* gain-of-function mutations to lack novel mechanistic insight. Yet, while Ennishi et al. recently showed that EZH2 inhibitors promote MHC expression in *EZH2*-mutant DLBCL cell lines⁶, we further find here that these mutations are sufficient to sensitize an antigen presentation gene expression program to activation by EZH2 inhibitor treatment independently of the cellular context. Our work therefore materially strengthens the findings of Ennishi et al. and altogether this represents an important, therapeutically relevant advance that goes beyond the excellent recent studies the reviewer mentions.

6. Ennishi, D. *et al.* Molecular and Genetic Characterization of MHC Deficiency Identifies EZH2 as Therapeutic Target for Enhancing Immune Recognition. *Cancer Discov* **9**, 546–563 (2019).

- The reviewer implies that our findings on *H3.3K27M* are insufficiently novel by stating that they are “consistent with previous studies” while at the same time faulting them because they “do not effectively recapitulate previous studies” (passages highlighted in green). We are not sure where we should stand with respect to these opposing lines of critique, except to say that we address an existing area of debate in the field using well-defined tools and provide a clear conclusion. We believe that the data as presented in Fig. 1f,g and Supplementary Fig. 1h in our revised manuscript make the case that *H3.3K27M* acts as a straightforward PRC2 loss-of-function mutation in a manner that is novel and more compelling than previously published evidence.
- Lastly, if the impression that the experiments “may have been conducted inadequately” stems from doubts regarding our ChIP-seq normalization procedures, we now realize we did not sufficiently stress that the bioinformatic package we used (csaw) includes a TMM-based normalization step that performs similarly to spike-in-based approaches in our hands. This is further detailed in our response to point 4 below.

Therefore, while the study provides some insights into the role of aberrant PRC2 activity in tumorigenesis, it falls well short of providing sufficient novel information to justify publication in *Nature Communications*.

For the reasons argued above and in light of our responses to the reviewer’s specific comments below, our view is that our study does indeed provide the level of novel information that befits an article in *Nature Communications*.

1) The initial title of the results section, “The H3.3K27M oncohistone induces a straightforward PRC2 loss of function,” is not an accurate reflection of the complex effects of the K27M histone mutation on gene expression. While some studies have reported global reductions in H3K27me3 associated with K27M, others have shown that this mutation can lead to residual peaks of H3K27me3 at specific genes that remain repressed.

In contrast to the complicated effects of the H3.3K27M mutation on PRC2 function, mutations in EZH2 associated with Weaver Syndrome appear to result in a more straightforward loss of function of PRC2. In a study by Lee et al. published in *Molecular Cell* in 2018, the authors investigated the effects of mutations in EZH2 on PRC2 function. They found that mutations that disrupted the SET domain of EZH2, which is responsible for its catalytic activity, resulted in a significant reduction in both H3K27me2 and H3K27me3 levels.

The effects of the H3.3K27M mutation on gene expression have been a subject of much research and have yielded some intriguing and paradoxical results. While it is often associated with global reductions in the H3K27me3 histone modification, which is typically associated with gene repression, it is also associated with remaining peaks of H3K27me3 at specific genes that remain repressed. This apparent contradiction has been resolved by recent studies. Specifically, that K27M inhibits the allosterically activated form of PRC2, which may lead to global reductions in H3K27me3, as well as the upregulation of many genes that are normally repressed by PRC2 (Stafford et al 2018, Jain et al. 2020). However, the targeting of PRC2 to CpG islands and H3K27me3 near target sites is unaffected. This proposed mechanism helps to reconcile the seemingly paradoxical results of ChIP-seq experiments on K27M histones and provides a more comprehensive understanding of the effects of this mutation on gene expression.

We are in agreement with the reviewer that the literature on the *H3.3K27M* mutation since its discovery 10 years ago is varied in its conclusions. Our main assertion regarding *H3.3K27M* in this study is that its impact on PRC2 activity is to cause a loss of function, and that this is distinguishable only by degree of severity from other instances of loss of function such as deletions of subunits such as those we have made here or Weaver-syndrome-associated mutations such as those examined in the Lee et al., 2018 *Mol. Cell* paper⁷ or in the recent preprint from the Bracken group⁸. Our view challenges the interpretations put forward in some studies of *H3.3K27M*, which have contended that H3K27me3 undergoes significant focal gains. We do not observe this phenomenon when we search for it systematically; indeed, Fig. 1f shows that gains are anecdotal. We conclude that the intrinsic impact of the mutation, as assessed in our isogenic system, does not include this feature, but are careful to acknowledge that it may exist as a specificity of certain cell types. Importantly, our contention that *H3.3K27M* causes a loss of function less severe than *Eed*-KO and *Ezh2*-KO is completely compatible with some residual H3K27me3 (as we show for instance in Fig. 1d, previously Fig. 1c) and residual gene repression; these observations do not undermine our conclusions.

7. Lee, C.-H. *et al.* Allosteric Activation Dictates PRC2 Activity Independent of Its Recruitment to Chromatin. *Molecular Cell* (2018). doi:10.1016/j.molcel.2018.03.020
8. Deevy, O. *et al.* Dominant negative and directional dysregulation of Polycomb function in EZH2-mutant human growth disorders. *bioRxiv* 2023.06.01.543208 (2023). doi:10.1101/2023.06.01.543208

We therefore stand by the heading of the first subsection of the Results, “The H3.3K27M oncohistone induces a straightforward PRC2 loss of function.” To strengthen the evidence in support of this statement, we conducted additional analysis now included in the revised

version of our manuscript. Specifically, we compared the effect of *H3.3K27M* on H3K27me3 intensity within peak regions genome-wide with that of treating cells with a low dose of the EZH2 inhibitor UNC1999. We found that the two perturbations led to a remarkably similar distribution of changes at peaks across the genome (Fig. 1f,g), with ranked plots of the individual peaks confirming highly concordant effects (Supplementary Fig. 1h). Separately, we performed a pre-ranked gene-set enrichment analysis (GSEA) to determine whether genes that are up- or down-regulated by the loss-of-function mutations *Eed*-KO and *Ezh2*-KO are also respectively up- or down-regulated upon introduction of *H3.3K27M* (as a complement to the Venn diagrams included previously). This analysis (Fig. 1c and Supplementary Fig. 1f in our revised manuscript), confirms that *H3.3K27M*-expressing cells indeed undergo gene expression changes that overlap extensively with those observed in *Eed*-KO and *Ezh2*-KO, and thus present a strong PRC2 loss-of-function signature, albeit of a lower magnitude than *Eed*-KO and *Ezh2*-KO. Altogether, the new evidence we include in the revised manuscript reinforces the notion that cells carrying the *H3.3K27M* oncohistone exhibit a straightforward loss of function of PRC2, comparable to that induced by partial pharmacological inhibition of the enzyme.

2) The authors of the study appear to have used a Western blot and ChIP-seq for H3K27me3 in K27M cells to support their conclusion that the mutation induces a straightforward loss of function in PRC2. However, these experiments actually show anything but a straightforward loss of function. For example, Figure 1C of the study demonstrates that residual peaks of H3K27me3 remain in K27M cells, suggesting that the mutation is not universally disrupting PRC2 function. Additionally, Figure 1B shows that only a subset of genes become derepressed or upregulated upon *H3.3K27M*, further indicating that the effects of the mutation are complex and not necessarily straightforward.

We would like to stress that while we think *H3.3K27M* induces a loss of function in PRC2—as strongly supported by the comparison with the effects of EZH2 inhibitor treatment described in response to point 1 above for example (Fig. 1f, g and Supplementary Fig. 1h)—this loss of function is of a lesser magnitude than that caused by deletion of *Ezh2* and *Eed*. As a consequence, the loss of H3K27me3 peaks and the target gene de-regulation are similar to *Ezh2*-KO and *Eed*-KO in terms of affected genes and regions, but lesser in degree, as described in detail in our response to point 1 above. This accounts for the observations pointed out by the reviewer, but we see no contradiction here between the data we present and the conclusion we draw. Our newest analysis strengthens our view that the effects of the *H3.3K27M* mutation can be categorized straightforwardly as a loss of function of PRC2.

3) Moreover, the Western blot in Figure 1A contradicts published reports of a "clear decrease" in H3K27me2/3 in cells expressing *H3.3K27M*. This discrepancy goes beyond the nuance of K27M effects on K27me3 - it suggests a technical/experimental flaw. Where is the K27me3 that apparently exists in K27M cells? It almost certainly isn't outside of K27me3 (WT) peaks but if it is then it needs to be identified/quantified. Something is awry - either the western or the CUT/RUN data.

We acknowledge that the H3K27me3 Western blot shown in Fig. 1a, which is strongly exposed in order to reveal subtle differences between *Ezh2*-KO and *Eed*-KO, does not make the clear reduction of H3K27me3 levels in H3.3K27M-expressing cells as evident as it ought to be. A new Western blot comparing the same wild-type and H3.3K27M samples is now presented in Supplementary Fig. 1c at an exposure level that more clearly shows the decrease, which indeed occurs in line with expectations.

4) The data presented in Figure 1 of CUT&RUN experiments have been read-normalized. However, it is critical to note that quantitative comparisons made without spike-in normalization are highly questionable. Spike-in normalization is essential to compare the CUT&RUN data with different samples and experiments accurately. Without spike-in normalization, it is difficult to ascertain the actual differences between samples, and any comparisons made may be misleading.

We agree with the reviewer that scaling CUT&RUN data solely by the size of the sequencing libraries can lead to erroneous conclusions; uniform changes in enrichment in particular may not be apparent.

For this reason our CUT&RUN data were not merely read-normalized, but also further scaled to correct for composition biases using the trimmed mean of M-values (TMM) method within the *csaw* package. This is a widely used approach (see for example refs. ⁹⁻¹¹, and ref. ¹² for a review). The relevant text in the “CUT&RUN-seq data analysis” subsection of the Methods is as follows:

“In addition to scaling by library size, in order to remove any composition biases, we calculated a normalizing factor using the Bioconductor *csaw::normFactors* function^{13,14}. This function counts reads in 10- kb genome-wide non-overlapping bins and uses the trimmed mean of M-values (TMM) method to correct for any systematic fold change in the coverage of bins. The normalization factor was passed onto DeepTools *bamCoverage* with the parameter `--scaleFactor`.”

9. Galvis, L. A. *et al.* Repression of *Igf1* expression by *Ezh2* prevents basal cell differentiation in the developing lung. *Development* **142**, 1458–1469 (2015).
10. Kolev, H. M. *et al.* H3K27me3 Demethylases Maintain the Transcriptional and Epigenomic Landscape of the Intestinal Epithelium. *Cell Mol Gastroenterol Hepatol* **15**, 821–839 (2023).
11. Nagano, M. *et al.* Nucleome programming is required for the foundation of totipotency in mammalian germline development. *EMBO J* **41**, e110600 (2022).
12. Servant, N. Bioinformatics Methods for ChIP-seq Histone Analysis. *Methods Mol Biol* **2529**, 267–293 (2022).
13. Lun, A. T. L. & Smyth, G. K. De novo detection of differentially bound regions for ChIP-seq data using peaks and windows: controlling error rates correctly. *Nucleic Acids Res* **42**, e95 (2014).
14. Lun, A. T. L. & Smyth, G. K. *csaw*: a Bioconductor package for differential binding analysis of ChIP-seq data using sliding windows. *Nucleic Acids Res* **44**, e45 (2016).

In parallel, we also calculated scaling factors using *Drosophila melanogaster* reads from spike-in material included in our experiments but not discussed in the manuscript. These

scaling factors are nearly proportional to the ratio of mouse to *Drosophila* reads, but not precisely so, since they are computed based on a median ratio across bins. This procedure is further described in ref. ¹⁵.

15. Holoch, D. *et al.* A cis-acting mechanism mediates transcriptional memory at Polycomb target genes in mammals. *Nat Genet* **53**, 1686–1697 (2021).

The table below shows the read numbers and scaling factors for the wild-type and H3.3K27M H3K27me3 CUT&RUN experiments presented in Fig. 1.

Sample	Genotype	Total reads	M. musculus reads	D. melanogaster reads	Scaling factor (csaw)	Scaling factor (spike-in)
D127C53	WT	41 486 896	41 258 918	214 322	1.604102	1.16242
D127C54	WT	46 194 320	45 935 832	242 484	1.487783	1.0181
D127C59	H3K27M	36 464 682	35 545 448	872 278	0.7813	0.2822
D127C60	H3K27M	28 221 356	26 733 984	1 413 468	0.7390	0.1738

Due to the nature of the FL patient biopsies used for ChIP-seq, we could not quantify the starting chromatin with sufficient accuracy to enable addition of an exogenous spike-in. Therefore, in order to apply a uniform normalization procedure throughout the study, we opted to use the TMM-based scaling provided by csaw for both the FL samples and the iMEF CUT&RUN.

However, it is worth noting that had we used spike-in-based normalization for the CUT&RUN analysis, the H3K27me3 enrichment levels in the *H3.3K27M* samples would have been even lower relative to wild-type (compare the scaling factors in the last two columns of the table). Detection of retained peaks in this scenario would thus have been even less likely.

5) The authors of this study appear to suggest that their isogenic system is unique, but in reality, it is not a novel approach. Many other studies have previously manipulated K27M mutations in different isogenic contexts, and this is a well-established method in the field.

It is of course the case that other studies have examined the impact of *H3K27M* mutations using isogenic cell lines. However, we stand by the statement that the full set of isogenic cell lines presented in our manuscript with different mutations affecting PRC2 activity, also including subunit deletions and a gain-of-function mutant, is unique (first paragraph of the result section). Yet, we have now removed this adjective from the first paragraph of the discussion where we did not specifically refer to "the full set of isogenic cell lines".

6) It is important to note that, despite some incongruencies and suboptimal spike-in calibration in Figures 1A and 1D, the decreased K27me3 signal in 1D is not necessarily inconsistent with other reports. The studies referenced by the authors and others have reported a paradoxical retention of K27me3 to occur at a subset of PRC2 target sites, while other sites show decreased K27me3 signal in K27M cells. Furthermore, some studies have reported similar K27me3 enrichment only at retained sites, while others that display K27me3 enrichment (by heatmap or metaplot) at all target sites in WT cells show the same decreased K27me3 signal as reported in Figure 1D.

The average density plot shown in Fig. 1e of our revised manuscript (Fig. 1d in the previous version), which appears to show a collapse in H3K27me3 at target sites when *H3.3K27M* is introduced, could indeed, as the reviewer suggests, mask a heterogeneous situation in which a small subset of sites retains a high degree of enrichment.

However, this scenario appears to be ruled out by our analysis of the change in enrichment at H3K27me3 peaks genome-wide, included in the previous version of the manuscript and now displayed in Fig. 1f. Indeed, only a very small share of peaks remain near their wild-type enrichment levels in the presence of *H3.3K27M* (log fold-change ≈ 0), and these show relatively low starting levels of H3K27me3 to begin with. The vast majority of peaks show a several-fold reduction in H3K27me3, and this effect is more pronounced the higher the initial H3K27me3 enrichment (note the downward slope of the cloud). Interestingly, these changes mirror those induced by treatment with a low dose of an EZH2 inhibitor (compare to Fig. 1g in our revised manuscript). Importantly, a plot of H3K27me3 density across all of these regions (Supplementary Fig. 1h in our revised manuscript) further confirms the absence of H3K27me3 signal retention.

We conclude that while the H3K27me3 losses caused by *H3.3K27M* may be accompanied by retained peaks in certain cell types, this is not an intrinsic property of the mutation since it is not observed in our cell line.

We address the technical concerns the reviewer raises in our responses to points 3 and 4 above.

7) The authors' conclusion that the *H3.3K27M* mutation leads to a straightforward loss of function of PRC2, with paradoxical local effects being specific to certain cellular contexts rather than a general property of H3K27M, is not accurate. Their conclusion does not align with the literature on this topic. Furthermore, the authors' own data does not support their conclusion that the *H3.3K27M* mutation causes a global loss of K27me3. Their K27M+ MEFs display near-equal levels of K27me3 globally compared to WT, as shown in Figure 1A by western blot analysis. This finding suggests that the global loss of K27me3 seen in other systems is context-dependent. As Stafford et al. 2018 demonstrated, the molar ratio of PRC2:K27M in cells determines the degree of PRC2 inhibition and, therefore, the global K27me3 levels. The PRC2:K27M molar ratio in these MEFs may be too high to cause dramatic decreases in K27me3, whereas in human DMGs, the opposite is true.

As detailed in our response to point 3 above, the impression that H3K27me3 levels in *H3.3K27M*-expressing iMEFs are nearly equal to those of their wild-type counterparts based on Fig. 1a has more to do with the exposure of the Western blot (designed to reveal slight but important differences between *Ezh2*-KO and *Eed*-KO) than with the actual levels. Supplementary Fig. 1c in our revised manuscript confirms unambiguously, using the same samples, that H3K27me3 is substantially reduced.

We fully agree with the reviewer and with the elegant Stafford et al., 2018 study¹⁶ that the cellular abundance of H3.3K27M relative to PRC2 is an important determinant of the impact of the mutation on H3K27me3 levels. Our contention here is simply that the intrinsic action of *H3.3K27M* is to cause a loss of function in PRC2—which we think the previous and new data we describe in our response to point 1 above demonstrate compellingly—but that previously noted paradoxical effects are not part of that intrinsic action. If they were, then the assays and analyses we conducted using the iMEFs would have detected them. We do not dispute these phenomena, but merely conclude that they are not a systematic consequence of the mutation that doesn't depend on the cell type context.

16. Stafford, J. M. *et al.* Multiple modes of PRC2 inhibition elicit global chromatin alterations in H3K27M pediatric glioma. *Sci Adv* **4**, eaau5935 (2018).

8) The results presented in Figure 2 are not surprising since H3K27-methylated nucleosomes persisted in Y641F cells even after depletion of EZH1 and wildtype EZH2. This suggests that Y641F might be responsible for maintaining the methylation of these nucleosomes in the absence of the wildtype enzyme. However, to fully understand Y641F's ability to methylate unmodified nucleosomes in a cellular setting, it is crucial to delete both wildtype EZH2 and EZH1, which would effectively start with no H3K27 methylation, and then express Y641F. Only then can we accurately assess the role of Y641F in maintaining H3K27 methylation. The authors cannot make conclusions about Y641F in the context of pre-existing K27me1.

We thank the reviewer for raising this interesting question: indeed, the ability of PRC2-Ezh2^{Y641F} as the only variant of PRC2 in the cell to continue to trimethylate H3K27 could depend on the presence of H3K27 methylation previously installed by wild-type *Ezh1* and *Ezh2*. To test this idea we followed the reviewer's suggestion and deleted both *Ezh1* and *Ezh2* from iMEFs, before then re-introducing *Ezh2*^{WT} or *Ezh2*^{Y641F} *de novo*. We found that H3K27me3 levels were similar in these two conditions (Fig. 2b in our revised manuscript). We conclude that PRC2-Ezh2^{Y641F} can mediate the accumulation of high levels of cellular H3K27me3 without cooperating with a wild-type enzyme and independently of pre-existing H3K27 methylation.

Furthermore, Sneeringer et al. 2010 demonstrated in their Figure 2 and Table 1 that Y641F has some ability to methylate unmodified H3K27, unlike Y641N/H/S mutations. Therefore, it is possible that if the authors had used a Y641N mutation in the current study, they might not have observed the persistence of H3K27me3 in the same way as Y641F. This highlights the importance of studying different Y641 mutations to gain a better understanding of their effects on H3K27 methylation and their role in disease development. However, it is worth noting that Y641 mutations always occur as heterozygotes in the disease, and it is unclear

how the present study expands our understanding of Y641F or its role in the development of DLBCL or FL.

We take the reviewer's point that the behavior of Ezh2^{Y641F} may not be representative of the totality of gain-of-function mutants observed in FL and DLBCL. We therefore examined the H3K27 methylation ability of Ezh2^{Y641N} both *in vitro* and in cells. Consistently with the Sneeringer et al., 2010 study⁵, we observed a far lower activity for Ezh2^{Y641N} than for Ezh2^{Y641F} in an *in vitro* histone methyltransferase assay using recombinant nucleosomes, and the activity remained very weak even in the presence of JARID2, AEBP2 and H3K27me3 stimulatory peptides (Fig. 2c in the revised version of our manuscript). Meanwhile, Ezh2^{Y641N} was also unable to appreciably methylate H3K27 *de novo* in cells when introduced as the only PRC2 catalytic subunit, unlike Ezh2^{Y641F} (Fig. 2b in our revised manuscript).

5. Sneeringer, C. J. *et al.* Coordinated activities of wild-type plus mutant EZH2 drive tumor-associated hypertrimethylation of lysine 27 on histone H3 (H3K27) in human B-cell lymphomas. *Proceedings of the National Academy of Sciences* **107**, 20980–20985 (2010).

We therefore now draw a more nuanced conclusion, namely that EZH2 gain-of-function mutants may need to cooperate with wild-type PRC2 in order to drive aberrant H3K27 methylation patterns, but that, as illustrated by the case of Y641F, this is not an obligate property of all of these mutants. We understand the reviewer's doubts regarding the relevance of these findings, given that *EZH2*-mutant tumors are invariably heterozygous, but we would argue that it is valuable to understand precisely which molecular features are common to all of the mutants and which characterize only a subset.

9) The authors conducted a comprehensive survey of follicular lymphoma samples containing EZH2 mutations and wildtype EZH2, using ChIP-seq to analyze the H3K27me3 and H3K27ac profiles. The results revealed that EZH2 gain-of-function mutations create a distorted H3K27me3 profile independent of other mutations, which was not unexpected given the known role of EZH2 in H3K27 methylation. The small differences observed in H3K27ac profiles between wildtype and mutant EZH2 ChIP-seq patterns were mostly expected, as the major H3K27ac peaks are found at gene enhancers related to cell type identity ('super enhancers') and would persist despite gain of EZH2 activity elsewhere in the genome. The small number of peaks that do change in Figure 4I are likely related to the differences in gene expression resulting from the gain of H3K27me3 in EZH2 mutants.

While the longitudinal studies on follicular lymphomas containing the EZH2 mutations showed differences in EZH2 status over time, the authors did not provide clear conclusions regarding these experiments, except to say that there is heterogeneity among different patients. It is possible that there are many patient-specific variables that are unknown, which could affect the outcome of the study. Moreover, it is unclear how this result informs our understanding of EZH2 mutations or follicular lymphoma.

We respectfully disagree with the reviewer. The extent of our ChIP-seq analysis on FL patient samples is without precedent in the field, and while it is certainly not unexpected that *EZH2*

mutations should be associated with altered H3K27me3 patterns, our study reveals the detailed nature of these alterations and their systematic occurrence regardless of other genetic lesions. These represent novel insights.

Perhaps most crucially, our longitudinal study demonstrates that individual patients harbor multiple clones of differing *EZH2* mutational status more frequently than was previously appreciated. The therapeutic implications of this observation are tremendous given the differential efficacy of PRC2 inhibitor treatment as a function of *EZH2* status. In the future, taking this potential heterogeneity into account for patient stratification could yield significant benefits in terms of disease outcomes.

Overall, while the survey conducted by the authors provides important insights into the distorted H3K27me3 profile caused by *EZH2* gain-of-function mutations, the results are not entirely surprising. However, the small differences observed in H3K27ac profiles and the longitudinal studies on follicular lymphomas containing *EZH2* mutations provide additional information that could be further explored in future studies.

We thank the reviewer for recognizing the value of the work and of its future directions, as well as for the critical feedback.

Reviewer #3 (Remarks to the Author):

Romero et al. report on the effects of various *EZH2* mutations on H3K27 methylation profiles and the transcriptome, both in embryonic fibroblasts (iMEF cells) and follicular lymphoma (FL) samples. Using isogenic lines of iMEFs, they carefully document the profiles observed as a result of different *EZH2* levels of activity, with a particular focus on a mutation that causes *EZH2* to be hyperactive (Y641F). Whereas H3K27 methylation profiles can be rescued by inhibiting the hyperactive *EZH2* enzyme, the transcriptome effects cannot, a finding that is currently not explained.

In the second part of the manuscript, they analyze the effects of hyperactive *EZH2* in human patient FL samples with or without the mutation. The global profiles of H3K27 methylation tend to correlate well with *EZH2* mutation status.

This is a manuscript with many interesting observations and a careful experimental approach. The comparison between 'clean' genetic changes (in the iMEF cells) and human patient FL samples adds interest, depth and complexity to the study. The overall redistribution of H3K27me3 with hyperactive *EZH2* is not a new finding (for example Souroullas et al. Nat Med. 2016, cited by the authors). On a technical level, the authors need to address how quantitative the CUT&RUN analyses are and how this impacts the analysis. Also, the relationships between H3K27 methylation profiles and transcriptomic changes remain unclear.

We thank the reviewer for his/her positive feedback while taking note of the potential weaknesses he/she identifies. We are confident in the quantitative accuracy of our CUT&RUN analyses and provide supporting arguments to this effect in our response to point 2 below. Meanwhile, we agree that the decoupling of H3K27 methylation profiles and transcriptomic changes in *Ezh2*-mutant iMEFs treated with the EZH2 inhibitor is one of the most intriguing aspects of our results; we endeavor to clarify our interpretation of these findings in our response to points 6 and 7 below.

Specific comments:

1. Regarding the H3K27M oncohistone, the authors report, in contrast to other studies, “In our isogenic model, however, H3K27me3 losses are widespread and individual H3K27me3 peaks almost universally display a substantial reduction in enrichment (Figure 1D-E).” However, Fig. 1D appears to show for one particular interval on chr13, relatively strong retention of H3K27me3 at some peaks and a much stronger reduction elsewhere. In a global quantitative analysis (MA density plot over a set of 17,000 peaks; Fig. 1E), many peaks are reduced by about 5-fold ($2\text{LogFC} \sim -2$). However, the figure shows a cloud with a very broad distribution of fold changes in oncohistone versus WT cells, ranging from unchanged ($\text{Log}_2\text{FC} \sim 0$) to strongly down ($\text{Log}_2\text{FC} \sim -5$). To me this all seems in relatively good agreement with previous conclusions contradicted by the authors. This also relates to the conclusion at the end of the first paragraph of the results “that H3.3K27M represents a straightforward loss-of-function mutation”. This appears to be an oversimplification that is not fully supported by the data shown.

We agree with the reviewer that the fold-change (FC) values of H3K27me3 in oncohistone *versus* wild-type are heterogenous; however, FC can be deceptive and should be interpreted with caution. For instance, when considering poorly enriched regions, we could either observe high FC for modest changes or conversely low FC for significant changes due to background. Besides, since PRC2 is subjected to regulation including a positive feedback loop, it is expected that partially reducing its activity will not translate into a linear reduction of its catalytic output.

To test more directly whether *H3.3K27M*-expressing cells indeed display the effects of a straightforward reduction in PRC2 activity, we repeated the same FC analysis as with *H3.3K27M* (presented in Fig. 1e of the original version and 1f of the revised version of our manuscript) but this time partially inhibiting PRC2 activity in wild-type cells with UNC1999. As shown in Fig. 1g of our revised manuscript, we once again observed a relatively broad distribution of FC consistent with our results with *H3.3K27M*. The striking similarity in the outcomes of these two analyses strengthens our view that *H3K27M* is best understood as a straightforward loss-of-function mutation.

2. How quantitative is the CUT&RUN-seq approach, as implemented? There is no spike-in and the data would need to be quantitative in order to compare conditions. The need for quantitation is more or less a general point for all ChIP/CUT&RUN-type analyses, but it is

even more important in the analyses of EZH2[Y641F], in which case outside-of-peaks reads may represent real signal rather than background noise. RPKM normalization typically corrects for sequencing depth. This works generally well in cases where the majority of sequenced reads corresponds to non-specific genomic background (and therefore is independent of different mutations). With hyperactive EZH2 this requirement appears to be violated, because EZH2 deposits methyl groups genome-wide. This concern applies to both the iMEF experiments as well as the FL analyses (Figs. 2D and 4F). Whereas the genomic redistribution of H3K27me3 in EZH2[Y641F] cells beyond doubt is real, could the observed reduction of signal in WT-peaks be an artefact of normalization?

We share the reviewer's view that quantitative normalization is critical for the proper interpretation of CUT&RUN and ChIP-seq data. As he/she notes, the change in the *relative* distribution of H3K27me3, with proportionally more accumulation of the mark outside of peak regions in mutant cells, is clear and does not depend on the normalization method. But, as the reviewer rightfully emphasizes, whether the *absolute* enrichment of H3K27me3 actually declines at peak regions depends heavily on how the data are scaled.

We describe our quantitative normalization procedure in our response to Reviewer 2, point 4, above. Briefly, we do not solely normalize according to sequencing depth, but also apply a TMM-based scaling approach (as part of the csaw package) which corrects for composition biases of the type identified by the reviewer. Importantly, this allowed us to adopt a uniform normalization method across the study, including our ChIP-seq experiments on FL patient samples which did not lend themselves to spike-in-based normalization due to limitations in quantifying the starting chromatin. Nevertheless, we did include a *Drosophila* spike-in in our iMEF CUT&RUN experiments and explore this normalization strategy, with details described in our response to Reviewer 2, point 4, above and in ref. ¹⁵. The scaling factors we computed for the H3K27me3 samples of interest are in the table below:

Sample	Genotype	Total reads	M. musculus reads	D. melanogaster reads	Scaling factor (csaw)	Scaling factor (spike-in)
D127C53	WT	41 486 896	41 258 918	214 322	1.604102	1.16242
D127C54	WT	46 194 320	45 935 832	242 484	1.487783	1.0181
D127C57	Y641F/WT	78 910 504	78 667 180	226 088	0.7959	1.1025
D127C58	Y641F/WT	59 173 698	58 970 964	188 702	0.7366	1.3220

As the reviewer will notice, had we used the scaling factors resulting from the spike-in analysis rather than the TMM-based factors, the H3K27me3 signal in *Ezh2*^{Y641F/WT} cells would have been approximately twice as high. Our genome-wide analysis presented in the violin plots (now displayed in Supplementary Fig. 2c of the revised manuscript) indicates that the H3K27me3 enrichment at peak regions falls by more than twofold (lower-left plot). Thus, using the spike-in scaling factors, the conclusion that H3K27me3 peak enrichment is

diminished in mutant cells would still hold, though the degree of the decrease would be more modest. In order to independently evaluate the extent of the loss of H3K27me3 peak enrichment, we used quantitative PCR to analyze *Foxa1*, a PRC2 target in this iMEF line that we have studied extensively in the past¹⁵. The result, shown just below, is consistent with the genome-wide picture rendered by TMM-based normalization.

Consequently, we are confident that H3K27me3 peak enrichment is considerably reduced in *Ezh2*^{Y641F}-expressing cells and believe we have used a quantitation strategy that is both accurate and applicable throughout the whole study, including to the patient samples.

15. Holoch, D. *et al.* A cis-acting mechanism mediates transcriptional memory at Polycomb target genes in mammals. *Nat Genet* **53**, 1686–1697 (2021).

3. “addition of AEBP2 and JARID2 stimulated both the WT and the mutant enzymes, allowing the latter to exceed the basal activity of the WT (Figure 2B).” The figure seems to suggest the WT enzyme (left side of Fig. 2B) to be more active than the mutant one (right side).

The reviewer is correct and we apologize for the ambiguity in the wording. We meant to point out that the *Ezh2*^{Y641F} mutant complex appears to become more active when AEBP2 and JARID2 are added than the WT complex is in the absence of these cofactors. We have reworded this sentence in the text to improve its clarity. These data appear in Fig. 2c of the revised manuscript.

4. In Supplemental Figure 2D, there does not seem to be a global increase in H3K27me3 in WT compared to EZH2[Y641F] cells, in contrast to main Figure 2A. Please explain. What is the fold difference? Is the blot overexposed? Can you quantify?

We have now quantified the intensity of the H3K27me3 bands in the Western blots in Fig. 2a and Supplementary Fig. 2f (formerly Supplementary Fig. 2d), subtracting the background and normalizing according to the corresponding background-subtracted H4 signal. As discerned by the reviewer, the abundance of H3K27me3 in the *Ezh1*-KO; *Ezh2*^{Y641F/-} iMEFs is not as high, relative to wild-type cells, in Supplementary Fig. 2f (1.38 times) as it is in Fig. 2a (2.37 times) (see below). Although we used fluorescently labeled secondary antibodies that should

generate a linear signal, the simplest explanation is that this merely represents experimental variability. We remain confident in our conclusion that cells containing PRC2-Ezh2^{Y641F} as their only source of PRC2 have higher overall H3K27me3 than wild-type cells.

From Fig. 2a

From Supplementary Fig. 2f

5. The authors show that pharmacological inhibition of EZH2 in EZH2[Y641F] cells restores a WT-like pattern of H3K27me3. This is an important and interesting experiment that suggests that it is the increased catalytic activity of EZH2[Y641F] that is driving the changes in EZH2[Y641F] cells, rather than some other property of the enzyme that affects its targeting to genomic regions. However, these observations do not explain why canonical H3K27me3 peaks are found to be reduced in EZH2[Y641F] cells (unless they are not really reduced; cf. comment on normalization).

We believe that the direct link we have established between the altered genomic distribution of H3K27me3 in *Ezh2*^{Y641F}-expressing cells and the increased catalytic activity of *Ezh2*^{Y641F} is important, and we are encouraged that the reviewer is of the same mind. We also acknowledge, as the reviewer points out, that the basis of this link is still not fully defined. As we write in the Discussion section of the original version of our manuscript:

“Intriguingly, increased total H3K27me3 accompanied by increased spreading of the mark at the expense of peak enrichment, as occurs upon *Ezh2*^{Y641F} expression, has also been observed in mouse embryonic stem cells (mESCs) maintained in the ground state of pluripotency with MEK1 and GSK3 inhibitors as well as in mESCs or MEFs with impaired DNA methylation. This apparent link between H3K27me3 abundance and H3K27me3 distribution, further supported by our PRC2 inhibition experiment, therefore seems to apply to a range of situations. Understanding the mechanistic underpinnings of this phenomenon will require further study.”

We favor a model in which excess genomic H3K27me3 leads to titration of limiting Polycomb pathway components away from peak regions, where they are normally required in high amounts to produce strong enrichment. This could explain the paradoxically lower accumulation of the mark at canonical peaks. We reasoned that if this hypothesis is correct,

we should expect to observe a partial redistribution of PRC2 genome wide. We therefore performed CUT&RUN to detect Suz12 (shown in Supplementary Fig. 2d in the revised manuscript) and analyzed how its binding to target sites is modulated by *Ezh2*^{Y641F}. Consistently with the reduction of H3K27me3 at these specific regions, we observed a lower enrichment of Suz12. This would tend to support a titration-based model, but further investigation will be required to determine the precise underlying mechanism.

6. Using RNA-sequencing, the authors show there is very little rescue of the EZH2[Y641F]-mediated changes in the transcriptome by EZH2 inhibition (Fig. 3E). “Indirect” or “secondary” as the transcriptional consequences may be, how can this be reconciled with the patterns of H3K27me3 (Fig. 2J) which are interpreted as a rescue? One would expect indirect gene expression effects to be rescued as well. The authors acknowledge this issue, but simply state that the hyperactive EZH2 mutant “reprograms its response to PRC2 inhibition”. Clearly, the enhanced catalytic activity of EZH2[Y641F] is not “necessary and sufficient” to cause the transcriptomic effects. What does that mean, and how would this work if not through genomic histone modification patterns?

The failure to rescue the transcriptome of mutant cells whose H3K27me3 profile has ostensibly been rescued is indeed fascinating and perplexing.

The first point that is important to emphasize is that, as we are careful to specify in the manuscript, it is the average H3K27me3 profiles that are restored upon partial PRC2 inhibition in mutant cells. The pervasive aberrant H3K27me3 landscape is virtually completely corrected (Fig. 2j), but we do not claim that the new pattern is identical to that of wild-type cells at every locus.

Instead, what the transcriptomic data show is that in spite of a return to a seemingly wild-type mode of H3K27me3 deposition overall, the fact that the cells previously experienced the uninhibited *Ezh2*^{Y641F/WT} state has led to gene expression changes that are impossible to reverse simply by restoring physiological PRC2 activity. Thus, the excess catalytic activity of *Ezh2*^{Y641F} need not be present continuously for transcriptional disruptions to persist. In a strict sense, then, the reviewer is right to say that the enhanced catalytic activity of *Ezh2*^{Y641F} is not necessary for transcriptomic alterations to be observed at a given time; this is because previous experience of this enhanced catalytic activity, as our data show, leaves a lasting trace on the transcriptome.

In the Discussion, we relate these observations to our recent study¹⁵, in which we showed that transient inhibition of PRC2 in wild-type iMEFs and neural progenitor cells triggers transcriptional changes that persist even once normal PRC2 activity is restored. This further illustrates the fact that reversing an alteration in a chromatin modification activity does not automatically reverse the resulting transcriptional changes.

15. Holoch, D. *et al.* A cis-acting mechanism mediates transcriptional memory at Polycomb target genes in mammals. *Nat Genet* **53**, 1686–1697 (2021).

To address the question the reviewer poses at the end of his/her comment (“how would this work if not through genomic histone modification patterns?”), we would like to draw attention to the following sentence in our Discussion:

“We also find that PRC2 inhibition leads to radically different transcriptomic changes in WT and *Ezh2*^{Y641F/WT} cells, suggesting that the environment of *trans*-acting regulators of transcription is also substantially altered.”

Thus, beyond the changes in H3K27me3 brought about by *Ezh2*^{Y641F} and which the drug rescues *at a global level*, the changes in expressed transcription factors, resulting directly from H3K27me3 changes or resulting indirectly in turn from gain or loss of expression of other transcription factors, must also be considered. It is not clear to us that all of these changes are expected to be readily reversed by restoring physiological PRC2 activity, and, on this matter, we disagree with the reviewer. Some of the feedbacks that arise could remain in place even after the excess PRC2 activity is reduced, and thus contribute to the persistence of transcriptional alterations. This would then explain why *overall* genomic histone modification patterns do not, alone, control whether the transcriptomic changes are rescued.

7. Also, the summarizing conclusion on p. 10 (“We also find major transcriptomic changes that primarily reflect the indirect impact of a shifting chromatin landscape.”) is not adequate, as there must be something in addition to indirect effects of altered H3K27me3 patterns to explain the lack of transcriptomic rescue of when inhibiting EZH2 activity. This inadequate reasoning is also found on p. 12 (“... through complex secondary effects on the gene-regulatory network”) and in the abstract (“our results provide a mechanistic foundation for understanding how oncogenic PRC2 mutations disrupt chromatin and transcription”).

Here we must respectfully disagree with the reviewer. There are two major types of evidence we present in the paper to argue that the transcriptomic alterations caused by *Ezh2*^{Y641F} are not primarily due to local changes in H3K27me3 at the genes whose RNA levels are altered:

- Genes that become up-regulated upon introduction of *Ezh2*^{Y641F} are largely distinct from those that become up-regulated when wild-type cells are treated with a high dose of a PRC2 inhibitor (Fig. 3b, top). This indicates that most up-regulation events in mutant cells do not involve straightforward de-repression after local loss of H3K27me3; otherwise they would strongly overlap with genes de-repressed by the inhibitor. Similarly, genes that become down-regulated upon introduction of *Ezh2*^{Y641F} are largely distinct from those that become up-regulated when these mutant cells are then treated with the high dose of PRC2 inhibitor (Fig. 3b, bottom). This indicates that most down-regulation events in mutant cells do not involve straightforward *de novo* repression after local invasion of H3K27me3; otherwise the corresponding genes would strongly overlap with those then de-repressed by the inhibitor. Together, these observations quite clearly show that the transcriptomic

impact of the mutation is largely indirect. It is on this basis that we conclude that we “find major transcriptomic changes that primarily reflect the indirect impact of a shifting chromatin landscape.”

- As discussed thoroughly above in response to point 6, restoring physiological PRC2 activity in mutant cells does not restore a wild-type transcriptome. This indicates that besides the shifts in H3K27me3 that occur upon introduction of Ezh2^{Y641F} and which are largely reversible by PRC2 inhibitor treatment, other events take place (and we think the term “complex secondary effects” is appropriate here) that lock many mis-regulated genes into an altered state.

In total, our data demonstrate that the effects of gain-of-function *EZH2* mutations cannot be reduced to simple H3K27me3 acquisition and loss events at individual loci with the expected transcriptional consequences, and furthermore that the effect of PRC2 inhibition in mutant cells is not to restore a wild-type-like gene expression state. We believe these are novel and important insights that are indeed adequately supported by our experimental evidence.

8. The authors conclude on p. 12 that “EZH2 gain-of-function mutations are necessary and sufficient to create a distorted H3K27me3 landscape in FL, independently of the other genomic anomalies”, but in fact only show that the gain-of-function mutation can explain a distorted H3K27me3 landscape in FL. The iMEF experiments go further (“necessary and sufficient”), but it would take a rescue experiment to establish the same for FL. I would suggest to simply rephrase this particular conclusion.

We agree that in the absence of a genetic manipulation the criteria for a “necessary and sufficient” statement are not met, and we thank the reviewer for this important correction. We have rephrased the sentence in question as follows:

“We conclude that *EZH2* gain-of-function mutations are consistently associated with a distorted H3K27me3 landscape in FL, independently of the other genomic anomalies...”

REVIEWER COMMENTS

Reviewer #1 (Remarks to the Author):

I would like to express my gratitude to the authors for their efforts in addressing all the questions and comments raised during the initial review. I particularly appreciate the authors' positive attitude towards improving their manuscript and the insightful scientific discussion that has emerged throughout this review process. I hope that my comments contribute to the publication of this work in Nature Communications.

The raised concerns have been thoroughly addressed in the revised manuscript and the rebuttal letter. I would like just to comment on the cell line used in points 1 and 2.

I agree with the authors that the use of the Oci-Ly19 cell line appears to be a valuable model to address the first and second points. I fully understand that the current available Follicular Lymphoma cell lines do not fully recapitulate the disease, but I have a question regarding the Oci-Ly19 cell line: This GCB-DLBCL cell line is classified as a High-grade B-cell lymphoma (HGBL) with BCL2 and BCL6 rearrangements in this case. Additionally, it harbors multiple copies of MYC gene (3-4), leading to high expression levels of MYC (1) (Expression data of MYC obtained from <https://www.proteinatlas.org/>).

Considering that BCL6 and MYC play an important role in epigenetic regulation, and particularly in EZH2 regulation, how might this condition affect to the effect of EZH2 mutations in H3K27 trimethylation and therefore the alteration in gene expression patterns?

(1) Li W, Gupta SK, Han W, Kundson RA, Nelson S, Knutson D, Greipp PT, Elswa SF, Sotomayor EM, Gupta M. Targeting MYC activity in double-hit lymphoma with MYC and BCL2 and/or BCL6 rearrangements with epigenetic bromodomain inhibitors. *J Hematol Oncol*. 2019 Jul 9;12(1):73. doi: 10.1186/s13045-019-0761-2. PMID: 31288832; PMCID: PMC6617630.

Reviewer #2 (Remarks to the Author):

The authors have responded to many of the points raised on the first submission of their manuscript. The authors extract two seemingly conflicting ideas from the review, aiming to highlight a potential contradiction in the comments. First, the degree of methylation loss for H3.3K27M, EZH2, and EED KO data are consistent with previous studies (original figure 1A-C). However, the data in original figure 1 D-E are not consistent with previous studies.

Again, the data do not substantiate the claim that H3.3K27M induces a 'straightforward' loss of function in PRC2. In Figure 1B, the limited overlap between EED KO and EZH2 KO with H3.3K27M is evident. In Figure 1D, peaks of H3K27me3 are observed in H3.3K27M but not in EZH2 and EED KOs. The authors rely on the western blot data in Figure 1A as the primary argument for the apparent functional equivalence of loss of EED and EZH2, and H3.3K27M.

The authors state that 'non-saturating doses' of UNC1999 also recapitulate the same H3K27me3 profile as H3.3K27M, but they do not show the data in Figure 1B or Figure 1D. Instead, they point to a heatmap in SupFigure 1H lacks the resolution to adequately compare the ChIP seq plots. The range of the plots are marked 'beginning and end', and even with the low resolution heat map, one can see differences in the patterns between the H3.3K27M and UNC1999 datasets. The authors should include IGV screenshots of H3K27me3 peaks in figure 1D. Additionally, an RNA-seq dataset from UNC1999 could be compared to the EZH2-KO, EED-KO, and H3.3K27M gene expression profiles for figure 1B (and Supplemental 1D).

The new data with UNC1999 stand in stark contrast to the findings by Mohammad et al. in Nature Medicine 2017. In that study, it was revealed that EZP6438, an inhibitor closely resembling UNC1999, failed to replicate H3.3K27M. Instead, the combination of H3.3K27M and EZP6438 resulted in distinctly different patterns of H3K27me3 in cells (refer to Figure 5A-C, Mohammad et al. Nature Medicine 2017).

As previously noted, the manuscript's strength lies in the comprehensive collection of follicular lymphoma patient samples. However, notable weaknesses persist, such as the absence of new mechanistic insights into histone or EZH2 mutations and a limited comprehension of the disease mechanism involving EZH2 mutations in follicular lymphoma.

Reviewer #3 (Remarks to the Author):

Many of my previous comments have been addressed in some way. I generally support publication, but I still have some comments.

1. Regarding the normalization issues, it is satisfying to see that the spike-in controls for the ChIP support the reduction in signal claimed in the paper, even though the effect size would be reduced by a sizable 50% (~ 2 -fold effect rather than the ~ 4 -fold effect in Fig. S2c (third panel from the left)). The authors suggest that their normalization method with the csaw bioconductor package works better. For one particular locus (*foxa1*, figure shown in rebuttal) this is shown by qPCR, however the fold change with csaw and spike-in scaling factors for this locus are not compared to support the claim. As a broader point, many people (and reviewers) in the field consider spike-ins the golden standard for normalization/scaling, whereas in our experience the use of spike-ins can be tricky. Based on the rebuttal, I take it the authors would agree. It would be worthwhile documenting this in the paper by including the table with the scaling factors and the PCR result as supplemental material (now only provided in the rebuttal).

2. The authors have clarified what they mean by the indirect effects that are maintained at the transcriptional level, even if the H3K27me3 levels are restored pharmacologically. Their explanation that some transcription factors (TFs) may have become derepressed and stabilize the transcriptomic changes seems plausible, but carries some implications that could have been addressed with the data the authors already have. What specific transcription factors likely play a role in this? Is there any supporting evidence in terms of TF binding motif enrichment in the regulatory elements of the (stably) deregulated (or upregulated) genes? Are any of the candidate TFs Polycomb targets themselves? If so, are they exceptions to the global pharmacological rescue of H3K27me3 patterns (subject to increased activation themselves, breaking the feedback loop)? These kinds of analyses would have provided some mechanistic insight in terms of a shifted balance of activation and repression mediated by transcription factors. In the absence of such analyses, a claim of mechanistic insight (for example in the last sentence of the abstract: "our results provide a mechanistic foundation") should not be made.

We thank the reviewers for this additional opportunity to improve our manuscript and clarify our conclusions. Below is our point-by-point response to their remarks.

Reviewer #1 (Remarks to the Author):

I would like to express my gratitude to the authors for their efforts in addressing all the questions and comments raised during the initial review. I particularly appreciate the authors' positive attitude towards improving their manuscript and the insightful scientific discussion that has emerged throughout this review process. I hope that my comments contribute to the publication of this work in Nature Communications.

We, in turn, wish to thank the reviewer for his/her constructive critique of our manuscript, for the stimulating conversation and improved study that resulted, and for these kind words.

The raised concerns have been thoroughly addressed in the revised manuscript and the rebuttal letter. I would like just to comment on the cell line used in points 1 and 2. I agree with the authors that the use of the Oci-Ly19 cell line appears to be a valuable model to address the first and second points. I fully understand that the current available Follicular Lymphoma cell lines do not fully recapitulate the disease, but I have a question regarding the Oci-Ly19 cell line: This GCB-DLBCL cell line is classified as a High-grade B-cell lymphoma (HGBL) with BCL2 and BCL6 rearrangements in this case. Additionally, it harbors multiple copies of MYC gene (3-4), leading to high expression levels of MYC (1) (Expression data of MYC obtained from <https://www.proteinatlas.org/>). Considering that BCL6 and MYC play an important role in epigenetic regulation, and particularly in EZH2 regulation, how might this condition affect to the effect of EZH2 mutations in H3K27 trimethylation and therefore the alteration in gene expression patterns?

(1) Li W, Gupta SK, Han W, Kundson RA, Nelson S, Knutson D, Greipp PT, ElSawa SF, Sotomayor EM, Gupta M. Targeting MYC activity in double-hit lymphoma with MYC and BCL2 and/or BCL6 rearrangements with epigenetic bromodomain inhibitors. *J Hematol Oncol.* 2019 Jul 9;12(1):73. doi: 10.1186/s13045-019-0761-2. PMID: 31288832; PMCID: PMC6617630.

We agree with the reviewer that the OCI-Ly19 GCB-DLBCL cell line presents some limitations for trying to model Follicular Lymphoma. Among them is the question of the proliferation rate which is particularly elevated in OCI-Ly19, likely as a direct consequence of *c-MYC* and *BCL6* rearrangements. Since *EZH2* expression is coupled to proliferation (through the pRB-E2F pathway), it is likely that *EZH2* is more abundant in OCI-Ly19 than in Follicular Lymphoma. This difference in *EZH2* levels might indeed, as the reviewer envisions, have some impact on the extent to which the *EZH2* mutation alters the chromatin landscape and gene expression patterns. It is not straightforward, however, to evaluate this possibility directly using our currently available tools. Nevertheless, the flattening of H3K27me3 deposition that we observed in *EZH2*-mutant patient biopsies is a highly penetrant effect (Fig. 4g and Supplementary Fig. 4e). Considering that this finding is recapitulated in our

isogenic MEFs, which were immortalized with *c-Myc*, we cautiously propose that the overall character of the H3K27me3 changes is robust to differences in proliferation and to the levels of c-MYC and EZH2.

Reviewer #2 (Remarks to the Author):

The authors have responded to many of the points raised on the first submission of their manuscript. The authors extract two seemingly conflicting ideas from the review, aiming to highlight a potential contradiction in the comments. First, the degree of methylation loss for H3.3K27M, EZH2, and EED KO data are consistent with previous studies (original figure 1A-C). However, the data in original figure 1 D-E are not consistent with previous studies.

We thank the reviewer for acknowledging our efforts to address a number of the concerns he/she raised on the first submission. We regret that we have not (yet) managed to persuade him/her of our interpretation of the effects of *H3.3K27M* on PRC2 function, and further strive to do so in our specific responses below.

Again, the data do not substantiate the claim that H3.3K27M induces a 'straightforward' loss of function in PRC2. In Figure 1B, the limited overlap between EED KO and EZH2 KO with H3.3K27M is evident.

If we understand the reviewer's logic correctly, we should expect partial inhibition of PRC2 (such as that we argue occurs in the presence of H3.3K27M) to result in differentially expressed genes (DEGs) that are essentially a strict subset of those observed upon full inhibition of PRC2 (such as functionally occurs in the *Eed* KO).

We do not agree with this view. Indeed, it rests on the assumption that each target gene is regulated independently by PRC2 and that each gene's behavior in different conditions is a simple function of its own individual threshold of sensitivity to reductions in PRC2 activity. We do not consider this to be a fair assumption.

Instead, because PRC2 target genes include many genes encoding transcription factors, we expect that a given degree of PRC2 inhibition will have a complex array of direct and indirect impacts on gene expression. This can readily explain how a gene can be up-regulated upon partial inhibition of PRC2 (because of loss of silencing by PRC2) but not upon full inhibition (while loss of silencing by PRC2 still occurs, it does not translate into gene activation as the expression of the activator controlling its transcription has changed in this condition).

In direct support of this model, we observed only a partial overlap between genes up-regulated in WT cells treated with 0,25mM UNC1999 *versus* genes up-regulated in cells treated with 2 mM UNC1999 (see diagram below, comparing the sets of 98 and 361 genes represented respectively by the dark green and light green circles in Fig. 3c of the manuscript).

In light of these observations, we do not think that the lack of a full overlap between genes up-regulated in *H3.3K27M*-expressing cells and in *Eed*-KO cells in any way undermines our conclusion that *H3.3K27M* induces a PRC2 loss of function.

In Figure 1D, peaks of H3K27me3 are observed in *H3.3K27M* but not in EZH2 and EED KOs. The authors rely on the western blot data in Figure 1A as the primary argument for the apparent functional equivalence of loss of EED and EZH2, and *H3.3K27M*.

Regarding the first sentence, indeed H3K27me3 peaks are still observed in *H3.3K27M* but not in *Ezh2*- and *Eed*-KO conditions. This is fully consistent with our argument that *H3.3K27M* causes a loss of function of PRC2 that is less severe than that caused by deletion of *Ezh2* or *Eed*.

Regarding the second sentence, we respectfully disagree with the reviewer, as the “functional equivalence” (i.e., that *H3.3K27M*, *Ezh2* KO and *Eed* KO all cause a loss of PRC2 function, albeit, as we emphasize, to different degrees) is supported by several lines of concordant evidence:

- 1) *In vitro* assays that previously established that H3K27M inhibits PRC2 activity^{1,2}.
- 2) Western blots from previous publications¹⁻³ as well as those shown in Fig. 1a and Supplementary Fig. 1c, which all demonstrate that the *H3.3K27M* mutation leads to a reduced global level of H3K27me3.
- 3) Our CUT&RUN analysis reported in Fig. 1d-f and Supplementary Fig. 1g, h showing that the *H3.3K27M* mutation leads to a reduced enrichment of H3K27me3 at chromatin.
- 4) Our transcriptomic analysis shown in Fig. 1b,c.

References:

1. Bender, S. et al. Reduced H3K27me3 and DNA hypomethylation are major drivers of gene expression in K27M mutant pediatric high-grade gliomas. *Cancer Cell* **24**, 660–672 (2013).
2. Lewis, P. W. et al. Inhibition of PRC2 activity by a gain-of-function H3 mutation found in pediatric glioblastoma. *Science* **340**, 857–861 (2013).
3. Chan, K. M. et al. The histone H3.3K27M mutation in pediatric glioma reprograms H3K27 methylation and gene expression. *Genes Dev* **27**, 985–990 (2013).

The authors state that 'non-saturating doses' of UNC1999 also recapitulate the same H3K27me3 profile as H3.3K27M, but they do not show the data in Figure 1B or Figure 1D.

Indeed, these experiments were done for another purpose and are presented in more detail in Figs. 2 and 3. Besides, they were not performed concurrently with the experiments displayed in Fig. 1b, d, and while the overall profiles are highly consistent from one experiment to another, local variations due to cell culture conditions can arise. For this reason, we prefer to show a genome-wide view of these data, matched to their respective controls cultured in parallel, as we did in Fig. 1g and Supplementary Fig. 1h.

Instead, they point to a heatmap in SupFigure 1H lacks the resolution to adequately compare the ChIP seq plots.

Supplementary Fig. 1h provides a comprehensive comparison of the H3K27me3 ChIP-seq data in the *H3.3K27M* and 0.25 μ M UNC1999 conditions, revealing the extent of their similarity across the genome. Bigwig files are available as part of our GEO submission for an easy and unbiased evaluation of the plots at high resolution.

The range of the plots are marked 'beginning and end', and even with the low resolution heat map, one can see differences in the patterns between the H3.3K27M and UNC1999 datasets.

As indicated in the figure legend, the heatmaps represent peaks which by essence are of different sizes and have been rescaled in order to be displayed together. A uniform scale with base-pair distances therefore cannot be provided.

Also, the degrees of PRC2 inhibition by *H3.3K27M* and 0.25 μ M UNC1999 are in the same range; however, they are not identical, and slight variations are therefore fully expected. Nevertheless, the overall trend displayed by the two conditions is essentially the same.

The authors should include IGV screenshots of H3K27me3 peaks in figure 1D. Additionally, an RNA-seq dataset from UNC1999 could be compared to the EZH2-KO, EED-KO, and H3.3K27M gene expression profiles for figure 1B (and Supplemental 1D).

For the reasons described above, we prefer not to mix data from sets of experiments not conducted in parallel within the same figure in the manuscript.

Nonetheless, the reviewer can see in the figure below (same genomic region as Fig. 1d) that the retention of a subset of peaks is a common feature of both the *H3.3K27M* and partial EZH2 inhibition conditions.

H3K27me3, comparison data from Figures 1 and 2

Furthermore, following the reviewer's suggestion to compare the corresponding RNA-seq data, we performed pre-ranked gene-set enrichment analysis (GSEA). As shown below, we find that, indeed, genes up-regulated upon treatment with 0.25 μ M UNC1999 are significantly enriched among genes up-regulated in *H3.3K27M*-expressing cells, while conversely genes down-regulated upon treatment with 0.25 μ M UNC1999 are significantly enriched among genes down-regulated in *H3.3K27M*-expressing cells. We conclude that the impact of *H3.3K27M* on overall gene expression is comparable to that of partially inhibiting PRC2.

The new data with UNC1999 stand in stark contrast to the findings by Mohammad et al. in Nature Medicine 2017. In that study, it was revealed that EZP6438, an inhibitor closely resembling UNC1999, failed to replicate H3.3K27M. Instead, the combination of H3.3K27M and EZP6438 resulted in distinctly different patterns of H3K27me3 in cells (refer to Figure 5A-C, Mohammad et al. Nature Medicine 2017).

We would like to stress that the experimental conditions under consideration in the two studies are not comparable. In Mohammad et al., the authors compared the effect of the *H3.3K27M* mutation to a full inhibition of PRC2 using 3 μ M EPZ6438 (see H3K27me3 Western blot in Fig. 4c of their article, pertaining to the same cell type as that analyzed by ChIP-seq in Fig. 5a-c mentioned by the reviewer). In contrast, we compared *H3.3K27M* to a partial inhibition of PRC2 using 0.25 μ M UNC1999. The prominent resemblance we observe between *H3.3K27M* and partial PRC2 inhibition does not contradict the dissimilarity that Mohammad et al. observed between *H3.3K27M* and full inhibition.

As previously noted, the manuscript's strength lies in the comprehensive collection of follicular lymphoma patient samples. However, notable weaknesses persist, such as the absence of new mechanistic insights into histone or EZH2 mutations and a limited comprehension of the disease mechanism involving EZH2 mutations in follicular lymphoma.

Reviewer #3 (Remarks to the Author):

Many of my previous comments have been addressed in some way. I generally support publication, but I still have some comments.

1. Regarding the normalization issues, it is satisfying to see that the spike-in controls for the ChIP support the reduction in signal claimed in the paper, even though the effect size would be reduced by a sizable 50% (~2-fold effect rather than the ~4-fold effect in Fig. S2c (third panel from the left)). The authors suggest that their normalization method with the csaw bioconductor package works better. For one particular locus (*foxa1*, figure shown in rebuttal) this is shown by qPCR, however the fold change with csaw and spike-in scaling factors for this locus are not compared to support the claim. As a broader point, many people (and reviewers) in the field consider spike-ins the golden standard for normalization/scaling, whereas in our experience the use of spike-ins can be tricky. Based on the rebuttal, I take it the authors would agree. It would be worthwhile documenting this in the paper by including the table with the scaling factors and the PCR result as supplemental material (now only provided in the rebuttal).

As the reviewer surmised, we have also found that the use of spike-ins is not always as straightforward as might first appear on paper. We agree that the comparison we made between normalization methods is worth including in the manuscript and therefore now present the table with the scaling factors as Supplementary Table 1.

The CUT&RUN-qPCR experiment we provided in our previous rebuttal was conducted separately from the main series of CUT&RUN analyses presented in the study. Consequently, we think its place is within this exchange, which will be publicly available in the event of publication, rather than in the manuscript proper. We sequenced the corresponding samples and the enrichment patterns over *Foxa1* most closely match the qPCR results when csaw rather than spike-in is used for normalization, as shown below. This further strengthens our confidence in our normalization strategy for this study.

We should note, however, that in the absence of a more exhaustive look into this matter, we refrain from making more general claims regarding the relative soundness of different normalization methods used elsewhere in the literature. Each has potential merits and drawbacks, and broader conclusions on this point are beyond the scope of our study.

2. The authors have clarified what they mean by the indirect effects that are maintained at the transcriptional level, even if the H3K27me3 levels are restored pharmacologically. Their explanation that some transcription factors (TFs) may have become derepressed and stabilize the transcriptomic changes seems plausible, but carries some implications that could have been addressed with the data the authors already have. What specific transcription factors likely play a role in this? Is there any supporting evidence in terms of TF binding motif enrichment in the regulatory elements of the (stably) deregulated (or upregulated) genes? Are any of the candidate TFs Polycomb targets themselves? If so, are they exceptions to the global pharmacological rescue of H3K27me3 patterns (subject to increased activation themselves, breaking the feedback loop)? These kinds of analyses would have provided some

mechanistic insight in terms of a shifted balance of activation and repression mediated by transcription factors. In the absence of such analyses, a claim of mechanistic insight (for example in the last sentence of the abstract: “our results provide a mechanistic foundation”) should not be made.

We are grateful to the reviewer for considering our proposed model a reasonable possibility and we acknowledge that we did not go to sufficient lengths to test it in the previous version of our manuscript. Following the reviewer’s suggestions, we searched across the promoters of all genes significantly de-regulated in iMEFs upon expression of *Ezh2*^{Y641F} (i.e., those displayed in the volcano plot in Fig. 3a of the manuscript) for significantly enriched TF binding motifs using MEME. We identified a number of enriched motifs whose corresponding TFs were also significantly up-regulated in *Ezh2*^{Y641F}-expressing cells. We show these results in Fig. 3g of the newly updated version of the manuscript. As the reviewer anticipated, a handful of these TFs appear to be direct targets of PRC2 (with a H3K27me3 peak at their promoter in WT cells) and, as shown in Fig. 3h, three of these lie at the center of a predicted protein-protein interaction network. Two of these three (*Etv4* and *Foxa1*, but interestingly not *Gata3*), fail to regain local H3K27me3 patterns in *Ezh2*^{Y641F}-expressing cells upon treatment with 0.25 μM UNC1999 (Supplementary Fig. 3c), in contrast to the general trend of H3K27me3 restoration observed across the genome (Fig. 2j). Altogether, we believe that this new analysis supports the idea that mutant EZH2 leads to persistent alterations in the TF environment that could help explain why partial PRC2 inhibition does not reverse most of the mutant-induced changes in gene expression even though a wild-type-like regime of PRC2 activity is reinstated.

REVIEWERS' COMMENTS

Reviewer #1 (Remarks to the Author):

I have no further comments on this manuscript.

Reviewer #2 (Remarks to the Author):

This study excels in its comprehensive compilation of follicular lymphoma patient samples, and it significantly advances our comprehension of the disease mechanism associated with EZH2 mutations in follicular lymphoma.

The authors are strongly encouraged to incorporate the IGV screenshot displaying the H3K27me3 peaks from Figure 1D, as presented in the rebuttal letter. This inclusion will allow readers to evaluate the comparability of the H3K27me3 ChIP-Seq patterns between 'K27M' and '0.25 μ M UNC1999.' The authors assert that K27M signifies a straightforward loss of function, essentially equating it to the impact of adding a subsaturating dose of UNC1999. It is imperative that readers have the opportunity to scrutinize the evidence and draw their own conclusions regarding this argument.

Reviewer #3 (Remarks to the Author):

No further comments.

We thank the reviewers for their constructive feedback on the manuscript throughout the peer review process. We respond to their final comments below.

REVIEWERS' COMMENTS

Reviewer #1 (Remarks to the Author):

I have no further comments on this manuscript.

Reviewer #2 (Remarks to the Author):

This study excels in its comprehensive compilation of follicular lymphoma patient samples, and it significantly advances our comprehension of the disease mechanism associated with EZH2 mutations in follicular lymphoma.

We thank the reviewer for this positive remark.

The authors are strongly encouraged to incorporate the IGV screenshot displaying the H3K27me3 peaks from Figure 1D, as presented in the rebuttal letter. This inclusion will allow readers to evaluate the comparability of the H3K27me3 ChIP-Seq patterns between 'K27M' and '0.25 μ M UNC1999.' The authors assert that K27M signifies a straightforward loss of function, essentially equating it to the impact of adding a subsaturating dose of UNC1999. It is imperative that readers have the opportunity to scrutinize the evidence and draw their own conclusions regarding this argument.

We agree that it is worthwhile to include this comparison explicitly in the manuscript and are confident that it will lead readers to share our interpretation. The relevant ChIP-seq tracks are now displayed together in Supplementary Fig. 1i.

Reviewer #3 (Remarks to the Author):

No further comments.